# LW2G: Learning Whether to Grow for Prompt-based Continual Learning

## Abstract

Continual Learning (CL) aims to learn in non-stationary scenarios, progressively acquiring and maintaining knowledge from sequential tasks. Recent Prompt-based Continual Learning (PCL) has achieved remarkable performance with Pre-Trained Models (PTMs). These approaches grow a prompt sets pool by adding a new set of prompts when learning each new task (*prompt learning*) and adopt a matching mechanism to select the correct set for each testing sample (*prompt retrieval*). Previous studies focus on the latter stage by improving the matching mechanism to enhance Prompt Retrieval Accuracy (PRA). To promote cross-task knowledge facilitation and form an effective and efficient prompt sets pool, we propose a plug-in module in the former stage to **Learn Whether to Grow (LW2G)** based on the disparities between tasks. Specifically, a shared set of prompts is utilized when several tasks share certain commonalities, and a new set is added when there are significant differences between the new task and previous tasks. Inspired by Gradient Projection Continual Learning, our LW2G develops a metric called Hinder Forward Capability (HFC) to measure the hindrance imposed on learning new tasks by surgically modifying the original gradient onto the orthogonal complement of the old feature space. With HFC, an automated scheme Dynamic Growing Approach adaptively learns whether to grow with a dynamic threshold. Furthermore, we design a gradient-based constraint to ensure the consistency between the updating prompts and pre-trained knowledge, and a prompts weights reusing strategy to enhance forward transfer. Extensive experiments show the effectiveness of our method.

## 1 Introduction

Compared to learning in stationary scenarios, Continual Learning (CL) equips systems with the ability to learn in non-stationary environments, which is a core step toward achieving human-level intelligence and human-like adaptation. In this learning paradigm, Deep Neural Networks (DNNs) need to learn from a sequential tasks while retaining past knowledge and acquiring novel knowledge. However, simply utilizing standard optimization methods Diederik (2014); Ruder (2016) for training DNNs inevitably erases the parametric representations of old tasks with new input representations during updating. Therefore, a well-known problem Catastrophic Forgetting (CF) arises French (1999); Ramasesh et al. (2021); McCloskey & Cohen (1989); Rebuffi et al. (2017); Lewandowsky & Li (1995), where DNNs suffer severe performance degradation on old tasks due to the absence of old data and domain shift in data distributions, making CL an extremely challenging problem.

Recently, **Prompt-based Continual Learning (PCL)** offers fresh insights into addressing CF Wang et al. (2024a); Douillard et al. (2022); Smith et al. (2023b); Zhou et al. (2023a); Wang et al. (2022a;b); Zhou et al. (2022). These methods leverage frozen Pre-Trained Models (PTMs) rather than training from scratch and employ Parameter-Efficient Fine-Tuning techniques (PEFTs) (Zhu et al., 2023; Dettmers et al., 2024; Wang et al., 2020; Houlsby et al., 2019; Jia et al., 2022; Hu et al., 2021), e.g., prompt. Specifically, PCL involves two stages: (a) *prompt learning*: learning a task-wised set of prompts to conditionally guide the PTM for the current task, which are stored in an expanding prompt sets pool, and (b) *prompt retrieval*: predicting which task each testing sample belongs to and choosing the corresponding prompt set. Recent studies Wang et al. (2024a); Huang et al. (2024); Tran et al. (2023) have found that Prompt Retrieval Accuracy (PRA) can significantly influence the performance, since an incorrect set for the testing samples results in a performance decline.

Additionally, learning each task individually not only limits the potential for cross-task knowledge facilitation but also leads to parameter redundancy Yu et al. (2024); Rypeść et al. (2024).

One simple solution to this problem is to mimic humans' integration of information Roediger & McDermott (1995); Hunt (2006); Arndt (2006). For instance, when several tasks share certain commonalities, they can use a shared set of prompts. However, when tasks differ significantly, a new set should be added. Thus, by adaptively learning whether to grow a new set for PCL, the amount

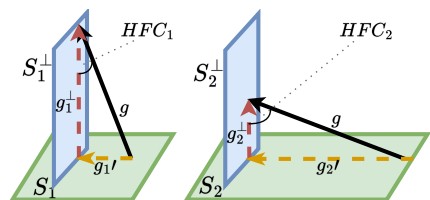

Figure 1: Illustration of HFC. $\mathcal{S}_i$ represents the feature space spanned by the old task $i$, while $\mathcal{S}_i^\perp$ denotes the orthogonal complement to $\mathcal{S}_i$. Then, HFC$(\boldsymbol{g}, \boldsymbol{g}_i^\perp)$ is denoted as HFC$_i$.

of selectable options is reduced, and the divergence between sets is increased, thereby improving PRA. Furthermore, aggregating multiple tasks' knowledge into a single set can also facilitate mutual knowledge utilization and promotion among tasks. Nevertheless, establishing suitable metrics to measure this commonality and obtaining task information *a priori* – all of which are challenging in practice. Moreover, gradually integrating knowledge from multiple tasks into a single set also presents an unresolved query, as the knowledge from different tasks can interfere with each other during sequential learning.

Thanks to Gradient Projection-based Continual Learning (GPCL) Zhao et al. (2023); Saha et al. (2021); Lopez-Paz & Ranzato (2017), which proposes that learning would not forget if the updated gradient is orthogonal to the feature space spanned by old tasks (denoted as *orthogonal condition*), we propose to use the *orthogonal condition* in GPCL to integrate the knowledge from multiple tasks into a single set of prompts. Specifically, in Figure 1, the gradient $\boldsymbol{g}$ of the new task is modified to its projection $\boldsymbol{g}_1^\perp$ onto $\mathcal{S}_1^\perp$, and $\boldsymbol{g}_1^\perp$ serves as the real gradient for updating parameters, thereby reducing the forgetting of old knowledge in task 1. Furthermore, to address the dilemma of whether *to grow* (i.e., initializing a new set of prompts) or *not to grow* (i.e., selecting an old set of prompts from the pool), we introduce a novel metric called **Hinder Forward Capability (HFC)**. *HFC is calculated as the angle $\theta$ between the gradient of the new task $\boldsymbol{g}$ and its' projection $\boldsymbol{g}^\perp$.* As illustrated in Figure 1, as HFC$_1$ < HFC$_2$ then $\boldsymbol{g}_1^\perp > \boldsymbol{g}_2^\perp$, it implies that the hindrance to learning on the set of prompts to task 2 is larger than that on the set of prompts to task 1 when updating under the *orthogonal condition*. Thus, when the hindrance on learning a new task is severe, PCL should choose *to grow* a new set; conversely, it tends *not to grow*. Meanwhile, $\boldsymbol{g}$ presents a large projection onto $\mathcal{S}_2$ indicating higher similarity between the new task and task 2 than with task 1.

Based on the analysis, we propose a plug-in module within PCL to **Learn Whether to Grow (LW2G)**, consisting of three components: Dynamic Growing Approach (DGA), Consistency with Pre-trained Knowledge (CPK), and Facilitation for Forward Transfer (FFT). DGA is an automated scheme to learn whether *to grow* (adopt a new set of prompts and store it in the pool) or *not to grow* (utilize an existing set of prompts from the pool) for new tasks based on the introduced HFC metric. Specifically, to incorporate knowledge from multiple tasks into a single set of prompts, we first employ the *orthogonal condition* to learn new tasks without forgetting and calculate the hindrance on learning with each set in the pool through HFC. Meanwhile, we consider an ideal scenario to generate a dynamic threshold, which learn the new task on the pre-trained knowledge feature space $\mathcal{S}^{\text{pre}}$ without any obstacles from old tasks. DGA chooses *to grow* if all HFC values are above this threshold, indicating that learning with each set in the pool encounters excessive hindrance. Conversely, DGA chooses *not to grow* by selecting the old set of prompts with the minimum HFC and learning the new task under the *orthogonal condition*. CPK aims to balance the disruption to pre-trained knowledge caused by continual learning on new tasks and the reduced plasticity brought by strict orthogonality to the entire pre-trained feature space $\mathcal{S}^{\text{pre}}$. Therefore, we propose applying a soft constraint to the gradient when learning new tasks, aiming to align the gradient direction as closely as possible with the feature space of the pre-trained knowledge, ensuring consistency between prompt updates and pre-trained knowledge. Finally, FFT reuses the frozen weights from the existing set of prompts with the maximum HFC to enhance forward transfer.

The contributions of this paper can be summarized as follows:

- We propose an automated learning scheme within PCL, by learning whether to grow or not to grow set of prompts. We aim to form an effective and efficient prompt sets pool where each single set contains knowledge from multiple tasks, thus facilitating cross-task promotion.

- We introduce HFC metric, which not only measures the difference between new and old tasks but also evaluates the hindrance on learning new tasks under the strict *orthogonal condition*.

- LW2G is a plug-in module within existing PCL. Extensive experiments demonstrate its superiority across multiple benchmarks and various CL settings.

## 2  RELATED WORK

**Continual Learning and Gradient Projection**    Numerous efforts have been made to alleviate the core issue of CF French (1999); Ramasesh et al. (2021); McCloskey & Cohen (1989), which can be roughly categorized into three main categories: (1) Architecture-based, (2) Rehearsal-based, and (3) Regularization-based. Architecture-based methods Rusu et al. (2016); Yoon et al. (2017); Li et al. (2019); Loo et al. (2020); Mallya & Lazebnik (2018); Serra et al. (2018); Ke et al. (2020) segregate components within the DNNs for each task by expanding the model or constraining the learning rate of part of parameters. However, most of them designed for Task-CL, which is not suitable for challenging Class-CL. Rehearsal-based methods Buzzega et al. (2020); Cha et al. (2021); Rebuffi et al. (2017); Wu et al. (2019); Ebrahimi et al. (2020); Pham et al. (2021); Zhao et al. (2021); De Lange et al. (2021); Wang et al. (2018) mitigate forgetting by replaying real or generated samples of old tasks, which raises concerns about efficiency and privacy. Regularization-based methods Kirkpatrick et al. (2017); Zenke et al. (2017) achieve a balance between new and old tasks by designing sophisticated regularization terms. Among them, GPCL methods Zhao et al. (2023); Saha et al. (2021); Lopez-Paz & Ranzato (2017); Qiao et al. (2023); Lin et al. (2022b;a); Zhu et al. (2023); Yu et al. (2020); Wang et al. (2021); Duncker et al. (2020); Wang et al. (2023); Smith et al. (2023a); Chen et al. (2020; 2022) focus on the gradient of the parameter. These methods project the gradient orthogonally to the feature space spanned by the old tasks, thereby not affecting the old knowledge.

**Prompt-based Methods and Transfer Learning**    PCL garnered significant attention due to their utilization of PEFT techniques (Zhu et al., 2023; Dettmers et al., 2024; Wang et al., 2020; Houlsby et al., 2019; Jia et al., 2022; Hu et al., 2021; Yang et al., 2024) to leverage PTMs, achieving rehearsal-free and promising performance Wang et al. (2024a); Douillard et al. (2022); Smith et al. (2023b); Zhou et al. (2023a); Wang et al. (2022a;b); Zhou et al. (2022); Qiao et al. (2023); Wang et al. (2022c); Huang et al. (2024); Zhou et al. (2024b;a; 2023b). Among them, DualPrompt Wang et al. (2022b) proposed partitioning the knowledge of tasks into general and specific categories, and learns them with g-prompt and e-prompt, respectively. Similarly, S-liPrompt and S-iPrompt Wang et al. (2022a) addressed Domain-CL by leveraging Vision-Language Models (VLMs) to further enhance the learning ability. CODAPrompt Smith et al. (2023b), S-Prompt++ Wang et al. (2024a) and HidePrompt Wang et al. (2024a) improved *prompt retrieval* stage through *attention mechanisms* and auxiliary adapter classifiers. Additionally, recent studies show that fine-tuning downstream tasks or continual learning with PTMs often leads to overfitting due to relatively limited downstream training data, resulting in degradation of pre-trained knowledge Lee et al. (2023); Li et al. (2024); Zheng et al. (2023); Zhu et al. (2023).

## 3  PRELIMINARIES AND NOTATIONS

**Continual Learning**    Assume there is a sequence of tasks and their corresponding training datasets $\left\{\mathcal{D}^i, i = 1, 2, \ldots\right\}$ without overlapping classes, where $\mathcal{D}^t = \{(\boldsymbol{x}_{i,t}, \boldsymbol{y}_{i,t})\}_{i=1}^{n_t}$ belongs to the task $t$. We denote the DNN as $\mathcal{W} = \left\{\theta^l\right\}_{l=1}^{L}$, where $\theta^l$ is the weight of layer $l$. Given a training sample $\boldsymbol{x}_{i,t}$, we denote $\boldsymbol{x}_{i,t}^l$ as the input of layer $l$ and the output is $\boldsymbol{x}_{i,t}^{l+1} = f^l\left(\theta^l, \boldsymbol{x}_{i,t}^l\right)$, where $f^l$ is the operation of layer $l$. We simplify the loss function for learning task $t$ as $\mathcal{L}_t(\mathcal{D}^t)$ and $\mathcal{W}_t = \left\{\theta_t^l\right\}_{l=1}^{L}$ as the DNN after training on task $t$.

**Gradient Projection Continual Learning**    [Revised:First, for $\boldsymbol{A} \in \mathbb{R}^{m \times n}$ and a subspace $\mathcal{S}$ in Euclidean space with its bases $\boldsymbol{B} \in \mathbb{R}^{n \times d}$, the projection of $\boldsymbol{A}$ onto the subspace $\mathcal{S}$ is denoted as

follows:]

$$\text{Proj}_{\mathcal{S}}(\boldsymbol{A}) = \boldsymbol{A}\boldsymbol{B}(\boldsymbol{B})^T, \tag{1}$$

where $(\cdot)^T$ is the matrix transpose.

[Revised: Then, following Saha et al. (2021), we briefly introduce how GPCL reduces the interference of old knowledge when learning new tasks. Specifically, the total process involves two stages.

**Stage (1) Building of the new feature space.** After training on task 1, for each layer GPCL construct a representation matrix $\boldsymbol{R}_1^l = \left[\boldsymbol{x}_{1,1}^l, \ldots, \boldsymbol{x}_{1,n}^l\right] \in \mathbb{R}^{n \times d}$ ($d$ is the output dimension of layer $l$) by concatenating representations of $n$ samples along the columns obtained from sending $n$ samples only from task 1 into the current DNN, $\mathcal{W}_1$. Next, GPCL perform SVD on $\boldsymbol{R}_1^l = \boldsymbol{U}_1^l \boldsymbol{\Sigma}_1^l (\boldsymbol{V}_1^l)^T$ followed by its $k$-rank approximation $(\boldsymbol{R}_1^l)_k$ according to the following criteria for the given threshold, $\epsilon_{\text{task}}$:

$$||(\boldsymbol{R}_1^l)_k||_F^2 \geq \epsilon_{\text{task}}||\boldsymbol{R}_1^l||_F^2. \tag{2}$$

Therefore, the feature space for layer $l$ is built by $\mathcal{S}_1^l = \text{span}\left\{\boldsymbol{B}_1^l\right\}$, where $\boldsymbol{B}_1^l = \left\{\boldsymbol{u}_1^l, \ldots, \boldsymbol{u}_k^l\right\}$ and $\boldsymbol{u}_i^l$ is the first $k$ vectors in $\boldsymbol{U}_1^l$. And $\mathcal{S}_1^l$ is stored in memory $\mathcal{M} = \left\{\mathcal{S}_1^l\right\}$.

When learning task 2, the gradient of layer $l$ is denoted as $\boldsymbol{g} = \nabla_{\theta^l} \mathcal{L}_2$. As illustrated in Figure 1, GPCL modify the gradient as follows:

$$\boldsymbol{g}_1^{\perp} = \text{Proj}_{\mathcal{S}_1^{\perp}}(\boldsymbol{g}), \tag{3}$$

where $\mathcal{S}_1^{\perp}$ is the orthogonal complement of $\mathcal{S}_1^l$ and $\boldsymbol{g}_1^{\perp}$ serves as the real gradient for updating layer $l$. Let $\Delta\theta_1^l$ denote the change in layer $l$ after learning task 2. For $\boldsymbol{x}_{i,1} \in \mathcal{S}_1^l$ from task 1, it follows that $\Delta\theta_1^l \boldsymbol{x}_{i,1} = 0$ due to the orthogonality of $\boldsymbol{g}_1^{\perp}$ with respect to $\mathcal{S}_1^l$ Zhang et al. (2021); Saha et al. (2021). Therefore, we can obtain:

$$\theta_2^l \boldsymbol{x}_{i,1}^l = (\theta_1^l + \Delta\theta_1^l)\boldsymbol{x}_{i,1}^l = \theta_1^l \boldsymbol{x}_{i,1}^l. \tag{4}$$

It demonstrates that there is no forgetting of knowledge of task 1, if the gradient for updating parameters is orthogonal to the old feature space. We denote the above condition as the *orthogonal condition*.

**Stage (2) Updating of old faeture space.** After learning task $i$, where $i \geq 2$, $\mathcal{S}_{i-1}^l$ in $\mathcal{M}$ needs to be updated to $\mathcal{S}_i^l$ with new task-specific bases from task $i$. To obtain such bases, for each layer $l$, we utilize the current DNN, $\mathcal{W}_i$, to construct a representation matrix $\boldsymbol{R}_i^l = \left[\boldsymbol{x}_{1,1}^l, \ldots, \boldsymbol{x}_{1,n}^l\right] \in \mathbb{R}^{n \times d}$ from task $i$ only. Before performing SVD and subsequent $k$-rank approximation, we first eliminate the common bases that already present in $\mathcal{S}_{i-1}^l$ so that newly added bases are unique and orthogonal to the existing bases in $\mathcal{S}_{i-1}^l$. To accomplish this, we proceed as follows:

$$\hat{\boldsymbol{R}}_i^l = \boldsymbol{R}_i^l - \boldsymbol{B}_{i-1}^l \left(\boldsymbol{B}_{i-1}^l\right)^T \left(\boldsymbol{R}_i^l\right) = \boldsymbol{R}_i^l - \boldsymbol{R}_{i,\text{proj}}^l. \tag{5}$$

Afterwards, SVD is performed on $\hat{\boldsymbol{R}}_i^l = \hat{\boldsymbol{U}}_i^l \hat{\boldsymbol{\Sigma}}_i^l (\hat{\boldsymbol{V}}_i^l)^T$, thus obtaining $h$ new orthogonal bases for minimun value of $h$ statisfying the following criteria for the given threshold, $\epsilon_{\text{task}}$:

$$||\boldsymbol{R}_{i,\text{proj}}^l||_F^2 + ||\hat{\boldsymbol{R}}_i^l||_F^2 \geq \epsilon_{\text{task}}||\boldsymbol{R}_i^l||_F^2. \tag{6}$$

$\boldsymbol{B}_{i-1}^l$ is then updated to $\boldsymbol{B}_i^l = \left[\boldsymbol{B}_{i-1}^l, \boldsymbol{u}_1^l, \ldots, \boldsymbol{u}_h^l\right]$ with $h$ new bases. And $\mathcal{S}_{i-1}^l$ is updated to $\mathcal{S}_i^l = \text{span}\left\{\boldsymbol{B}_i^l\right\}$. Details are in Appendix B.2.]

**Prompt-based Continual Learning** Recent studies Wang et al. (2024a); Smith et al. (2023b); Wang et al. (2022c;b;a) utilized prompts to leverage the PTMs. Therefore, the DNN is a Vision Transformer (ViT), and the operation of layer $l$, $f^l$, is the *attention mechanism* within each transformer block. Hence, the input of ViT after *patch embedding* is $\boldsymbol{x}_e \in \mathbb{R}^{L_e \times d}$, where $L_e$ is the token length. Specifically, VPT Jia et al. (2022); Li & Liang (2021) prepend a set of learnable tokens $\boldsymbol{p} \in \mathbb{R}^{L_p \times d}$ to $x_e$ and treat $[\boldsymbol{p}, \boldsymbol{x}_e] \in \mathbb{R}^{(L_e + L_p) \times d}$ as the input, minimizing $\mathcal{L}$ to encode task-specific knowledge into these prompts while keeping pre-trained weights frozen. PCL involves two stages: *prompt learning* and *prompt retrieval*. In *prompt learning*, PCL grows the prompt sets pool $\mathcal{P}$ by initializing a new set of prompt $(\boldsymbol{p}_i, \boldsymbol{k}_i)$ before learning each new task $i$, where $\boldsymbol{p}_i$ is combined with the training samples by the *attention mechanism*. Meanwhile, $\boldsymbol{k}_i$ is optimized by being pulled closer to the vanilla features of the training samples obtained by a ViT without combining with prompts. In *prompt retrieval*, $\boldsymbol{k}_i$ serves as the query vector for predicting which set of $\boldsymbol{p}_i$ to choose for each testing sample by a matching mechanism. More details are in Appendix C.

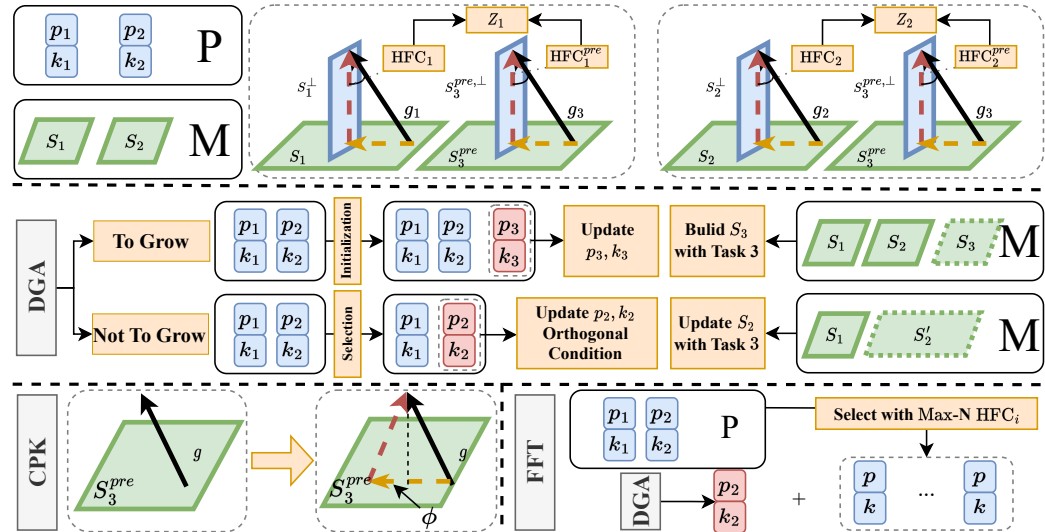

Figure 2: Illustration of three components in LW2G. Before learning task 3, assume there are two sets in $\mathcal{P} = \{(\boldsymbol{p}_1, \boldsymbol{k}_1), (\boldsymbol{p}_2, \boldsymbol{k}_2)\}$. In $\mathcal{P}$, blue represents frozen and unlearnable sets of prompts, whereas red represents learnable sets.

## 4 THEORY AND METHOD

In this section, we first present a theoretical analysis of GPCL concerning the hindrance on learning new tasks under the *orthogonal condition* (Theorem 1 and Definition 1). Subsequently, as illustrated in Figure 2, we introduce the plug-in module **Learning Whether to Grow (LW2G)**, which consists of three components: DGA, CPK, and FFT.

### 4.1 THEORETICAL ANALYSIS ON HINDRANCE IN GPCL

For simplicity, the notation of layer $l$ is omitted in the following analysis. While learning on task $i$, GPCL update the parameters under the *orthogonal condition* to avoid interfering with old knowledge. However, since the gradient represents the direction of local optimal descent for the loss function, modifying it inevitably results in a reduction of local information. To quantify the hindrance under the *orthogonal condition* in GPCL, we first define the following metric.

**Definition 1** (Hinder Forward Capability, HFC). *In GPCL, while continually encoding new knowledge into a single model under the orthogonal condition, Hinder Forward Capability (HFC) is defined to evaluate the hindrance on learning new tasks. HFC is the angle between the original gradient obtained through backpropagation $\boldsymbol{g}$ and its projection $\boldsymbol{g}^{\perp} = Proj_{\mathcal{S}_{old}^{\perp}}(\boldsymbol{g})$ onto $\mathcal{S}_{old}^{\perp}$,*

$$HFC(\boldsymbol{g}, \boldsymbol{g}^{\perp}) = \arccos\left(\frac{\mathbf{g} \cdot \mathbf{g}^{\perp}}{\|\mathbf{g}\|\|\mathbf{g}^{\perp}\|}\right).$$

As illustrated in Figure 1, a large HFC indicates a significant gap between original gradient $\boldsymbol{g}$ and the real gradient $\boldsymbol{g}^{\perp}$. Therefore, a large reduction of local information leads to greater hindrance on learning new tasks. Based on this, we formally present the following theorem (see Appendix B.1 for a detailed proof):

**Theorem 1.** *Given a space $\mathcal{S}_1 = span\{\boldsymbol{B}_1\}$, where $\boldsymbol{B}_1 = [\boldsymbol{b}_1, \ldots, \boldsymbol{b}_l] \in \mathbb{R}^{n \times l}$ is a set of $l$ bases for $\mathcal{S}_1$, and a space $\mathcal{S}_2 = span\{\boldsymbol{B}_2\}$, where $\boldsymbol{B}_2 = [\boldsymbol{b}_1, \ldots, \boldsymbol{b}_l, \boldsymbol{b}_{l+1}, \ldots, \boldsymbol{b}_{l+k}] \in \mathbb{R}^{n \times (l+k)}$ is a set of $l + k$ bases for $\mathcal{S}_2$. Then, $\forall \boldsymbol{\alpha}$ there always exists:*

$$HFC(\boldsymbol{\alpha}, Proj_{\mathcal{S}_1}(\boldsymbol{\alpha})) > HFC(\boldsymbol{\alpha}, Proj_{\mathcal{S}_2}(\boldsymbol{\alpha})).$$

The above Theorem 1 shows that fewer bases result in a larger HFC. As $\mathcal{S}_{old}$ in $\mathcal{M}$ continues to expand with new bases from each new task, its corresponding orthogonal complement $\mathcal{S}_{old}^{\perp}$ progressively shrinks. Consequently, the bases in $\mathcal{S}_{old}^{\perp}$ steadily decrease, leading to a large HFC and more severe hindrance on learning new tasks.

## 4.2 DYNAMIC GROWING APPROACH

Instead of naively growing a new set of prompts for each new task regardless of task dissimilarities, we propose a **Dynamic Growing Approach (DGA)**. DGA involves dynamically learning whether *to grow* (initialize a new set of prompts and store it in the pool) or *not to grow* (utilize an existing set from the pool).

For simplicity, we adopt an example with three tasks to illustrate our method in Figure 2. A more general description is presented in pseudocode, which can be found in Appendix A.

Before learning task 3, we first qualify the hindrance on each old set in the pool under the *orthogonal condition*. Specifically, we iteratively select an **old** set $(\boldsymbol{p}_1, \boldsymbol{k}_1)$ from $\mathcal{P}$ and $\mathcal{S}_1$ from $\mathcal{M}$, where $\mathcal{S}_1$ is the old feature space corresponding to task 1. We construct a subset of training dataset from task 3, denoted as $\mathcal{D}^3_{\text{sub}}$. For clarity, the gradient to update $(\boldsymbol{p}_1, \boldsymbol{k}_1)$ with $\mathcal{D}^3_{\text{sub}}$ is denoted as:

$$\boldsymbol{g}_1 = \nabla_{(\boldsymbol{p}_1, \boldsymbol{k}_1)} \mathcal{L}_3(\mathcal{D}^3_{\text{sub}}). \tag{7}$$

To prevent the influence of old knowledge contained in $(\boldsymbol{p}_1, \boldsymbol{k}_1)$ while learning task 3, the gradient $\boldsymbol{g}_1$ is required to be modified to $\text{Proj}_{\mathcal{S}_1^{\perp}}(\boldsymbol{g}_1)$, where $\mathcal{S}_1^{\perp}$ is the orthogonal complement of $\mathcal{S}_1$. Then, $\text{Proj}_{\mathcal{S}_1^{\perp}}(\boldsymbol{g}_1)$ serves as the real gradient for updating parameters. Based on Theorem 1, we evaluate the hindrance under the *orthogonal condition* while learning task 3 on $(\boldsymbol{p}_1, \boldsymbol{k}_1)$ as follows:

$$\text{HFC}_1 = \text{HFC}(\boldsymbol{g}_1, \text{Proj}_{\mathcal{S}_1^{\perp}}(\boldsymbol{g}_1)). \tag{8}$$

Besides, we define a dynamic threshold based on the task 3 and the PTM being used. Firstly, we initialize a **new** set with $(\boldsymbol{p}_1, \boldsymbol{k}_1)$ as follows:

$$(\boldsymbol{p}_3, \boldsymbol{k}_3) \Leftarrow (\boldsymbol{p}_1, \boldsymbol{k}_1). \tag{9}$$

Here, the newly initialized $(\boldsymbol{p}_3, \boldsymbol{k}_3)$ does not contain any knowledge from previous tasks (task 1 or task 2), which represents an ideal scenario for learning task 3. Likewise, the gradient to updated $(\boldsymbol{p}_3, \boldsymbol{k}_3)$ is denoted as:

$$\boldsymbol{g}_3 = \nabla_{(\boldsymbol{p}_3, \boldsymbol{k}_3)} \mathcal{L}_3(\mathcal{D}^3_{\text{sub}}). \tag{10}$$

Then, we can obtain a representation matrix $\boldsymbol{R}_3^{\text{pre}}$ by feeding $\mathcal{D}^3_{\text{sub}}$ into the ViT without prompts. We can newly build $\mathcal{S}_3^{\text{pre}}$ after performing SVD and $k$-rank approximation with pre-trained threshold, $\epsilon_{\text{pre}}$. Then, we can also calculate:

$$\text{HFC}_1^{\text{pre}} = \text{HFC}(\boldsymbol{g}_3, \text{Proj}_{\mathcal{S}_3^{\text{pre},\perp}}(\boldsymbol{g}_3)), \tag{11}$$

where $\mathcal{S}_3^{\text{pre},\perp}$ is the orthogonal complement of $\mathcal{S}_3^{\text{pre}}$. Here, $\text{HFC}_1^{\text{pre}}$ represents the relationship between the gradient of learning task 3 and the pre-trained knowledge from task 3. As $(\boldsymbol{p}_3, \boldsymbol{k}_3)$ is newly initialized specifically for training task 3, it contains no prior knowledge, and thus, there are no obstacles from old tasks. Therefore, $\text{HFC}_1^{\text{pre}}$ signifies the ideal scenario when learning new tasks in PCL, which is the *dynamic threshold to evaluate the relative magnitude of hindrance*. Based on this, the gap between learning on **old** set $(\boldsymbol{p}_1, \boldsymbol{k}_1)$ under the *orthogonal condition* and leaning on **new** set $(\boldsymbol{p}_3, \boldsymbol{k}_3)$ in an ideal scenario is denoted as follows:

$$Z_1 = \text{HFC}_1 - \text{HFC}_1^{\text{pre}}. \tag{12}$$

Thus, if $Z_1 > 0$, it indicates that learning on the **old** set $(\boldsymbol{p}_1, \boldsymbol{k}_1)$ from $\mathcal{P}$ encounters excessive hindrance.

Likewise, the gap between learning on **old** set $(\boldsymbol{p}_2, \boldsymbol{k}_2)$ under the *orthogonal condition* and leaning on **new** set $(\boldsymbol{p}_3, \boldsymbol{k}_3)$ in an ideal scenario can also be calculated as $Z_2$, where $(\boldsymbol{p}_3, \boldsymbol{k}_3)$ is a newly initialized set with $(\boldsymbol{p}_2, \boldsymbol{k}_2)$.

**Opting To Grow or Not To Grow**  Based on the analysis, we propose a dynamic growing approach as follows:

$$\begin{cases} \textit{To Grow} & \text{if} \quad \min_{m \in (1,2)} Z_m > 0 \\ \textit{Not To Grow} & \text{else} \quad \min_{m \in (1,2)} Z_m \leq 0. \end{cases} \tag{13}$$

Table 1: Results of adding LW2G on three baselines: DualPrompt, S-Prompt++, and HidePrompt. Since the official code of Hideprompt has a code implementation issue about prompt retrieval, we asked the authors for the fixed version of code and reproduced the following experimental results. More details about the issue and the fixed version of official code are provided in Appendix E.

| Settings | Methods | FAA ($\uparrow$) | PRA ($\uparrow$) | FFM ($\downarrow$) | SSP ($\downarrow$) |
|---|---|---|---|---|---|
| CIFAR_INC10_TASK10 | DualPrompt | 85.94$\pm$0.19 | 59.44$\pm$0.32 | 6.38$\pm$0.16 | 10 |
| | DualPrompt [+ LW2G] | **86.86$\pm$0.30** | **78.33$\pm$0.16** | **6.03$\pm$0.62** | **2** |
| | S-Prompt++ | 89.25$\pm$0.09 | 99.52$\pm$0.10 | 4.10$\pm$0.05 | 10 |
| | S-Prompt++ [+ LW2G] | **89.32$\pm$0.16** | **100.0$\pm$0.00** | **3.46$\pm$0.19** | **7** |
| | HidePrompt | 85.77$\pm$0.28 | 80.78$\pm$0.61 | 6.19$\pm$0.10 | 10 |
| | HidePrompt [+ LW2G] | **87.60$\pm$0.37** | **95.39$\pm$0.53** | **4.28$\pm$0.03** | **2** |
| IMR_INC20_TASK10 | DualPrompt | 63.63$\pm$0.30 | 41.05$\pm$0.94 | 6.41$\pm$0.14 | 10 |
| | DualPrompt [+ LW2G] | **65.60$\pm$0.52** | **80.40$\pm$1.36** | **5.72$\pm$0.07** | **2** |
| | S-Prompt++ | 63.26$\pm$0.12 | 44.31$\pm$1.03 | 6.22$\pm$0.05 | 10 |
| | S-Prompt++ [+ LW2G] | **65.44$\pm$0.32** | **79.35$\pm$1.44** | **6.01$\pm$1.01** | **5** |
| | HidePrompt | 62.42$\pm$0.12 | 62.07$\pm$0.90 | 8.89$\pm$0.15 | 10 |
| | HidePrompt [+ LW2G] | **63.23$\pm$0.36** | **65.13$\pm$0.59** | **7.19$\pm$0.01** | **6** |
| CUB_INC20_TASK10 | DualPrompt | 82.09$\pm$0.47 | 66.71$\pm$0.23 | 6.40$\pm$0.02 | 10 |
| | DualPrompt [+ LW2G] | **82.43$\pm$0.60** | **70.09$\pm$0.16** | **5.25$\pm$0.03** | **7** |
| | S-Prompt++ | 82.57$\pm$0.41 | 66.30$\pm$1.30 | 4.85$\pm$0.06 | 10 |
| | S-Prompt++ [+ LW2G] | **82.61$\pm$0.13** | **87.49$\pm$1.02** | **4.54$\pm$0.06** | **3** |
| | HidePrompt | 85.59$\pm$0.32 | 88.58$\pm$0.51 | 3.22$\pm$0.01 | 10 |
| | HidePrompt [+ LW2G] | **86.17$\pm$0.62** | **92.53$\pm$0.21** | **3.08$\pm$0.03** | **4** |

- While chosing **To Grow**, we initialize a new set ($\boldsymbol{p}_3, \boldsymbol{k}_3$). Then, update ($\boldsymbol{p}_3, \boldsymbol{k}_3$) with task 3 and build a new feature space $\mathcal{S}_3$ with threshold, $\epsilon_{\text{task}}$, from task 3 only and store $\mathcal{S}_3$ into $\mathcal{M}$.

- While chosing **Not To Grow**, we select an old set ($\boldsymbol{p}_t, \boldsymbol{k}_t$) from $\mathcal{P}$, where $t = \arg\min_{m \in (1,2)} Z_m$. Then, update ($\boldsymbol{p}_t, \boldsymbol{k}_t$) with task 3 under *orthogonal condition* and update the old feature space $\mathcal{S}_t$ with threshold, $\epsilon_{\text{task}}$, with new bases from task 3.

### 4.3 CONSISTENCY WITH PRE-TRAINED KNOWLEDGE
Recent studies in transfer learning and domain adaptation revealed that when employing PEFT for fine-tuning PTM, the performance after fine-tuning often falls short of the pre-trained knowledge of PTM itself. However, this aspect has not been extensively studied in PCL.

Therefore, we exploit two distinct level of forgetting issues faced in PCL: (1) continuous fine-tuning on downstream tasks leading to the forgetting of pre-trained knowledge, and (2) continual learning on new tasks resulting in the forgetting of old tasks.

To tackle the former issue, we adjust the gradient of the new tasks to be orthogonal to the pre-trained feature space. However, due to the domain gap between the incremental task training data and the pre-trained data, a fully orthogonal manner is too stringent and can significantly impact the plasticity. To achieve a balance between maintaining plasticity and fully utilization of the pre-trained knowledge, we propose to apply a soft constraint to the gradient as follows:

$$\boldsymbol{g} = \boldsymbol{g} - (1-\phi)\text{Proj}_{\mathcal{S}_3^{\text{pre}}}(\boldsymbol{g}), \tag{14}$$

where $\phi$ is the coefficient of the soft constraint to control the orthogonality and $\mathcal{S}_3^{\text{pre}}$ is the pre-trained feature space for task 3. When learning on task 3, the gradient can be obtained from Equation 7 while DGA chooses to grow, or from Equation 10 while DGA chooses not to grow. And $\phi$ can flexibly control the real gradient $\boldsymbol{g}$, aligning it as closely as possible with the feature space of the pre-trained knowledge, while ensuring the learning ability on new tasks.

### 4.4 FACILITATION FOR FORWARD TRANSFER
To facilitate forward knowledge transfer during learning task 3, we propose a simple yet effective method: *reusing the frozen weights of prompts* from $\mathcal{P}$. Specifically, before learning task 3, we can characterize the correlation between the new task 3 and the existing feature space in $\mathcal{M}$ with HFC metric. A larger HFC indicates more projection onto the old feature space $\mathcal{S}_2$ than $\mathcal{S}_1$, as illustrated in Figure 1. Therefore, it indicates that task 3 has higher similarity with task 2 than task 1. Consequently, naturally reusing the set of prompts corresponding to task 2 can effectively facilitate the learning of task 3.

$$\boldsymbol{p}_i^* = [\boldsymbol{p}, \text{stg}(\boldsymbol{p}_\mathcal{K})], \tag{15}$$

Table 2: Results on OMNI benchmark with two extreme settings: **30 tasks and 60 tasks**. Additionally, we provide SSP, FLOPS and Training Time (TT) to measure the computational overhead and methods' complexity.

| Settings | Methods | FAA ($\uparrow$) | PRA ($\uparrow$) | FFM ($\downarrow$) | SSP ($\downarrow$) | FLOPS (G) ($\downarrow$) | TT (h) ($\downarrow$) |
|---|---|---|---|---|---|---|---|
| OMNI_INC10_TASK30 | DualPrompt | 63.36 | 68.47 | 12.92 | 30 | 35.19 | 4.5 |
| | DualPrompt [+ LW2G] | 65.12 | 80.95 | **10.75** | 9 | 37.21 | 5.0 |
| | S-Prompt++ | 64.44 | 55.87 | 9.02 | 30 | 35.17 | 4.5 |
| | S-Prompt++ [+ LW2G] | **65.90** | 63.86 | 8.50 | 10 | 37.24 | 5.2 |
| OMNI_INC5_TASK60 | DualPrompt | 61.85 | 69.94 | 13.50 | 60 | 35.19 | 5.0 |
| | DualPrompt [+ LW2G] | 63.17 | 75.31 | 12.01 | 17 | 37.21 | 6.1 |
| | S-Prompt++ | 62.31 | 54.59 | 10.04 | 60 | 35.17 | 5.1 |
| | S-Prompt++ [+ LW2G] | **63.70** | 62.60 | 9.90 | 18 | 37.24 | 6.2 |

Table 3: Ablation study on three components in LW2G. Here we present FAA and PRA for all baselines and variants in LW2G, e.g., "DGA" refers to the use of Dynamic Growing Approach within the baseline methods, DualPrompt and S-Prompt++.

| Variants | FAA ($\uparrow$) | PRA ($\uparrow$) | Variants | FAA ($\uparrow$) | PRA ($\uparrow$) |
|---|---|---|---|---|---|
| DualPrompt (baseline) | 63.63 | 41.05 | S-Prompt++ (baseline) | 63.26 | 44.31 |
| DualPrompt [+ DGA] | 65.02 | 77.68 | S-Prompt++ [+ DGA] | 65.18 | 76.35 |
| DualPrompt [+ CPK] | 64.34 | 50.39 | S-Prompt++ [+ CPK] | 63.90 | 52.67 |
| DualPrompt [+ FFT] | 64.08 | 47.17 | S-Prompt++ [+ FFT] | 63.89 | 50.02 |
| DualPrompt [+ LW2G] | **65.60** | **80.40** | S-Prompt++ [+ LW2G] | **65.44** | **79.35** |

where $\text{stg}(\cdot)$ means *stop gradient* to frozen the $\boldsymbol{p}_{\mathcal{K}}$. Besides, $\boldsymbol{p}$ is a newly initialized set of prompts when DGA chooses *to grow* or an old set of prompts from $\mathcal{P}$ when DGA chooses *not to grow*. And $\boldsymbol{p}_{\mathcal{K}}$ is obtained as follows:

$$\mathcal{K} = \underset{\{u_i\}_{i=1}^N \in \{1,2\}}{\arg\max} \; \text{HFC}(\boldsymbol{g}_{u_i}, \text{Proj}_{\mathcal{S}_{u_i}}(\boldsymbol{g}_{u_i})), \tag{16}$$

where $\mathcal{K}$ represents a subset of sets with top-$N$ from $\mathcal{P}$.

# 5 EXPERIMENT

In this section, we first describe the experimental setups, and then present the experimental results.

## 5.1 EXPERIMENTAL SETUPS

**Benchmarks** We evaluate our method on multiple datasets against state-of-the-art baselines. Specifically, we use the following datasets: CIFAR100 Krizhevsky et al. (2009) (CIFAR), which contains 100 classes with 100 images per class; CUB200 Wah et al. (2011) (CUB), which consists of 11,788 images across 200 birds classes; ImageNet-R Hendrycks et al. (2021) (IMR), which includes 30,000 images from 200 classes that pose challenges for PTMs pre-trained on ImageNet; and Omnibenchmark Zhang et al. (2022) (OMNI), which comprises over 90,000 images from 300 classes. Besides, we denote different experimental settings as 'Dataset_IncN_TaskM', e.g., 'CIFAR_INC10_Task10', which means learning on CIFAR with 10 tasks and each task contains 10 classes.

**Baselines** We use DualPrompt Wang et al. (2022b), S-Prompt++ Wang et al. (2024a) and Hide-Prompt Wang et al. (2024a) as our baselines for Class-CL. Following Wang et al. (2024a), we record the average accuracy of all encountered classes after learning on each task, presenting the last one as the Final Average Accuracy (FAA). We also present the Final Forgetting Measure (FFM) of all tasks and Prompt Retrieval Accuracy (PRA) to measure the accuracy during *prompt retrieval*. Additionally, Selectable Sets of Prompts (SSP) is also provided to demonstrate the amount of sets in $\mathcal{P}$. Please refer to Appendix D.2 for more details.

**Implementations** Our LW2G needs to set the value of four hyperparameters: $\epsilon_{\text{task}}$, $\epsilon_{\text{pre}}$, $\phi$, and $N$. Details on different benchmarks are provided in Appendix D.1. We use ViT pretrained on ImageNet-21K for all experiments. All results are the average under three different random seeds. Furthermore, as the pre-trained feature space is built from PTM, we further validate the effectiveness of LW2G under other PTMs. Results are provided in Appendix F.6.

## 5.2 Main Results

**Typical Settings**  Table 1 presents the results of applying different state-of-the-art PCL methods and incorporating LW2G. We report four metrics FAA, PRA, FFM and SSP, where FAA and FFM are the typical metrics in CL to evaluate the performance. Additionally, PRA and SSP are unique for PCL. LW2G outperforms existing PCL by a large margin in each setting. For IMR, LW2G is better than DualPrompt, S-Prompt++ and Hideprompt by 1.97%, 2.17% and 0.81%, respectively on FAA. For CIFAR, it appears that LW2G brings a significant decent in anti-forgetting, especially comparing with S-Prompt++ and Hideprompt on FFM. As for the PCL unique metrics PRA and SSP, LW2G leads to notable improvements in PRA for all three baselines, with the largest improvement reaching up to 39.35%. Additionally, it also results in a substantial reduction in SSP. For example, DualPrompt combined with LW2G on CIFAR only requires 2 sets of prompts compared to the original DualPrompt, which utilizes 10 sets. The same reduction in parameters can be observed across multiple settings.

**Long Task Settings**  Learning in the context of long sequential tasks has long been regarded as a more challenging setting in CL. We showcase the performance of DualPrompt and S-Prompt++ on two extreme settings: OMNI_INC10_TASK30 and OMNI_INC5_TASK60 in Table 2. Existing baselines employ a pool with the size equivalent to the length of tasks, resulting in poor performance on PRA. However, incorporating the LW2G significantly enhances PRA, leading to noticeable improvements in both FAA and FFM. Moreover, we observe that LW2G requires to maitain a memory $\mathcal{M}$ for gradient modification, unavoidably introducing additional computational overhead and lengthening training time. Nevertheless, the results indicate that the extra cost compared to baselines is relatively modest. Additionally, we find that the adoption of LW2G results in a substantial decrease in the total amount of selectable sets, approximately by 70%.

## 5.3 Ablation Study

We conduct an extensive ablation study presented in Table 3 to validate the effectiveness of the three components in LW2G. Initially, we construct DualPrompt and S-Prompt++ as baselines and progressively incorporate the DGA, CPK, and FFT. Overall, optimizing each component yields clear benefits, with all contributing to the robust gains of LW2G. Interestingly, while CPK and FFT exhibits less pronounced improvements compared to the baseline, the enhancement from DGA is more significant. Besides, the combination of all three components provides the optimal performance, suggesting highly synergistic and complementary effects rather than operating in isolation. Moreover, it is noteworthy that CPK and FFT do not reduce SSP, hence the performance improvement solely stemmed from the enhanced representational capacity of prompts. DGA not only integrates knowledge from multiple tasks into a single set of prompts, thereby enhancing the representational capacity, but importantly, the notable improvement in PRA is attributed to the reduction in the total amount of available sets during *prompt retrieval*, thereby aiding PCL performance.

## 5.4 Detail Analysis

**Effectiveness of DGA**  While chosing *not to grow*, DGA utilized in LW2G selects the set $(\boldsymbol{p}_*, \boldsymbol{k}_*)$ with the Min-$Z$ from $\mathcal{P}$ when learning task $i$, and learns new knowledge based on this set, adjusting gradient to prevent forgetting of the old knowledge contained in $(\boldsymbol{p}_*, \boldsymbol{k}_*)$. After learning, $(\boldsymbol{p}, \boldsymbol{k})$ encompasses both the new knowledge from task $i$ and the existing old knowledge. Here, we explore the impact of different implementations of DGA on FAA. In Table 4, No-DGA represents base-

Table 4: Different implementations on DGA. Here we present FAA for all variants.

| DGA Variants | CIFAR | | IMR | |
| --- | --- | --- | --- | --- |
| | DualPrompt | S-Prompt++ | DualPrompt | S-Prompt++ |
| No-DGA (Baseline) | 85.94 | 89.25 | 63.63 | 63.26 |
| DGA-Rand | 85.99 | 88.32 | 64.82 | 64.76 |
| DGA-AG | 84.78 | 85.17 | 63.73 | 63.43 |
| DGA-Max HFC | 86.08 | 86.73 | 64.31 | 63.91 |
| DGA-Min HFC | **86.86** | **89.32** | **65.60** | **65.44** |

line methods, e.g., S-Prompt++ and DualPrompt. DGA-Rand represents randomly selecting an old set of prompts from $\mathcal{P}$. DGA-AG represents that $\mathcal{P}$ consists of only a single set, implying continuous learning of new knowledge on this set of parameters. DGA-Max HFC indicates selecting the set from $\mathcal{P}$ with the maximum HFC value. The results clearly demonstrate the superiority of DGA-Min HFC employed in LW2G over other variants, aligning with the conclusion in Theorem 1.

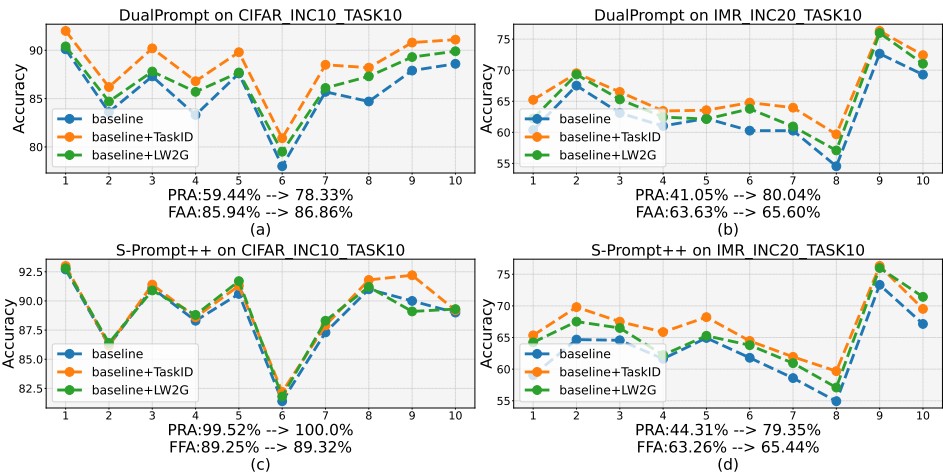

Figure 3: The x-axis denotes the enhancement in PRA with LW2G compared to the baseline. Apart from baseline and LW2G, we also present the results of Task-CL. Task-CL ensures the real upper bound of PCL by providing a correct prompt set for each testing sample through a given task ID.

Table 5: Variation process of DualPrompt [+ LW2G] on IMR.

| Task | Calculation Process | Minimal Z | Option | Prompt sets pool |
|---|---|---|---|---|
| 1 | / | / | To Grow a new $(\boldsymbol{p}_1, \boldsymbol{k}_1)$ | $(\boldsymbol{p}_1, \boldsymbol{k}_1) \to$ Task 1 |
| 2 | $HFC_1$=13.90, $HFC_1^{pre}$=40.23 | $Z_1$=-26.33<0 | Not To Grow with $(\boldsymbol{p}_1, \boldsymbol{k}_1)$ | $(\boldsymbol{p}_1, \boldsymbol{k}_1) \to$ Task 1,2 |
| 3 | $HFC_1$=20.22, $HFC_1^{pre}$=40.80 | $Z_1$=-20.58<0 | Not To Grow with $(\boldsymbol{p}_1, \boldsymbol{k}_1)$ | $(\boldsymbol{p}_1, \boldsymbol{k}_1) \to$ Task 1,2,3 |
| 4 | $HFC_1$=25.09, $HFC_1^{pre}$=41.50 | $Z_1$=-16.41<0 | Not To Grow with $(\boldsymbol{p}_1, \boldsymbol{k}_1)$ | $(\boldsymbol{p}_1, \boldsymbol{k}_1) \to$ Task 1,2,3,4 |
| 5 | $HFC_1$=29.15, $HFC_1^{pre}$=42.92 | $Z_1$=-13.77<0 | Not To Grow with $(\boldsymbol{p}_1, \boldsymbol{k}_1)$ | $(\boldsymbol{p}_1, \boldsymbol{k}_1) \to$ Task 1,2,3,4,5 |
| 6 | $HFC_1$=32.85, $HFC_1^{pre}$=42.78 | $Z_1$=-9.33<0 | Not To Grow with $(\boldsymbol{p}_1, \boldsymbol{k}_1)$ | $(\boldsymbol{p}_1, \boldsymbol{k}_1) \to$ Task 1,2,3,4,5,6 |
| 7 | $HFC_1$=36.35, $HFC_1^{pre}$=41.85 | $Z_1$=-5.5<0 | Not To Grow with $(\boldsymbol{p}_1, \boldsymbol{k}_1)$ | $(\boldsymbol{p}_1, \boldsymbol{k}_1) \to$ Task 1,2,3,4,5,6,7 |
| 8 | $HFC_1$=39.39, $HFC_1^{pre}$=42.42 | $Z_1$=-3.03<0 | Not To Grow with $(\boldsymbol{p}_1, \boldsymbol{k}_1)$ | $(\boldsymbol{p}_1, \boldsymbol{k}_1) \to$ Task 1,2,3,4,5,6,7,8 |
| 9 | $HFC_1$=42.54, $HFC_1^{pre}$=41.37 | $Z_1$=1.17>0 | To Grow a new $(\boldsymbol{p}_2, \boldsymbol{k}_2)$ | $(\boldsymbol{p}_1, \boldsymbol{k}_1) \to$ Task 1,2,3,4,5,6,7,8 $(\boldsymbol{p}_2, \boldsymbol{k}_2) \to$ Task 9 |
| 10 | $HFC_1$=42.54, $HFC_1^{pre}$=40.92 $HFC_2$=13.81, $HFC_2^{pre}$=41.81 | $Z_2$=-28.00<0 | Not To Grow with $(\boldsymbol{p}_2, \boldsymbol{k}_2)$ | $(\boldsymbol{p}_1, \boldsymbol{k}_1) \to$ Task 1,2,3,4,5,6,7,8 $(\boldsymbol{p}_2, \boldsymbol{k}_2) \to$ Task 9,10 |

**Gains on Each Task**  Figure 3 presents detailed accuracy on each task. Here, we provide a comparison between DualPrompt and S-Prompt++ on two benchmarks. The x-axis of each plot represents the change from *baseline* to *baseline+LW2G* in terms of PRA. Apart from (*c*), the addition of LW2G all leads to consistent improvements in accuracy on each task, as the PRA of the baseline method in (*c*) has already reached 99.52%. In the other three settings, PRA experiences significant increasment, thereby enhancing classification accuracy. Additionally, we also provide results for *baseline+taskID*, i.e., PCL on Task-CL. In this setting, during inference, taskid is provided to select the correct set for each testing sample, which is considered as the upper bound of PCL. It further demonstrates that our proposed LW2G can effectively reduce the optionality during *prompt retrieval* while ensuring the integration of old and new knowledge, thereby improving performance.

**Visualization of the Dynamic Growing Process**  In the proposed LW2G method, the DGA module determines whether to grow a new set of prompts or reuse an existing set from the prompt sets pool based on the HFC metric, which can measure the hindrance on learning new tasks while maintaining old knowledge under orthogonal condition. We provide a detailed dynamic process in the following Table 5. Before learning each task (except task 1), LW2G first calculates the HFC value and subsequently decides whether to perform dynamic expansion based on the minimum Z value using Equation 12 and 13. Further results can be found in Appendix F.5.

## 6  CONCLUSION

In this paper, we propose a plug-in module within existing Prompt-based Continual Learning (PCL), called Learning Whether To Grow (LW2G). Specifically, LW2G enables PCL to dynamically learn to whether to add a new set of prompts for each task (*to grow*) or to utilize an existing set of prompts (*not to grow*) based on the relationships between tasks. Inspired by Gradient Projection-based Continual Learning (GPCL), we utilize the *orthogonal condition* to form an effective and efficient prompt sets pool. Besides, we also provide a theoretical analysis on hindrance under the *orthogonal condition* in GPCL. Extensive experiments show the effectiveness of our method.

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

# A  ALGORITHM

---

**Algorithm 1** LW2G: Learning Whether to Grow.

---

**Input**: Task length $T$, Datasets for each task: $\{\mathcal{D}^1, \mathcal{D}^2, \cdots, \}$, Pool $\mathcal{P} = \{\}$, Memory $\mathcal{M} = \{\}$, Training Epochs $E$.
**Output**: Updated Pool $\mathcal{P}$ and $\mathcal{M}$.

1: **for** $i = 1, 2, \cdots, T$ **do**
2:    **if** $i = 1$ **then**                                                      ▷ DGA learns to grow or not to grow
3:       **DGA** chose to grow;
4:       Initialization $(p_i, k_i)$ and Store in $\mathcal{P}$;
5:    **else**
6:       Get a subset from $\mathcal{D}_{\text{sub}}^i$.
7:       Get all selectable sets in $\mathcal{P}$, denoted as L;
8:       **for** j in $L$ **do**
9:          Get the old set from $\mathcal{P}$, $(p_j, k_j)$;
10:         Get the old feature space from $\mathcal{M}$, $\mathcal{S}_j$;
11:         Get $\boldsymbol{g}$ on $(p_j, k_j)$ with $\mathcal{D}_{\text{sub}}^i$;
12:         Get HFC$_j$ via Equation 8 and HFC$_{\text{pre}}$ via Equation 11 and $Z_j$ via Equation 12;
13:       **DGA** chose to grow or not to grow via Equation 13;
14:       **if DGA** chose to grow **then**
15:         Initialization $(p_i, k_i)$ and Store in $\mathcal{P}$;
16:       **else**
17:         Selection $(p_t, k_t)$, where $t = \arg\max_{j \in L} Z_j$;
18:         Change $(p_t, k_t)$ to $(p_i, k_i)$;
19:         Change $\mathcal{S}_t$ to $\mathcal{S}_i$;
20:    **for** $e = 1, 2, \cdots, E$ **do**                                    ▷ **Start Training**
21:       Get sets of most similar tasks via Equation 16;           ▷ FFT to forward facilitate
22:       Get $\boldsymbol{g}$ on $(p_i, k_i)$ with $\mathcal{D}^i$;
23:       Apply soft constraints on $\boldsymbol{g}$ via Equation 14;        ▷ CPK to apply soft constraints
24:       Update $(p_i, k_i)$;
25:    Build or update space $\mathcal{S}_i$ in $\mathcal{M}$ via Appendix B.2;     ▷ DGA dynamically build or update space
     **return** $\mathcal{P}, \mathcal{M}$;

---

# B  THEORETICAL FOUNDATION

## B.1  PROOF OF THEOREM 1

Given a space $\mathcal{S}_1 = \text{span}\{\boldsymbol{B}_1\}$, where $\boldsymbol{B}_1 = [\boldsymbol{b}_1, \ldots, \boldsymbol{b}_{k_1}] \in \mathbb{R}^{n \times k_1}$ is a set of $k_1$ bases for $\mathcal{S}_1$, and a space $\mathcal{S}_2 = \text{span}\{\boldsymbol{B}_2\}$, where $\boldsymbol{B}_2 = [\boldsymbol{b}_1, \ldots, \boldsymbol{b}_{k_1}, \boldsymbol{b}_{k_1+1}, \ldots, \boldsymbol{b}_{k_1+k_2}] \in \mathbb{R}^{n \times (k_1+k_2)}$ is a set of $k_1 + k_2$ bases for $\mathcal{S}_2$. $\forall \boldsymbol{\alpha} \in \mathbb{R}^{n \times 1}$, denoted $\boldsymbol{\alpha}$ on space $\mathcal{S}_i$ is $\text{Proj}_{\mathcal{S}_i}(\boldsymbol{\alpha})$. Following Definition 1, the ange between $\boldsymbol{\alpha}$ and $\text{Proj}_{\mathcal{S}_i}(\boldsymbol{\alpha})$ is denoted as $\text{HFC}(\boldsymbol{\alpha}, \text{Proj}_{\mathcal{S}_i}(\boldsymbol{\alpha}))$. Then there always exists:

$$\text{HFC}(\boldsymbol{\alpha}, \text{Proj}_{\mathcal{S}_1}(\boldsymbol{\alpha})) \geq \text{HFC}(\boldsymbol{\alpha}, \text{Proj}_{\mathcal{S}_2}(\boldsymbol{\alpha})). \tag{17}$$

*Proof.* $\forall \boldsymbol{\alpha} \in \mathbb{R}^{n \times 1}$, $\boldsymbol{\alpha} = [\alpha_1, \ldots, \alpha_n]^T$. Without loss of generality, $\{\boldsymbol{b}_i, i = 1, \ldots, k_1 + k_2\}$ is a set of *standard orthonormal basis*. As we defined, $\text{Proj}_{\mathcal{S}_1}(\boldsymbol{\alpha}) = [g_1, \ldots, g_{k_1}] \in \mathbb{R}^{k_1 \times 1}$ and $\text{Proj}_{\mathcal{S}_2}(\boldsymbol{\alpha}) = [g_1, \ldots, g_{k_1}, g_{k_1+1}, \ldots, g_{k_1+k_2}] \in \mathbb{R}^{(k_1+k_2) \times 1}$, where $g_i = \langle \boldsymbol{\alpha}, \boldsymbol{b_i} \rangle$.

Then, we have

$$\begin{aligned} cos(\boldsymbol{\alpha}, \text{Proj}_{\mathcal{S}_1}(\boldsymbol{\alpha})) &= \frac{\boldsymbol{\alpha} \cdot \text{Proj}_{\boldsymbol{S}_1}(\boldsymbol{\alpha})}{\|\boldsymbol{\alpha}\| \|\text{Proj}_{\boldsymbol{S}_1}(\boldsymbol{\alpha})\|} \\ &= \frac{\sum_{i=1}^{k_1} (g_i)^2}{\sqrt{\sum_{i=1}^{k_1} (g_i)^2} \sqrt{\sum_{i=1}^{n} (g_i)^2}} \end{aligned} \tag{18}$$

Likewise, we have

$$
\begin{aligned}
cos(\boldsymbol{\alpha}, \mathrm{Proj}_{\boldsymbol{S}_2}(\boldsymbol{\alpha})) &= \frac{\boldsymbol{\alpha} \cdot \mathrm{Proj}_{\boldsymbol{S}_2}(\boldsymbol{\alpha})}{\|\boldsymbol{\alpha}\|\|\mathrm{Proj}_{\boldsymbol{S}_2}(\boldsymbol{\alpha})\|} \\
&= \frac{\sum_{i=1}^{k_1+k_2}(g_i)^2}{\sqrt{\sum_{i=1}^{k_1+k_2}(g_i)^2}\sqrt{\sum_{i=1}^{n}(g_i)^2}}
\end{aligned}
\tag{19}
$$

In addition,

$$
\begin{aligned}
\frac{cos(\boldsymbol{\alpha}, \mathrm{Proj}_{\boldsymbol{S}_2}(\boldsymbol{\alpha}))}{cos(\boldsymbol{\alpha}, \mathrm{Proj}_{\boldsymbol{S}_1}(\boldsymbol{\alpha}))} &= \frac{\sum_{i=1}^{k_1+k_2}(g_i)^2}{\sum_{i=1}^{k_1}(g_i)^2}\frac{\sqrt{\sum_{i=1}^{k_1}(g_i)^2}}{\sqrt{\sum_{i=1}^{k_1+k_2}(g_i)^2}} \tag{20} \\
&= \frac{1+C}{\sqrt{(1+C)}} \tag{21} \\
&= \sqrt{(1+C)} \geq 1. \tag{22}
\end{aligned}
$$

Where $C = \frac{\sum_{i=k_1+1}^{k_1+k_2}(g_i)^2}{\sum_{i=1}^{k_1}(g_i)^2} \geq 0$. Thus, $cos(\boldsymbol{\alpha}, \mathrm{Proj}_{\boldsymbol{S}_2}(\boldsymbol{\alpha})) \geq cos(\boldsymbol{\alpha}, \mathrm{Proj}_{\boldsymbol{S}_1}(\boldsymbol{\alpha}))$. Thus, $\mathrm{HFC}(\boldsymbol{\alpha}, \mathrm{Proj}_{\boldsymbol{S}_1}(\boldsymbol{\alpha})) \geq \mathrm{HFC}(\boldsymbol{\alpha}, \mathrm{Proj}_{\boldsymbol{S}_2}(\boldsymbol{\alpha}))$.

This finishes the proof.

## B.2 BUILDING AND UPDATING OF FEATURE SPACE

In GPCL, a feature space spanned by the old tasks is required during gradient modification, involving two stages: (1) Building of the new feature space, and (2) Updating of old faeture space. We first introduce the technique used in matrix factorization, Singular Value Decomposition (SVD). Then, details on building or updating of the feature space are also provided.

**Singular Value Decomposition (SVD)**   SVD is a general geometrical tool used in matrix factorization to factorize a given matrix $\boldsymbol{A} \in \mathbb{R}^{m \times n}$ into the product of three matrices as follows Deisenroth et al. (2020):

$$
\boldsymbol{A} = \boldsymbol{U}\boldsymbol{\Sigma}(\boldsymbol{V})^T,
\tag{23}
$$

where $\boldsymbol{U} \in \mathbb{R}^{m \times m}$ and $\boldsymbol{V} \in \mathbb{R}^{n \times n}$ are orthogonal. $\boldsymbol{\Sigma} \in \mathbb{R}^{m \times n}$ contains the sorted singular values along its main diagonal. Specifically, the diagonal value $\sigma_i = \boldsymbol{\Sigma}_{ii}$ are the *singular values* of $\boldsymbol{A}$ and the number of non-zero $\sigma_i$ is equal to $r = \mathrm{rank}(\boldsymbol{A})$. Besides, the columns of $\boldsymbol{U}$ and the rows of $(\boldsymbol{V})^T$ are two sets of **orthogonal bases** $\{\boldsymbol{u}_1, \boldsymbol{u}_2, \ldots, \boldsymbol{u}_m\}$ and $\{\boldsymbol{v}_1, \boldsymbol{v}_2, \ldots, \boldsymbol{v}_n\}$, respectively. As the singular values are sorted in $\boldsymbol{\Sigma}$ along its diagonal, the SVD of $\boldsymbol{A}$ can be also denoted as follows:

$$
\boldsymbol{A} = \sum_{i=1}^{r} \sigma_i \boldsymbol{u}_i \boldsymbol{v}_i'.
\tag{24}
$$

Therefore, the $k$-rank approximation $(\boldsymbol{A})_k$ of $\boldsymbol{A}$ can be denoted as follows:

$$
||(\boldsymbol{A})_k||_F^2 \geq \epsilon||\boldsymbol{A}||_F^2,
\tag{25}
$$

where $\epsilon$ is a given error tolerance and $||\cdot||_F^2$ is the Frobenius norm.

**Building of the New Feature Space**   After training on task 1, for each layer we construct a representation matrix $\boldsymbol{R}_1^l = \left[\boldsymbol{x}_{1,1}^l, \ldots, \boldsymbol{x}_{1,n_1}^l\right] \in \mathbb{R}^{n \times d}$ by concatenating representations of $n$ samples along the columns obtained from sending $n$ samples only from task 1 into the current DNN, $\mathcal{W}_1$. Next, we perform SVD on $\boldsymbol{R}_1^l = \boldsymbol{U}_1^l \boldsymbol{\Sigma}_1^l (\boldsymbol{V}_1^l)^T$ followed by its $k$-rank approximation $(\boldsymbol{R}_1^l)_k$ according to the following criteria for the given threshold, $\epsilon_{\mathrm{task}}$:

$$
||(\boldsymbol{R}_1^l)_k||_F^2 \geq \epsilon_{\mathrm{task}}||\boldsymbol{R}_1^l||_F^2.
\tag{26}
$$

Therefore, the feature space for layer $l$ is built by $\mathcal{S}_1^l = \mathrm{span}\left\{\boldsymbol{B}_1^l\right\}$, where $\boldsymbol{B}_1^l = \{\boldsymbol{u}_1^l, \ldots, \boldsymbol{u}_k^l\}$ and $\boldsymbol{u}_i^l$ is the first $k$ vectors in $\boldsymbol{U}_1^l$. And $\mathcal{S}_1^l$ is stored in memory $\mathcal{M} = \left\{\mathcal{S}_1^l\right\}$.

**Updating of the Old Feature Space** After learning task $i$, where $i \geq 2$, $\mathcal{S}_{i-1}^l$ in $\mathcal{M}$ needs to be updated to $\mathcal{S}_i^l$ with new task-specific bases from task $i$. To obtain such bases, for each layer $l$, we utilize the current DNN, $\mathcal{W}_i$, to construct a representation matrix $\boldsymbol{R}_i^l = \left[\boldsymbol{x}_{1,1}, \ldots, \boldsymbol{x}_{1,n}\right] \in \mathbb{R}^{n \times d}$ from task $i$ only. Before performing SVD and subsequent $k$-rank approximation, we first eliminate the common bases that already present in $\mathcal{S}_{i-1}^l$ so that newly added bases are unique and orthogonal to the existing bases in $\mathcal{S}_{i-1}^l$. To accomplish this, we proceed as follows:

$$\hat{\boldsymbol{R}}_i^l = \boldsymbol{R}_i^l - \boldsymbol{B}_{i-1}^l \left(\boldsymbol{B}_{i-1}^l\right)^T \left(\boldsymbol{R}_i^l\right) = \boldsymbol{R}_i^l - \boldsymbol{R}_{i,\text{proj}}^l. \tag{27}$$

Afterwards, SVD is performed on $\hat{\boldsymbol{R}}_i^l = \hat{\boldsymbol{U}}_i^l \hat{\boldsymbol{\Sigma}}_i^l (\hat{\boldsymbol{V}}_i^l)^T$, thus obtaining $h$ new orthogonal bases for minimun value of $h$ statisfying the following criteria for the given threshold, $\epsilon_{\text{task}}$:

$$||\boldsymbol{R}_{i,\text{proj}}^l||_F^2 + ||\hat{\boldsymbol{R}}_i^l||_F^2 \geq \epsilon_{\text{task}} ||\boldsymbol{R}_i^l||_F^2. \tag{28}$$

$\boldsymbol{B}_{i-1}^l$ is then updated to $\boldsymbol{B}_i^l = \left[\boldsymbol{B}_{i-1}^l, \boldsymbol{u}_1^l, \ldots, \boldsymbol{u}_h^l\right]$ with $h$ new bases. And $\mathcal{S}_{i-1}^l$ is updated to $\mathcal{S}_i^l = \text{span}\left\{\boldsymbol{B}_i^l\right\}$.

## C  REVIEW OF EXISTING PCL

In this section, we review existing PCL with its pipeline. As illustrated in Figure 4, existing PCL such as HidePrompt Wang et al. (2024a), S-Prompt++ Wang et al. (2024a), DualPrompt Wang et al. (2022b), L2P Wang et al. (2022c), S-liPrompt, and S-iPrompt Wang et al. (2022a) generally involves two stages: (1) *prompt learning*, and (2) *prompt retrieval*.

**Prompt Learning** Given a pre-trained model, such as a Vision Transformer (denoted as ViT), an image after *patch embedding* is denoted as $\boldsymbol{x}_e \in \mathbb{R}^{\mathcal{L}_e \times d}$, where $\mathcal{L}_e$ is the length of the patch tokens and $d$ denotes the length of the channels. Before learning task $i$, PCL follows Houlsby et al. (2019); Jia et al. (2022) by utilizing a task-wised set of prompts $\boldsymbol{p}_i \in \mathbb{R}^{\mathcal{L}_p \times \mathcal{L}_b \times d}$, where $\mathcal{L}_p$ is the length of layer-wised prompts and $\mathcal{L}_b$ represents the depth of the blocks into which the prompts is inserted. The new knowledge in task $i$ can be encoded into these newly initialized $\boldsymbol{p}_i$ as follows:

$$\left[\text{cls\_token}^l, \boldsymbol{x}_e^l, \boldsymbol{p}^l\right] = \text{block}^l\left(\left[\text{cls\_token}^{l-1}, \boldsymbol{x}_e^{l-1}, \boldsymbol{p}_i^{l-1}\right]\right) \qquad l = 1, 2, \ldots, N \tag{29}$$

$$\boldsymbol{y} = \text{Head}^i(\text{cls\_token}^N). \tag{30}$$

Here, $\boldsymbol{p}_i^{l-1} \in \mathbb{R}^{\mathcal{L}_p \times d}$ represents the prompts for block $l$. $\boldsymbol{x}_e^{l-1}$ is the original input of block $l$. Additionally, $\text{Head}^i$ represents the classifier head corresponding to task $i$. Since PCL typically considers Class-CL scenarios, a unified classifier head is adopted. This means that while learning task $i$, the weights of the unified classifier head from tasks 1 to $i-1$ are frozen. Then, $\boldsymbol{p}_i$ is optimized using the *cross entropy* loss. Meanwhile, PCL sent $\boldsymbol{x}_e \in \mathbb{R}^{\mathcal{L}_e \times d}$ into the ViT without any prompts as follows:

$$\left[\text{cls\_token}^l, \boldsymbol{x}_e^l\right] = \text{block}^i\left(\left[\text{cls\_token}^{l-1}, \boldsymbol{x}_e^{l-1}\right]\right) \qquad l = 1, 2, \ldots, N. \tag{31}$$

Here, we use $\boldsymbol{q} = \text{cls\_token}^N$ from the output of the last block as the valinia feature of the input sample. Then, $\boldsymbol{k}_i$ is optimized by minimizing the distance between $\boldsymbol{q}$ and $\boldsymbol{k}_i$. There are various methods to measure this distance, such as using cosine similarity as in S-Prompt++ Wang et al. (2024a), DualPrompt Wang et al. (2022b), and L2P Wang et al. (2022c); using KNN in S-liPrompt and S-iPrompt Wang et al. (2022a); or, in the case of HidePrompt Wang et al. (2024a), forgoing $\boldsymbol{k}_i$ and instead utilizing an auxiliary classifier head. Overall, the goal is to design a metric that brings $\boldsymbol{k}_i$ closer to $\boldsymbol{q}$, so that during *prompt retrieval*, the correct $\boldsymbol{p}_i$ can be selected for each testing sample.

After learning task $i$, PCL stores $(\boldsymbol{p}_i, \boldsymbol{k}_i)$ as a pair into the pool $\mathcal{P} = \{(\boldsymbol{p}_i, \boldsymbol{k}_i), i = 1, 2, \ldots\}$.

**Prompt Retrieval** In Class-CL, we do not have access to the task ID. Therefore, given a testing sample, PCL needs to predict which task it belongs to and select the corresponding set from the pool $\mathcal{P}$. Briefly, they first obtain the vanilla feature by sending the testing sample into the ViT without prompts. Then, they use the vanilla feature as a query vector to match $\{\boldsymbol{k}_i, i = 1, 2, \ldots\}$ in the pool

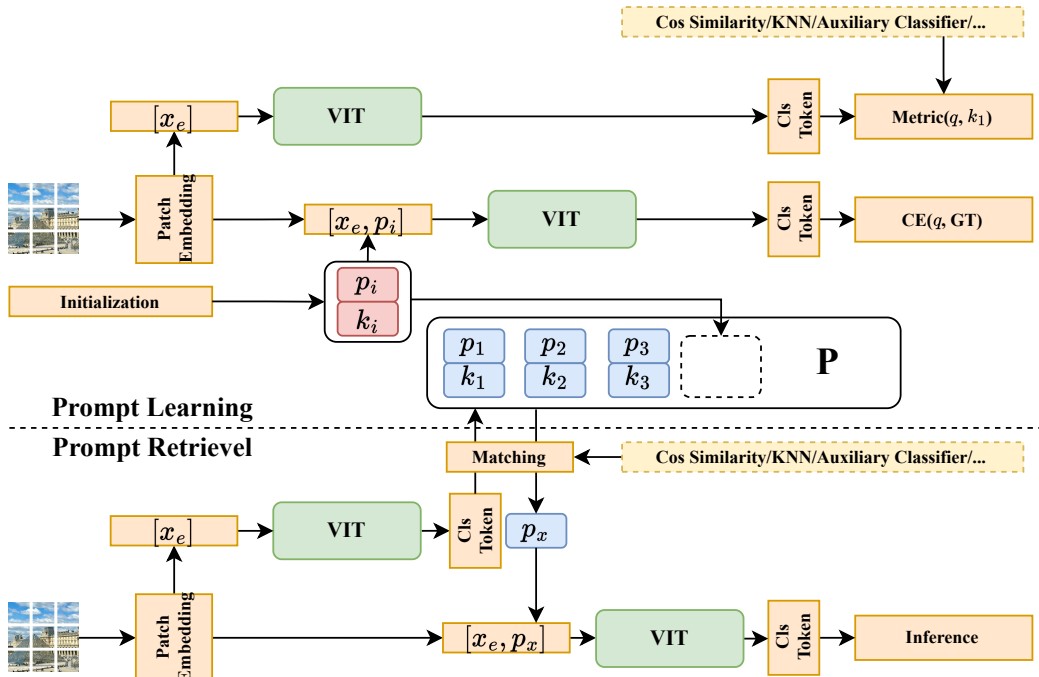

Figure 4: Pipline of existing PCL. Here, we separate it into two stages: *prompt learning* and *prompt retrieval*. In $\mathcal{P}$, blue represents frozen and unlearnable set of prompts, whereas red represents learnable prompt sets.

$\mathcal{P}$ through the metric used in *prompt learning*. After selecting the $\boldsymbol{k}_x$, the $\boldsymbol{p}_x$ is combined with $\boldsymbol{x}_e$ for further inference.

Therefore, predicting the *ground truth* set of prompts for each testing sample is a crucial step for PCL, enabling it to achieve appealing performance.

## D    IMPLEMENTATION DETAILS

In this section, we provide the implementation details of all experiments.

### D.1    TRAINING REGIME AND HYPERPARAMETERS

Following the implementations of previous work Wang et al. (2024a), we train DualPrompt on CIFAR, IMR and CUB with 40, 50, and 50 epochs, respectively; Hideprompt on CIFAR, IMR and CUB with 50, 150, and 50 epochs, respectively; S-Prompt++ on CIFAR, IMR and CUB with 40, 120, and 40 epochs, respectively. The length of prompts $\mathcal{L}_e$ is 20 for all settings. Depth of prompts are as follows: In DualPrompt: g-prompts are inserted in the block $0-1$ and e-prompts are inserted in the block $2-4$. In HidePrompt and S-Prompt++ prompts are inserted in the block $0-4$. **All the experimental results in this paper are averaged over five trials with five different random seeds.** We use 1 4090 GPU for experiments in typical setting and 1 A800 GPU for experiments in long task settings.

For LW2G, the detailed settings for $\epsilon_{\text{task}}$, $\epsilon_{\text{pre}}$, $\phi$, and $N$ are illustrated in Table 6.

### D.2    EVALUATION METRICS

We utilize four evaluation metrics for PCL, including the Final Average Accuracy (FAA), Final Forgetting Measure (FFM), Prompt Retrieval Accuracy (PRA) and Selectable Sets of Prompts (SSP).

Table 6: Hyperparameters of $\epsilon_{\text{task}}$, $\epsilon_{\text{pre}}$, $\phi$, and $N$ in typical settings.

| Settings | Methods | $\epsilon_{\text{task}}$ | $\epsilon_{\text{pre}}$ | $\phi$ | $N$ |
|---|---|---|---|---|---|
| | DualPrompt | 0.95 | 0.95 | 0.5 | 1 |
| CIFAR_INC10_TASK10 | S-Prompt++ | 0.95 | 0.95 | 1.0 | 1 |
| | HidePrompt | 0.99 | 0.99 | 0.5 | 1 |
| | DualPrompt | 0.99 | 0.99 | 0.6 | 1 |
| IMR_INC20_TASK10 | S-Prompt++ | 0.99 | 0.99 | 0.4 | 1 |
| | HidePrompt | 0.90 | 0.90 | 0.2 | 1 |
| | DualPrompt | 0.90 | 0.90 | 0.3 | 1 |
| CUB_INC20_TASK10 | S-Prompt++ | 0.99 | 0.99 | 0.9 | 1 |
| | HidePrompt | 0.95 | 0.95 | 0.7 | 1 |

FAA and FFM are common evaluation metrics in Continual Learning and are formally defined as follows:

$$\text{FAA} = \frac{1}{T} \sum_{i=1}^{T} A_{i,T}, \tag{32}$$

$$\text{FFM} = \frac{1}{T-1} \sum_{i=1}^{T-1} \max_{t \in \{1,\dots,T-1\}} (A_{i,t} - A_{i,T}), \tag{33}$$

where $T$ is the length of the sequential tasks, $A_{i,T}$ is the classification accuracy on the task $i$ after learning the last task $T$.

As analyzed in Appendix C, predicting the *ground truth* set of prompts for each testing sample is a crucial step in PCL. Therefore, we adopt a unique evaluation metric, Prompt Retrieval Accuracy (PRA), for PCL, which is formally defined as follows:

$$\text{PRA} = \frac{1}{T} \sum_{i=1}^{T} R_{i,T}, \tag{34}$$

where $R_{i,T}$ is the accuracy of predicting the set of prompts for each testing sample on task $i$ after learning the last task $T$. Besides, we also use Selectable Sets of Prompt (SSP) to represent the total amount of selectable sets of prompts in the pool $\mathcal{P}$. SSP is not only positively correlated with the number of learnable parameters, but it also effectively reflects how the LW2G proposed in this paper can significantly reduce the selectable amount in baseline methods, thereby benefiting PRA.

# E    REPRODUCTION OF BASELINES

In this section, we first analyze the specific locations and sources of the implementation issues in the official code (Appendix E.1). Subsequently, we further analyze the impact of these implementation issues on model performance and the resulting task ID information leakage problem (Appendix E.2). Finally, after fixing this implementation issue, we observed a significant decline in the performance of the baseline method, which led us to perform a grid search on the hyperparameters in HidePrompt (Appendix E.3).

## E.1    AN IMPLEMENTATION ISSUE ABOUT PROMPT RETRIEVAL

For the compared methods, DualPrompt, S-Prompt++ and HidePrompt, we use the official code[1] from HidePrompt Wang et al. (2024a). However, after inspecting the code line by line, we identified an implementation issue that leads to significant discrepancies between the specific implementation and the method itself. Specifically, the issue occurs during *prompt retrieval* at `https://github.com/thu-ml/HiDe-Prompt/blob/fcb6c7a29ce97e07426fa20f3817c975da3c3b3e/peft/prompt/hide_prompt.py#L109-L111`, which is provided as following Listing 1.

---

[1] https://github.com/thu-ml/HiDe-Prompt

Listing 1: prompt retrieval before fixing the typo.

```
num_layers, dual, batch_size, top_k, length, num_heads,     1
    heads_embed_dim = batched_prompt_raw.shape
batched_prompt = batched_prompt_raw.reshape(                2
    num_layers, batch_size, dual, top_k * length, num_heads, 3
        heads_embed_dim
)                                                           4
```

As analyzed in Appendix C, in the *prompt retrieval* stage, PCL methods (DualPrompt, S-Prompt++, and HidePrompt) need to predict the *ground truth* set of prompts for each testing sample. The tensor 'batched_prompt_raw' in Listing 1 is the prompt sets predicted for each sample during *prompt retrieval*. Since DualPrompt, S-Prompt++, and HidePrompt all utilize **pre-fix tuning methods**, they can be divided into three steps:

1. obtaining representations from input samples via patch embedding,

2. multiplying the representations with the Q, K, and V matrices in the attention mechanism to get the Q, K, and V values, respectively,

3. dividing the selected prompt into two parts, prompt_k and prompt_v, and prepending them to the K and V values, respectively. Here, prompt_k corresponds to key 1 in Figure 5, and prompt_v corresponds to value 1.

Therefore, the purpose of Listing 1 is to swap the dimensions 'dim=1' and 'dim=2' of the tensor 'batched_prompt_raw'. However, when swapping two dimensions of a tensor, we should use the 'permute operation' instead of the 'reshape operation', as the 'reshape operation' can disrupt the order of the element in the tensor. To further illustrate the impact of this erroneous operation, we provide a floatmap in Figure 5. As shown in Figure 5, if a 'reshape operation' is used, key 2 will be prepended to the V value of sample 1 instead of value 1. This would render the *prompt retrieval* module ineffective, because while it can accurately predict the required prompt sets for each sample, the incorrect use of a 'reshape operation' causes confusion between prompt_k and prompt_v across samples. In contrast, using a 'permute operation' will avoid this issue.

Furthermore, we checked the official code implementation of DualPrompt[2] and found the same issue at `https://github.com/JH-LEE-KR/dualprompt-pytorch/blob/7eb457d988409a6abf97af2b121ffa62dd4b498a/prompt.py#L119-L122`. Since HidePrompt is built upon the DualPrompt, this issue has persisted. Additionally, we discovered that other researchers have raised the same concern in the issue of DualPrompt repository: `https://github.com/JH-LEE-KR/dualprompt-pytorch/issues/8`. We also found that other researchers have identified similar problems in their ongoing work based on this series of studies like `https://github.com/JingyangQiao/prompt-gradient-projection/issues/4` and `https://github.com/gulzainali98/LGCL/issues/3`. **Therefore, this implementation issue is a commonly recognized problem within the Prompt-based Continual Learning community.** We have corrected this implementation issue, using the fix mentioned in `https://github.com/JH-LEE-KR/dualprompt-pytorch/issues/8`, as illustrated in the following Listing 2. After the correction, we reproduced the experimental results of the three comparing methods, DualPrompt, S-Prompt++ and HidePrompt. **Finally, we also communicated with the authors of HidePrompt via email to request their assistance. The authors acknowledged this typo and expressed their approval of our correction plan and the reproduced experimental results in Table 1.**

Listing 2: prompt retrieval after fixing the typo.

```
num_layers, dual, batch_size, top_k, length, num_heads,          1
    heads_embed_dim = batched_prompt_raw.shape
batched_prompt_raw = batched_prompt_raw.permute(0, 2, 1, 3, 4, 5, 6) 2
batched_prompt = batched_prompt_raw.reshape(                      3
    num_layers, batch_size, dual, top_k * length, num_heads,     4
        heads_embed_dim
)                                                                5
```

---

[2]https://github.com/JH-LEE-KR/dualprompt-pytorch

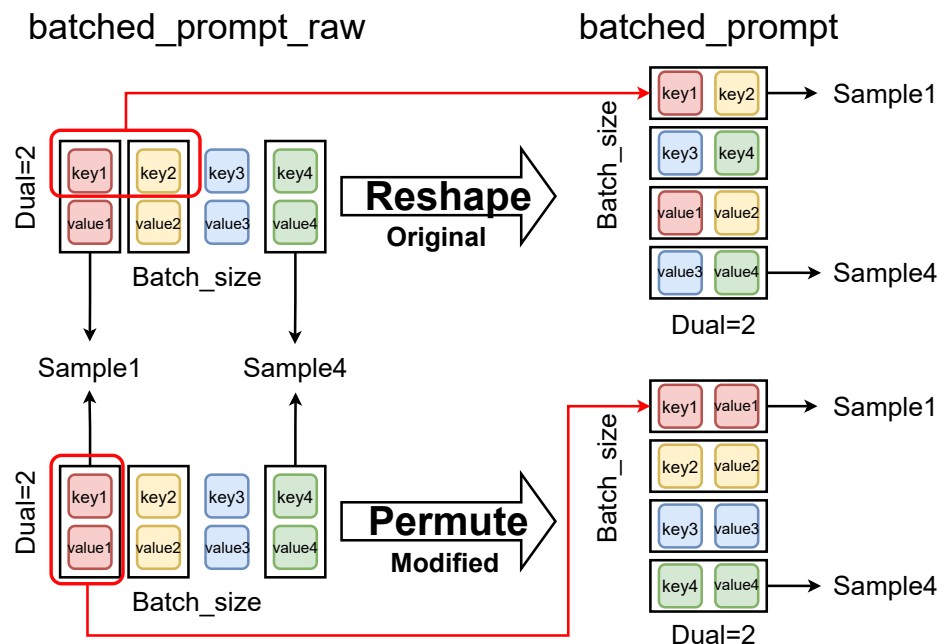

Figure 5: A floatmap shows the difference between the original code and the corrected code.

### E.2 HOW THE IMPLEMENTATION ISSUE AFFECT THE PERFORMANCE

First, the implementation issue may lead to the leakage of task ID information during testing, thereby improving performance. To better illustrate the effect of the implementation issue, we provide a specific example. Consider a batch of testing samples with a batch size of 4, all from task 3. Suppose the *prompt retrieval* module predicts the prompt sets for the 4 testing samples as: 3, 3, 2, 3, respectively. The implementation issue in the official code utilized a reshape operation (refer to Figure 5). If using a reshape operation, then sample 1 will add key3 and key3; sample 2 will add key2 and key3; sample 3 will add value3 and value3; and sample 4 will add value2 and value3. In this combination, each testing sample contains at least part of its ground truth prompt set, which increases the probability of correct predictions and thus enhances the model's performance.

Specifically, testing samples (e.g., Sample 3 from task 3) has an incorrect prompt retrieval results (where Sample 3 is misidentified as belonging to task 2), but it still utilizes the task 3 related prompt set. However, in fact, according to the basic design of PCL methods, each testing sample should utilize the prompt set predicted by the *prompt retrieval* module (e.g., Sample 3 should use the prompt set related to task 2).

Such operations can be considered as task ID information leakage (not utilizing the task ID prediction from the *prompt retrieval* module). These observations indicate that the implementation issue leads to incorrect testing processes, with task ID leakage contributing to the performance improvement.

Table 7: The results reproduced by the original official code (which has an implementation issue) and our corrected version. Here, we present the FAA results for all experiments.

| Methods | CIFAR | IMR |
|---|---|---|
| HidePrompt(-Before) | 91.07 | 72.05 |
| HidePrompt(-Before without leak information about task id) | 85.56 | 62.33 |
| HidePrompt(-After) | 85.77 | 62.42 |
| HidePrompt(-After with leak information about task id) | 92.91 | 72.69 |

To further illustrate the validity of the above analysis, we conducted ablation experiments using the original official code (which has an implementation issue) and our corrected version. The results

are shown in Table 7. Specifically, {HidePrompt(-Before)} is the result reproduced from the official code from HidePrompt Wang et al. (2024a). {HidePrompt(-After)} is the results reproduced from the corrected version. Besides, we additionally provide two experimental results: {HidePrompt (-Before without leak information about task ID)} and {HidePrompt (-After with leak information about task ID)}. Based on the above analysis, the official code of HidePrompt contains an implementation issue that leaks task ID information, allowing the model to achieve high performance. In {HidePrompt (-Before without leak information about task ID)}, we removed the task ID information leakage and observed a significant drop in model performance, which was similar to the results of {HidePrompt (-After)}. In {HidePrompt (-After with leak information about task ID)}, we mimicked the implementation in the official code and incorporated task ID information in our corrected version, resulting in a significant improvement in performance.

Table 8: Reproduced results of 3 baselines before and after fixing the implementation issue. Here, we present the FAA for all experiments.

| Methods | CIFAR | IMR | CUB |
|---|---|---|---|
| DualPrompt(-Before) | 86.16 | 65.09 | 81.50 |
| DualPrompt(-After) | 85.94 | 63.63 | 82.09 |
| S-Prompt++(-Before) | 88.73 | 65.10 | 81.89 |
| S-Prompt++(-After) | 89.26 | 63.26 | 82.57 |
| HidePrompt(-Before) | 92.47 | 72.05 | 86.56 |
| HidePrompt(-After) | 85.77 | 62.42 | 85.59 |

### E.3 HYPERPARAMETER SEARCH RESULTS

After addressing the issue mentioned in Appendix E.1, we reproduced the results of the three baselines adpoted in this paper: DualPrompt, S-Prompt++, and HidePrompt. It is important to note that we still used the official code of HidePrompt, with the only difference being that we modified the 'reshape operation' to a 'permute operation' after consulting the author, as shown in Listing 1 and Listing 2. We compared the reproduced results before and after fixing the implementation issue, as illustrated in Table 8.

We found that the performance (FAA) of DualPrompt and S-Prompt++ did not decrease after the implementation was corrected; in fact, it improved in some settings. This indicates that the implementation issue fundamentally affected the effectiveness of the *prompt retrieval* module, thus hindering the performance of PCL. Additionally, we observed a significant decrease in the performance (FAA) of HidePrompt on CIFAR and IMR, while the changes on CUB were minimal. We suspect this may be due to the fact that the previously used hyperparameters are likely no longer applicable after the corrections. Therefore, based on the author's suggestions, we conducted a grid search for the following hyperparameters of HidePrompt. The adjustable hyperparameters in HidePrompt are listed as follows:

1. sched, This hyperparameters determines how the learning rate (LR) changes during model updates as the number of epochs increases.
We search for sched from {constant, cosine, step}.

2. prompt momentum, This hyperparameters determines the proportion of prompt sets from old tasks that are retained in the prompt set for new tasks.
We search for prompt momentum from {0.01, 0.1}.

3. reg, This hyperparameters sets the weight of the contrastive loss in HidePrompt.
We search for it from {0.001, 0.01, 0.1, 0.5}.

Since HidePrompt experienced a significant performance drop only on CIFAR and IMR while maintaining good performance on CUB, we conducted the grid search for hyperparameters solely on these two benchmarks. The results are shown in Table 9 and Table 10, respectively.

Table 9: Hyperparameters of sched, prompt momentum, and reg for HidePrompt on CIFAR_INC10_TASK10. Here, we present FAA and FFM for the performance.

| sched | prompt momentum | reg | FAA ↑ | FFM ↓ |
|---|---|---|---|---|
| step | 0.01 | 0.001 | 85.85 | 6.34 |
| | | 0.01 | 85.60 | 6.57 |
| | | 0.1 | 85.77 | 6.18 |
| | | 0.5 | 85.86 | 6.35 |
| | 0.1 | 0.001 | 85.94 | 6.15 |
| | | 0.01 | 85.78 | 6.31 |
| | | 0.1 | 85.91 | 6.37 |
| | | 0.5 | 85.92 | 6.21 |
| cosine | 0.01 | 0.001 | 85.55 | 6.37 |
| | | 0.01 | 85.47 | 6.38 |
| | | 0.1 | 85.41 | 6.43 |
| | | 0.5 | 85.48 | 6.44 |
| | 0.1 | 0.001 | 85.85 | 6.16 |
| | | 0.01 | 85.78 | 6.10 |
| | | 0.1 | 85.68 | 6.17 |
| | | 0.5 | 85.69 | 6.28 |
| constant | 0.01 | 0.001 | 86.22 | 6.14 |
| | | 0.01 | 85.95 | 6.32 |
| | | 0.1 | 86.03 | 6.33 |
| | | 0.5 | 86.01 | 6.26 |
| | 0.1 | 0.001 | 86.18 | 6.13 |
| | | 0.01 | 86.03 | 6.18 |
| | | 0.1 | 86.10 | 6.22 |
| | | 0.5 | 86.10 | 6.26 |

Table 10: Hyperparameters of sched, prompt momentum, and reg for HidePrompt on IMR_INC20_TASK10. Here, we present FAA and FFM for the performance.

| sched | prompt momentum | reg | FAA ↑ | FFM ↓ |
|---|---|---|---|---|
| step | 0.01 | 0.001 | 61.00 | 8.60 |
| | | 0.01 | 61.06 | 8.43 |
| | | 0.1 | 61.30 | 8.54 |
| | | 0.5 | 60.81 | 8.41 |
| | 0.1 | 0.001 | 60.84 | 8.40 |
| | | 0.01 | 61.05 | 8.64 |
| | | 0.1 | 61.22 | 8.28 |
| | | 0.5 | 60.80 | 8.73 |
| cosine | 0.01 | 0.001 | 62.93 | 8.27 |
| | | 0.01 | 62.57 | 8.27 |
| | | 0.1 | 62.47 | 8.43 |
| | | 0.5 | 62.40 | 8.14 |
| | 0.1 | 0.001 | 62.53 | 8.74 |
| | | 0.01 | 62.45 | 8.77 |
| | | 0.1 | 62.40 | 8.76 |
| | | 0.5 | 62.33 | 9.00 |
| constant | 0.01 | 0.001 | 62.21 | 8.61 |
| | | 0.01 | 63.01 | 8.12 |
| | | 0.1 | 62.86 | 7.98 |
| | | 0.5 | 62.56 | 8.78 |
| | 0.1 | 0.001 | 62.77 | 8.13 |
| | | 0.01 | 62.31 | 7.80 |
| | | 0.1 | 62.17 | 8.05 |
| | | 0.5 | 63.05 | 8.02 |

Table 11: Impact of Distinct Threshold of $\epsilon_{\text{task}}$, $\epsilon_{\text{pre}}$ on CIFAR_INC10_TASK10.

| Settings | $\epsilon_{\text{task}}$ | $\epsilon_{\text{pre}}$ | FAA ($\uparrow$) | PRA ($\uparrow$) | FFM ($\downarrow$) |
|---|---|---|---|---|---|
| DualPrompt | Na | Na | 85.94 | 59.44 | 6.38 |
| DualPrompt [+ LW2G] | 0.50 | 0.50 | 86.89 | 60.67 | 5.44 |
| | 0.90 | 0.90 | 87.03 | 65.57 | 5.77 |
| | **0.95** | **0.95** | **86.86** | **78.33** | **6.03** |
| | 0.99 | 0.99 | 86.48 | 100.0 | 7.12 |
| S-Prompt++ | Na | Na | 89.25 | 99.52 | 4.10 |
| S-Prompt++ [+ LW2G] | 0.50 | 0.50 | 89.28 | 99.76 | 4.33 |
| | 0.90 | 0.90 | 88.54 | 100.0 | 4.48 |
| | **0.95** | **0.95** | **89.32** | **100.0** | **3.46** |
| | 0.99 | 0.99 | 89.25 | 92.32 | 6.00 |
| HidePrompt | Na | Na | 85.77 | 80.78 | 6.19 |
| HidePrompt [+ LW2G] | 0.50 | 0.50 | 86.85 | 81.70 | 5.78 |
| | 0.90 | 0.90 | 86.57 | 84.93 | 5.14 |
| | 0.95 | 0.95 | 86.93 | 90.10 | 5.02 |
| | **0.99** | **0.99** | **87.60** | **95.39** | **4.28** |

# F FURTHER RESULTS

## F.1 ABLATION STUDIES ON FOUR HYPERPARAMETERS IN LW2G

$\epsilon_{\text{task}}$, $\epsilon_{\text{pre}}$: In Gradient Projection Continual Learning (GPCL), $\epsilon$ is usually used to construct the feature space in the SVD. Previous works set it between 0.9 and 0.99. In LW2G, $\epsilon_{task}$ and $\epsilon_{pre}$ are also used for feature space construction (old knowledge and pre-trained knowledge feature space). Thus, we follow the value in Saha et al. (2021); Qiao et al. (2023); Zhao et al. (2023) and set these two parameters with the same value. We performed a grid search for appropriate values under different settings. As shown in Table 11, LW2G consistently bring performance improvement for any of the aforementioned values.

$\phi$: $\phi$ controls the pre-trained knowledge and the acquisition of new task knowledge. We performed a grid search for $\phi$ and the results are shown in Table 12.

$N$: Experiments showed significant improvement at $N = 1$ compared to $N = 0$, with no added benefit and increased computational overhead at higher values. Table 1 in the main paper indicates that SSP remains small when combined with LW2G. Thus, for efficiency and generality, we chosed $N = 1$ as the default.

## F.2 ABLATION STUDIES ON THREE MODULES IN LW2G

In this section, we provide all experiments of any combination of proposed modules and the results are shown in Table 13. The performance of any combimation can consistently outperform that of the baseline, illustrating the effectiveness of these modules.

## F.3 OVERHEAD ABOUT CALCULATION BURDEN AND TIME COST

First, LW2G only requires selecting prompt sets from the pool to calculate gradients and HFC before learning each new task. The purpose is to decide whether to learn on a newly initialized set of prompts or reuse an existing set from the prompt pool when learning a new task. After this, if opting to grow, the parameter update process does not introduce additional computation compared to the baseline. If opting not to grow, gradient projection is used during parameter updates to minimize the impact on old tasks. The computational overhead introduced by this step is a common issue in Gradient Projection Continual Learning (GPCL). This is detailed in Table 2 of the main paper, where both FLOPS and TT (Training Time) are shown to increase.

Additionally, we further analyze the memory cost. In LW2G, the extra memory is divided into two parts: a set of bases for the pre-trained knowledge space and a set of bases for the old task feature

Table 12: Impact of Distinct Threshold of $\phi$ in DualPrompt [+ LW2G] on three typical settings.

(a) CIFAR_INC10_TASK10

| $\phi$ | 0.1 | 0.2 | 0.3 | 0.4 | 0.5 | 0.6 | 0.7 | 0.8 | 0.9 | 1.0 | Baseline |
|---|---|---|---|---|---|---|---|---|---|---|---|
| FAA | 78.33 | 78.33 | 78.33 | 74.03 | **78.33** | 72.66 | 74.03 | 72.66 | 72.66 | 64.81 | 59.44 |
| PRA | 86.42 | 86.61 | 86.52 | 86.18 | **86.86** | 86.38 | 86.82 | 86.39 | 86.49 | 86.68 | 85.94 |
| FFM | 6.25 | 6.15 | 6.04 | 6.04 | **6.03** | 5.74 | 6.48 | 5.73 | 5.50 | 5.70 | 6.38 |
| SSP | 2 | 2 | 2 | 3 | **2** | 3 | 3 | 3 | 3 | 5 | 10 |

(b) IMR_INC20_TASK10

| $\phi$ | 0.1 | 0.2 | 0.3 | 0.4 | 0.5 | 0.6 | 0.7 | 0.8 | 0.9 | 1.0 | Baseline |
|---|---|---|---|---|---|---|---|---|---|---|---|
| FAA | 87.65 | 87.68 | 80.39 | 80.39 | 80.39 | **80.39** | 80.39 | 80.39 | 76.26 | 54.81 | 41.05 |
| PRA | 65.33 | 65.29 | 65.56 | 65.48 | 65.34 | **65.59** | 65.58 | 65.36 | 65.17 | 64.36 | 63.63 |
| FFM | 6.27 | 6.29 | 5.75 | 5.82 | 6.00 | **5.72** | 5.77 | 5.92 | 5.98 | 5.11 | 6.41 |
| SSP | 2 | 2 | 2 | 2 | 2 | **2** | 2 | 2 | 2 | 5 | 10 |

(c) CUB_INC20_TASK10

| $\phi$ | 0.1 | 0.2 | 0.3 | 0.4 | 0.5 | 0.6 | 0.7 | 0.8 | 0.9 | 1.0 | Baseline |
|---|---|---|---|---|---|---|---|---|---|---|---|
| FAA | 69.05 | 69.05 | **70.10** | 70.11 | 70.94 | 70.04 | 68.71 | 69.05 | 70.04 | 66.52 | 66.71 |
| PRA | 81.57 | 81.50 | **82.43** | 82.22 | 82.01 | 82.07 | 81.58 | 81.64 | 82.07 | 82.51 | 82.09 |
| FFM | 6.21 | 6.42 | **5.25** | 5.59 | 6.12 | 5.88 | 6.68 | 6.08 | 5.93 | 5.60 | 6.40 |
| SSP | 7 | 7 | **7** | 6 | 7 | 7 | 8 | 7 | 7 | 8 | 10 |

Table 13: Ablation studies in any combination of LW2G.

| Variants | FAA | PRA | SSP |
|---|---|---|---|
| DualPrompt | 63.63 | 41.05 | 10 |
| DualPrompt [+ DGA] | 65.02 | 77.68 | 2 |
| DualPrompt [+ CPK] | 64.34 | 50.39 | 10 |
| DualPrompt [+ FFT] | 64.08 | 47.17 | 10 |
| DualPrompt [+ DGA, CPK] | 65.37 | 78.13 | 2 |
| DualPrompt [+ DGA, FFT] | 65.12 | 77.90 | 2 |
| DualPrompt [+ CPK, FFT] | 64.49 | 51.20 | 10 |
| DualPrompt [+ LW2G] | **65.60** | **80.40** | **2** |

space. The size of these two sets depends on the choice of $\epsilon$ during the SVD. In the following Table 14, we analyze the memory introduced by Gradient Projection as $\epsilon$ varies. The 'Bases' indicates the total number of bases for the two sets, 'Extra Memory' represents the additional memory required. Specifically, we calculate the memory by considering each base as a tensor of length 768, stored as float32.

It is also worth reiterating that the proposed LW2G, inspired by gradient projection methods, introduces a novel and dynamic prompt growing strategy for prompt continual learning. The calculation burden and time cost are common issues with GPCL methods, which we explicitly mention in the limitations section. Although addressing this problem is beyond the scope of this study, we will consider it as a direction for future research.

## F.4 COMPARISON WITH TWO CONCURRENT WORKS

We note that two concurrent works, SEED (Rypeść et al., 2024) and PGP Qiao et al. (2023), are closely related to our motivation and methodology, respectively. In this section, we compare our proposed LW2G with these approaches.

PGP first introduced Gradient Projection-based Continual Learning (GPCL) in the context of PCL, leveraging GPCL to ensure that old knowledge is not forgotten. They demonstrated that in the scenario of PCL, the construction of the feature space could be translated into the prompt space and input space. However, unlike PGP, LW2G aims to dynamically learn whether *to grow* (initialize a new set of prompts) or *not to grow* (reuse prompts in pool) for each new task based on specific commonalities between tasks. To achieve this, LW2G adopts the idea of the *orthogonal condition* in GPCL to integrate knowledge from multiple tasks into a single set of prompts while preserving

Table 14: Discussion of the effects of memory on IMR_INC20_TASK10.

|  | $\epsilon$ | FAA | Bases | Extra Memory |
|---|---|---|---|---|
| HidePrompt | / | 85.77 | 0 | 0 |
| HidePrompt [+ LW2G] | 0.90 | 86.57 | 429 | $\leq$ 5 MB |
|  | 0.95 | 86.93 | 509 | $\leq$ 5 MB |
|  | 0.99 | 87.60 | 640 | $\leq$ 5 MB |

Table 15: Results on typical and long task settings. Here, we present DualPrompt as the baseline, with PGP and LW2G added to the baseline respectively. The best results are highlighted in bold.

| Settings | Methods | FAA ($\uparrow$) | PRA ($\uparrow$) | FFM ($\downarrow$) | SSP ($\downarrow$) |
|---|---|---|---|---|---|
|  | DualPrompt | 85.94 | 59.44 | 6.38 | 10 |
| CIFAR_INC10_TASK10 | DualPrompt **[+ PGP]** | 86.72 | 59.15 | **6.01** | 10 |
|  | DualPrompt **[+ LW2G]** | **86.86** | **78.33** | 6.03 | **2** |
|  | DualPrompt | 63.63 | 41.05 | 6.41 | 10 |
| IMR_INC20_TASK10 | DualPrompt **[+ PGP]** | 63.82 | 41.18 | **5.65** | 10 |
|  | DualPrompt **[+ LW2G]** | **65.60** | **80.40** | 5.72 | **2** |
|  | DualPrompt | 82.09 | 66.71 | 6.40 | 10 |
| CUB_INC20_TASK10 | DualPrompt **[+ PGP]** | 81.58 | 66.88 | 7.01 | 10 |
|  | DualPrompt **[+ LW2G]** | **82.43** | **70.09** | **5.25** | **7** |
|  | DualPrompt | 63.36 | 68.47 | 12.92 | 30 |
| OMNI_INC10_TASK30 | DualPrompt **[+ PGP]** | 63.74 | 67.95 | 12.97 | 30 |
|  | DualPrompt **[+ LW2G]** | **65.12** | **80.95** | **10.75** | **9** |
|  | DualPrompt | 61.85 | 69.94 | 13.50 | 60 |
| OMNI_INC5_TASK60 | DualPrompt **[+ PGP]** | 62.24 | 68.68 | 14.64 | 60 |
|  | DualPrompt **[+ LW2G]** | **63.17** | **75.31** | **12.01** | **17** |

old knowledge. Additionally, we analyze the hindrance on learning new tasks caused by the *orthogonal condition* and use the degree of inhibition under this condition as an adaptive criterion for our Dynamic Growing Approach. Furthermore, in Table 15, we compare the results of the Baseline, Baseline + PGP, and Baseline + LW2G. In both typical and long task settings, Baseline + LW2G consistently outperforms Baseline + PGP. Moreover, LW2G significantly outperforms PGP in PRA and SSP, further highlighting our approach's focus on the amount of selectable sets during the *prompt retrieval* stage in PCL.

Meanwhile, SEED proposed a continual learning method based on Mixture-of-Experts (MoE). Specifically, SEED maintains multiple sets of experts and dynamically determines which expert should be used to learn new tasks with minimal impact on old tasks. However, SEED fixes the total number of experts at the start of training, which inevitably reduces plasticity as the amount of tasks increases. In contrast, LW2G achieves complete dynamic expansion of 'experts' (which are sets of prompts in PCL) by assessing the degree of inhibition on new tasks under the *orthogonal condition*, thus eliminating the need to predefine the amount of experts.

### F.5 VISUALIZATION OF DYNAMIC PROCESS OF LW2G WITH PCL

In this section, we further demonstrate how LW2G dynamically decides *to grow* or *not to grow* based on the HFC metric before learning each task. The results are illustrated in Table 16. It can be observed that HidePrompt [+ LW2G] only requires 6 sets of prompts to surpass HidePrompt (which requires 10 sets of prompts) on the IMR benchmark.

Table 16: Variation process of HidePrompt [+ LW2G] on IMR.

| Task | Calculation Process | Minimal $Z$ | Option | Prompt sets pool |
|---|---|---|---|---|
| 1 | / | / | To Grow a new $(\boldsymbol{p}_1, \boldsymbol{k}_1)$ | $(\boldsymbol{p}_1, \boldsymbol{k}_1) \to$ Task 1 |
| 2 | $HFC_1=8.81$, $HFC_1^{pre}=7.17$ | $Z_1=1.64>0$ | To Grow a new $(\boldsymbol{p}_2, \boldsymbol{k}_2)$ | $(\boldsymbol{p}_1, \boldsymbol{k}_1) \to$ Task 1 
 $(\boldsymbol{p}_2, \boldsymbol{k}_2) \to$ Task 2 |
| 3 | $HFC_1=8.83$, $HFC_1^{pre}=7.22$ 
 $HFC_2=9.24$, $HFC_2^{pre}=8.03$ | $Z_2=1.21>0$ | To Grow a new $(\boldsymbol{p}_3, \boldsymbol{k}_3)$ | $(\boldsymbol{p}_1, \boldsymbol{k}_1) \to$ Task 1 
 $(\boldsymbol{p}_2, \boldsymbol{k}_2) \to$ Task 2 
 $(\boldsymbol{p}_3, \boldsymbol{k}_3) \to$ Task 3 |
| 4 | $HFC_1=7.34$, $HFC_1^{pre}=8.82$ 
 $HFC_2=9.26$, $HFC_2^{pre}=8.00$ 
 $HFC_3=9.15$, $HFC_3^{pre}=8.97$ | $Z_1=-1.48<0$ | Not To Grow with $(\boldsymbol{p}_1, \boldsymbol{k}_1)$ | $(\boldsymbol{p}_1, \boldsymbol{k}_1) \to$ Task 1,4 
 $(\boldsymbol{p}_2, \boldsymbol{k}_2) \to$ Task 2 
 $(\boldsymbol{p}_3, \boldsymbol{k}_3) \to$ Task 3 |
| 5 | $HFC_1=9.24$, $HFC_1^{pre}=8.12$ 
 $HFC_2=9.11$, $HFC_2^{pre}=9.07$ 
 $HFC_3=12.95$, $HFC_3^{pre}=7.24$ | $Z_2=0.04>0$ | To Grow a new $(\boldsymbol{p}_4, \boldsymbol{k}_4)$ | $(\boldsymbol{p}_1, \boldsymbol{k}_1) \to$ Task 1,4 
 $(\boldsymbol{p}_2, \boldsymbol{k}_2) \to$ Task 2 
 $(\boldsymbol{p}_3, \boldsymbol{k}_3) \to$ Task 3 
 $(\boldsymbol{p}_4, \boldsymbol{k}_4) \to$ Task 5 |
| 6 | $HFC_1=9.23$, $HFC_1^{pre}=8.02$ 
 $HFC_2=9.29$, $HFC_2^{pre}=9.23$ 
 $HFC_3=12.94$, $HFC_3^{pre}=7.29$ 
 $HFC_4=9.03$, $HFC_4^{pre}=9.14$ | $Z_4=-0.11<0$ | Not To Grow with $(\boldsymbol{p}_4, \boldsymbol{k}_4)$ | $(\boldsymbol{p}_1, \boldsymbol{k}_1) \to$ Task 1,4 
 $(\boldsymbol{p}_2, \boldsymbol{k}_2) \to$ Task 2 
 $(\boldsymbol{p}_3, \boldsymbol{k}_3) \to$ Task 3 
 $(\boldsymbol{p}_4, \boldsymbol{k}_4) \to$ Task 5,6 |
| 7 | $HFC_1=9.23$, $HFC_1^{pre}=8.08$ 
 $HFC_2=12.96$, $HFC_2^{pre}=7.33$ 
 $HFC_3=9.14$, $HFC_3^{pre}=9.25$ 
 $HFC_4=12.84$, $HFC_4^{pre}=9.16$ | $Z_3=-0.11<0$ | Not To Grow with $(\boldsymbol{p}_3, \boldsymbol{k}_3)$ | $(\boldsymbol{p}_1, \boldsymbol{k}_1) \to$ Task 1,4 
 $(\boldsymbol{p}_2, \boldsymbol{k}_2) \to$ Task 2 
 $(\boldsymbol{p}_3, \boldsymbol{k}_3) \to$ Task 3,7 
 $(\boldsymbol{p}_4, \boldsymbol{k}_4) \to$ Task 5,6 |
| 8 | $HFC_1=9.21$, $HFC_1^{pre}=8.19$ 
 $HFC_2=12.94$, $HFC_2^{pre}=7.50$ 
 $HFC_3=12.86$, $HFC_3^{pre}=9.23$ 
 $HFC_4=12.60$, $HFC_4^{pre}=9.02$ | $Z_1=1.02>0$ | To Grow a new $(\boldsymbol{p}_5, \boldsymbol{k}_5)$ | $(\boldsymbol{p}_1, \boldsymbol{k}_1) \to$ Task 1,4 
 $(\boldsymbol{p}_2, \boldsymbol{k}_2) \to$ Task 2 
 $(\boldsymbol{p}_3, \boldsymbol{k}_3) \to$ Task 3,7 
 $(\boldsymbol{p}_4, \boldsymbol{k}_4) \to$ Task 5,6 
 $(\boldsymbol{p}_5, \boldsymbol{k}_5) \to$ Task 8 |
| 9 | $HFC_1=9.41$, $HFC_1^{pre}=8.08$ 
 $HFC_2=12.95$, $HFC_2^{pre}=7.26$ 
 $HFC_3=12.83$, $HFC_3^{pre}=9.26$ 
 $HFC_4=12.61$, $HFC_4^{pre}=9.17$ 
 $HFC_5=7.98$, $HFC_5^{pre}=7.50$ | $Z_5=0.48>0$ | To Grow a new $(\boldsymbol{p}_6, \boldsymbol{k}_6)$ | $(\boldsymbol{p}_1, \boldsymbol{k}_1) \to$ Task 1,4 
 $(\boldsymbol{p}_2, \boldsymbol{k}_2) \to$ Task 2 
 $(\boldsymbol{p}_3, \boldsymbol{k}_3) \to$ Task 3,7 
 $(\boldsymbol{p}_4, \boldsymbol{k}_4) \to$ Task 5,6 
 $(\boldsymbol{p}_5, \boldsymbol{k}_5) \to$ Task 8 
 $(\boldsymbol{p}_6, \boldsymbol{k}_6) \to$ Task 9 |
| 10 | $HFC_1=9.24$, $HFC_1^{pre}=7.99$ 
 $HFC_2=12.97$, $HFC_2^{pre}=7.29$ 
 $HFC_3=12.84$, $HFC_3^{pre}=9.10$ 
 $HFC_4=12.59$, $HFC_4^{pre}=9.03$ 
 $HFC_5=7.98$, $HFC_5^{pre}=8.99$ 
 $HFC_6=6.99$, $HFC_6^{pre}=7.53$ | $Z_5=-1.01<0$ | Not To Grow with $(\boldsymbol{p}_5, \boldsymbol{k}_5)$ | $(\boldsymbol{p}_1, \boldsymbol{k}_1) \to$ Task 1,4 
 $(\boldsymbol{p}_2, \boldsymbol{k}_2) \to$ Task 2 
 $(\boldsymbol{p}_3, \boldsymbol{k}_3) \to$ Task 3,7 
 $(\boldsymbol{p}_4, \boldsymbol{k}_4) \to$ Task 5,6 
 $(\boldsymbol{p}_5, \boldsymbol{k}_5) \to$ Task 8,10 
 $(\boldsymbol{p}_6, \boldsymbol{k}_6) \to$ Task 9 |

## F.6 PERFORMANCE UNDER OTHER PTMS

To show the efficacy of proposed method under different PTMs, we evaluate our method by extending three distinct PTMs, namely IBOT1k Zhou et al. (2021), IBOT21k Zhou et al. (2021) and DINO Caron et al. (2021). The results are shown in the Table 17, Table 18 and Table 19.

Table 17: Results under IBOT21k when comparing LW2G with three baselines. The best results are highlighted in bold.

| Settings | Methods | FAA (↑) | PRA (↑) | FFM (↓) | SSP (↓) |
|---|---|---|---|---|---|
| CIFAR_INC10_TASK10 | DualPrompt | 74.03 | 72.16 | 15.93 | 10 |
| | DualPrompt **[+ LW2G]** | **74.76** | **78.33** | **13.92** | **3** |
| | S-Prompt++ | 78.37 | 78.83 | 9.00 | 10 |
| | S-Prompt++ **[+ LW2G]** | **78.83** | **75.20** | **8.69** | **3** |
| | HidePrompt | 86.12 | 85.02 | 5.98 | 10 |
| | HidePrompt **[+ LW2G]** | **86.40** | **92.06** | **5.84** | **2** |
| IMR_INC20_TASK10 | DualPrompt | 47.96 | 38.62 | 5.36 | 10 |
| | DualPrompt **[+ LW2G]** | **49.13** | **64.05** | **5.33** | **3** |
| | S-Prompt++ | 46.20 | 37.77 | 7.01 | 10 |
| | S-Prompt++ **[+ LW2G]** | **48.97** | **71.04** | **6.30** | **3** |
| | HidePrompt | 62.00 | 67.28 | **5.63** | 10 |
| | HidePrompt **[+ LW2G]** | **63.67** | **82.18** | 5.80 | **3** |

Table 18: Results under IBOT1k when comparing LW2G with three baselines. The best results are highlighted in bold.

| Settings | Methods | FAA (↑) | PRA (↑) | FFM (↓) | SSP (↓) |
|---|---|---|---|---|---|
| CIFAR_INC10_TASK10 | DualPrompt | 71.58 | 84.72 | 19.41 | 10 |
| | DualPrompt **[+ LW2G]** | **71.79** | **84.90** | **18.99** | **3** |
| | S-Prompt++ | 75.70 | 83.76 | 9.46 | 10 |
| | S-Prompt++ **[+ LW2G]** | **76.01** | **84.37** | **8.91** | **3** |
| | HidePrompt | 84.83 | 83.50 | 6.48 | 10 |
| | HidePrompt **[+ LW2G]** | **85.54** | **88.02** | **5.75** | **3** |
| IMR_INC20_TASK10 | DualPrompt | 56.68 | 38.15 | 5.18 | 10 |
| | DualPrompt **[+ LW2G]** | **56.89** | **57.57** | **5.04** | **3** |
| | S-Prompt++ | 52.38 | 39.78 | 7.18 | 10 |
| | S-Prompt++ **[+ LW2G]** | **55.82** | **55.90** | **7.13** | **3** |
| | HidePrompt | 64.77 | 67.94 | 6.90 | 10 |
| | HidePrompt **[+ LW2G]** | **65.15** | **78.27** | **4.86** | **3** |

Table 19: Results under DINO when comparing LW2G with three baselines. The best results are highlighted in bold.

| Settings | Methods | FAA (↑) | PRA (↑) | FFM (↓) | SSP (↓) |
|---|---|---|---|---|---|
| CIFAR_INC10_TASK10 | DualPrompt | 69.46 | 88.80 | 18.96 | 10 |
| | DualPrompt **[+ LW2G]** | **70.13** | **89.01** | **18.03** | **3** |
| | S-Prompt++ | **74.62** | 87.60 | **10.71** | 10 |
| | S-Prompt++ **[+ LW2G]** | 71.36 | **89.30** | 12.38 | **2** |
| | HidePrompt | 82.89 | 82.05 | 7.45 | 10 |
| | HidePrompt **[+ LW2G]** | **83.58** | **88.57** | **7.08** | **3** |
| IMR_INC20_TASK10 | DualPrompt | 52.41 | 38.74 | 5.93 | 10 |
| | DualPrompt **[+ LW2G]** | **54.22** | **75.75** | **5.77** | **2** |
| | S-Prompt++ | 50.00 | 37.72 | 6.75 | 10 |
| | S-Prompt++ **[+ LW2G]** | **65.44** | **79.35** | **6.01** | **5** |
| | HidePrompt | 62.42 | 62.07 | 8.89 | 10 |
| | HidePrompt **[+ LW2G]** | **64.04** | **86.43** | **4.82** | **2** |

# G    FURTHER DISCUSSION AND COMPARISON ON MORE BASELINES

[Revised: In this section, we first provide an analysis and discussion of a broader range of baselines. Subsequently, we demonstrate the improvements achieved by the proposed plug-in module, LW2G, when added to these baselines.

In addition to DualPrompt Wang et al. (2022b), S-Prompt++ Wang et al. (2022a), and HidePrompt Wang et al. (2024a), we further analyze other prompt-based methods, including CODAPrompt Smith et al. (2023b), OSPrompt Kim et al. (2025), and CPrompt Gao et al. (2024). Besides, some latest Lora-based methods are also enloved, e.g., C-Lora Smith et al. (2023a), InfLora Liang & Li (2024), and Hide-Lora Wang et al. (2024b).

## G.1    SUMMARY OF PREVIOUS BASELINES

**CPrompt**    CPrompt also identified the inconsistency between the training and testing stages. Specifically, they noted that the task-wise prompt set specifically trained during the training stage might not always be accurately selected during the testing stage. This inconsistency is a fundamental bottleneck for prompt-based methods. To address this, CPrompt proposed a novel strategy: during prompt-tuning for the current task, instead of concatenating only the task-wise prompt set with the input sample and feeding it into the pre-trained model, they also randomly select other prompt sets as noise and concatenate them with the input for calculation. The motivation is that since incorrect prompt set selection is possible during the testing stage, the model should still be able to make accurate predictions even when an incorrect prompt set is chosen. Thus, CPrompt attempts to mitigate the PRA issue by introducing noise into the prompt set. However, as shown in Table 20, the PRA problem remains unresolved effectively.

**CODAPrompt and OSPrompt**    To address the PRA issue, these methods calculate the similarity between a query vector and each task-wise prompt set, using the similarity as a weight. Multiple prompt sets are then fused together using these weights and concatenated with the input. This approach effectively avoids the PRA problem.

**C-Lora and InfLora**    Unlike the previous prompt-based methods, these approaches learn a LoRA parameter set for each task. Due to the characteristics of LoRA, these parameters can not only integrate with the pre-trained model's weights but also fuse with the old LoRA sets. As a result, they effectively avoid the PRA problem.

**Hide-Lora**    This is an extension of HidePrompt, which replaces the prompt set with a LoRA set. However, it still suffers from the PRA problem.

**Overall**, while the methods mentioned above partially mitigate or even avoid the PRA problem, they still rely on learning a separate prompt set or LoRA set for each task. However, recent studies have shown that learning a separate set of parameters for each incremental task hinders the potential for cross-task knowledge sharing. For example, in Yu et al. (2024), the author stated: "The use of independent adapters neglects the potential for inter-task knowledge sharing and cooperation, resulting in a limited representation capability and efficacy." Similarly, in Rypeść et al. (2024), it was found that "the ensemble of multi-experts outperforms the best individual expert."

**Therefore, exploring an efficient and effective prompt pool can not only achieve a sub-linear increase in learnable parameters with respect to the number of tasks but also effectively leverage cross-task knowledge.**

## G.2    ADVANTAGES OF LW2G FOR BASELINES

The proposed LW2G serves as a plug-in module, which is completely orthogonal to the previously mentioned prompt-based or LoRA-based methods. The advantages it offers over these baselines primarily stem from two aspects:

**Prompt Effectiveness:** Sharing a prompt set among similar tasks facilitates cross-task knowledge transfer and reduces the overhead of learnable parameters.

**Prompt Efficiency:** Utilizing fewer prompt sets improves the accuracy of prompt retrieval (PRA).

## G.3 Improvements from LW2G Applied to More Baselines

In Table 20, we further demonstrate the integration of LW2G with the six aforementioned baselines, providing numerical results to highlight the improvements achieved.

Table 20: Performance comparison on CIFAR_INC10_TASK10 and IMR_INC20_TASK10 datasets.

| Settings | | Methods | FAA ($\uparrow$) | PRA ($\uparrow$) | FFM ($\downarrow$) | SSP ($\downarrow$) |
|---|---|---|---|---|---|---|
| **CIFAR_INC10_TASK10** | Lora-based | C-Lora | 82.97 | - | 6.73 | 10 |
| | | C-Lora **[+LW2G]** | **84.69** | - | **6.24** | **2** |
| | | InfLorab5 | 86.65 | - | 6.22 | 10 |
| | | InfLorab5 **[+LW2G]** | **86.81** | - | **6.03** | **2** |
| | | Hide-Lora | 91.21 | 81.60 | 3.36 | 10 |
| | | Hide-Lora **[+LW2G]** | **92.89** | **95.30** | **2.97** | **4** |
| | Prompt-based | CPrompt | 86.13 | 69.28 | 6.00 | 10 |
| | | CPrompt **[+LW2G]** | **86.93** | **80.17** | **4.72** | **5** |
| | | CODAPrompt | 86.72 | - | 4.04 | 10 |
| | | CODAPrompt **[+LW2G]** | **87.33** | - | **3.81** | **3** |
| | | OSPrompt | 86.96 | - | 3.90 | 10 |
| | | OSPrompt **[+LW2G]** | **87.59** | - | **3.53** | **3** |
| **IMR_INC20_TASK10** | Lora-based | C-Lora | 71.95 | - | 5.82 | 10 |
| | | C-Lora **[+LW2G]** | **72.69** | - | **5.71** | **2** |
| | | InfLorab5 | 73.05 | - | 5.73 | 10 |
| | | InfLorab5 **[+LW2G]** | **73.32** | - | **4.93** | **3** |
| | | Hide-Lora | 78.86 | 65.13 | 2.07 | 10 |
| | | Hide-Lora **[+LW2G]** | **79.65** | **89.64** | **1.85** | **5** |
| | Prompt-based | CPrompt | 74.83 | 63.20 | 7.26 | 10 |
| | | CPrompt **[+LW2G]** | **76.85** | **81.01** | **6.37** | **2** |
| | | CODAPrompt | 75.73 | - | 5.17 | 10 |
| | | CODAPrompt **[+LW2G]** | **76.63** | - | **4.39** | **3** |
| | | OSPrompt | 75.55 | - | 5.36 | 10 |
| | | OSPrompt **[+LW2G]** | **76.13** | - | **4.61** | **3** |

## H  Back Forward Transfer

[Revised: Following Wang et al. (2024c), we provide a comparison of the BWT (Backward Transfer) results between LW2G+CPrompt and CPrompt. BWT measures the influence of learning task $j$ on previously learned tasks $i = 1, 2, ..., j - 1$. A larger BWT indicates that learning new tasks has a stronger **positive** impact on previously learned tasks. The results are presented in the following Table 21.]

Table 21: Backward Transfer (BWT) Comparison for CIFAR and IMR Benchmarks.

| | **CIFAR_INC10_TASK10** | **IMR_INC20_TASK10** |
|---|---|---|
| **CPrompt** | -7.25 | -6.0 |
| **CPrompt+LW2G** | -6.36 | -4.75 |

## I  Core Contributions of LW2G

[Revised: We would like to emphasize the contributions of this paper:

**1.Prompt Pool Design:** The LW2G proposed in this paper is the first prompt-based CL method to suggest that task-relatedness should guide whether to expand the prompt pool. This approach results in an efficient and effective prompt pool, which not only reduces the parameter overhead of prompts but also facilitates cross-task knowledge transfer.

**2.Novel Metric (HFC):** This paper is the first to propose using the magnitude of gradient correction to measure the degree of hindrance to learning new tasks under orthogonal constraints and to define a concrete numerical metric, HFC (Hindrance Forward Capability). While prior works in gradient-based continual learning, such as [L], have mentioned that strict orthogonality conditions can hinder

learning new tasks, no prior work has provided a clear numerical metric to quantify this hindrance. HFC is the first such metric.

**3.Dynamic Prompt Pool Expansion Using HFC:** Furthermore, this paper innovatively proposes using HFC to dynamically determine prompt pool expansion. Compared to methods such as [O] and [P], which manually set thresholds for deciding expansion, the proposed HFC not only eliminates the need for manually setting parameters (which often requires strong prior assumptions) but also provides a solid theoretical foundation to support this decision.]

