# OpenReview forum: "LW2G: Learning Whether to Grow for Prompt-based Continual Learning"
_ICLR.cc/2025/Conference — Submitted to ICLR 2025_

### Official Review · Reviewer_rwYk · 2024-10-24

**Soundness:** 3
**Presentation:** 2
**Contribution:** 2
**Rating:** 6
**Confidence:** 3

**Summary:**

This paper introduces Learning Whether to Grow (LW2G), a plug-in module for Prompt-based Continual Learning (PCL). LW2G determines whether to expand the prompt pool by adding new prompts or reusing existing ones based on the similarity between tasks, aiming to improve efficiency and knowledge transfer. Inspired by Gradient Projection Continual Learning (GPCL), the method leverages orthogonal projections to measure task interference using a new metric called Hinder Forward Capability (HFC). LW2G also introduces strategies for gradient-based prompt consistency and prompt weight reuse to further optimize learning. Experimental results show that LW2G improved the performances of three baseline PCL models.

**Strengths:**

1. Dynamically growing the prompt pool for PCL methods aligns well with the continual learning paradigm.

2. Using a gradient-projection-based metric to assess task similarity looks reasonable.

**Weaknesses:**

While the idea of leveraging orthogonal conditions to mitigate forgetting is reasonable, the paper does not sufficiently formulate this method within the context of PCL. Section 3 introduces the formulation for layer $l$, but it lacks clarity on how this formulation applies specifically for PCL, where prompts serve as the primary trainable parameters. A detailed and precise formulation is essential to fully understand the proposed method.

The reported performance of baseline models appears lower than those reported in prior work. For instance, HiDe-Prompt, when initialized with ImageNet-21K pre-trained weights, achieves FAA scores of 92.61 on CIFAR-100, 75.06 on ImageNet-R, and 86.56 on CUB, which are much higher than the ones presented in Table 1 under the same task settings. These discrepancies raise concerns about the reliability of the reported results and warrant further explanation.

The differences between the proposed LW2G and other orthogonal continual learning methods are not sufficiently highlighted. A more explicit comparison would help better position LW2G and clarify its unique contributions.


Some terminologies require clearer definitions. For example, the term “matrix with suitable dimensions” is vague. A more precise description is needed to avoid confusion.

Some minors. 1) There seem to be some messy codes in line 082. 2) Use “ViT” instead of “VIT” to align with established conventions. 3) The notion $l$ is slightly reused for the layer index and the number of bases for space $\mathcal{S}_1$.

**Questions:**

1. Does the proposed method require computing the representation matrix for every layer after learning each task? Additionally, what specific samples are used to calculate the representation matrix?

2. What does the variable $N$ in line 165 refer to?

3. How is $\mathbf{B}_t^l$ obtained? While it appears to be introduced in the supplementary material, it is recommended to briefly explain this process in the main paper as it is essential for understanding the overall method.

---

> ### Author Response · Authors · 2024-11-21
> **Response to Reviewer rwYk_1**
>
> ## Q1. Confusion due to presentation.
> - Notation $l$.
>     We apologize for the confusion caused. In the proof of Theorem 1, we have replaced the symbol $l$ with another notation. *Please refer to the Appendix B.1 in our revised manuscript.*
> - Vague description.
>    *In our revised manuscript*:
>    1. We provided a precise description torwards gradient projection (see lines 160-162).
>    2. We have added details about the method for obtaining the $B_{t}^{l}$ matrix to the main paper to make our proposed method easier to understand. (See lines 167–203).
>    3. We followed [J] and utilized 768 samples to construct the representation matrix.
>
> ## Q2.Details about baselines HidePrompt.
>
> 1.**implementation Issue in HidePrompt:**
> We kindly ask the reviewer to refer to *the caption of Table 1 and Appendix E in the submitted manuscript*, where we mentioned that the **official code of HidePrompt** has an **implementation issue regarding prompt retrieval**. This implementation issue caused a leakage problem, which led to the overestimation of results reported in the original paper. We asked the authors of HidePrompt for the fixed version of the code and reproduced the results.
>
> 2.**Comparsion with Additional Baselines:**
> Following the reviewers' suggestion, we have made extensive efforts over the past week to reproduce 6 other baselines for a more systematic comparison, including CPrompt[A], CODAPrompt[B], OS-Prompt[C], InfLora[D], C-Lora[E], Hide-Lora[F]. *The results are provided in the following Table.* LW2G demonstrates consistent improvements over each baseline by 0.19% to 2.02%, while significantly reducing the number of parameters (prompt sets or LoRA sets). For instance:
>
> - In Hide-Lora, the number of LoRA sets is reduced from 10 to 2.
> - In CPrompt, the number of prompt sets is reduced from 10 to 5.
>
> **Overall**, the experimental results demonstrate that LW2G, as a plug-in module, exhibits excellent versatility. It can be flexibly integrated with many existing methods while achieving a balance between performance and parameter overhead.
>
> 3.**Updates in the Revised Manuscript:**
> We have added this discussion in our revised manuscript *(refer to lines 1566-1647 and Appendix G).*
>
> | Settings |  | Methods | FAA($\uparrow$) | PRA($\uparrow$) | FFM($\downarrow$) | SSP($\downarrow$) |
> |---|---|---|---|---|---|---|
> | CIFAR_INC10_TASK10 | Lora-based | C-Lora | 82.97 | / | 6.73 | 10 |
> |  |  | C-Lora+LW2G | 84.69 | / | 6.24 | 2 |
> |  |  | InfLorab5 | 86.65 | / | 6.22 | 10 |
> |  |  | InfLorab5+LW2G | 86.81 | / | 6.03 | 2 |
> |  |  | Hide-Lora | 91.21 | 81.60 | 3.36 | 10 |
> |  |  | Hide-Lora+LW2G | 92.89 | 95.30 | 2.97 | 4 |
> |  | Prompt-based | CPrompt | 86.13 | 69.28 | 6.0 | 10 |
> |  |  | CPrompt+LW2G | 86.93 | 80.17 | 4.72 | 5 |
> |  |  | CODAPrompt | 86.72 | / | 4.04 | 10 |
> |  |  | CODAPrompt+LW2G | 87.33 | / | 3.81 | 3 |
> |  |  | OSPrompt | 86.96 | / | 3.90 | 10 |
> |  |  | OSPrompt+LW2G | 87.59 | / | 3.53 | 3 |
> |  IMR_INC20_TASK10 | Lora-based | C-Lora | 71.95 | / | 5.82 | 10 |
> |  |  | C-Lora+LW2G | 72.69 | / | 5.71 | 2 |
> |  |  | InfLorab5 | 73.05 | / | 5.73 | 10 |
> |  |  | InfLorab5+LW2G | 73.32 | / | 4.93 | 3 |
> |  |  | Hide-Lora | 78.86 | 65.13 | 2.07 | 10 |
> |  |  | Hide-Lora+LW2G | 79.65 | 89.64 | 1.85 | 5 |
> |  | Prompt-based | CPrompt | 74.83 | 63.20 | 7.26 | 10 |
> |  |  | CPrompt+LW2G | 76.85 | 81.01 | 6.37 | 2 |
> |  |  | CODAPrompt | 75.73 | / | 5.17 | 10 |
> |  |  | CODAPrompt+LW2G | 76.63 | / | 4.39 | 3 |
> |  |  | OSPrompt | 75.55 | / | 5.36 | 10 |
> |  |  | OSPrompt+LW2G | 76.13 | / | 4.61 | 3 |

---

> ### Author Response · Authors · 2024-11-21
> **Response to Reviewer rwYk_2**
>
> ## Q3.How LW2G differs from existing orthogonal continual learning methods.
>
> We would like to clarify to the reviewer that the **core contribution of this paper is not to propose a new gradient-projection method, but rather to address the issues of prompt set redundancy in prompt-based methods**, which hinder cross-task facilitation and make it challenging to select the correct prompt set during testing (as reflected in the proposed PRA metric). Furthermore, we explicitly mention in the conclusion that proposing a new gradient-projection method is completely orthogonal to the LW2G approach proposed in this paper.
>
> Besides, while [J] introduces gradient projection into prompt-based CL, its purpose is entirely different from ours. We have also compared LW2G with [J], and we kindly ask the reviewer to refer to Appendix F.4.
>
> To further eliminate any misunderstanding from the reviewer, we will also revise the manuscript in the revised version to clarify the core contribution as follows:
>
> 1.**Prompt Pool Design:**
> The LW2G proposed in this paper is the first prompt-based CL method to suggest that task-relatedness should guide whether to expand the prompt pool. This approach results in an efficient and effective prompt pool, which not only reduces the parameter overhead of prompts but also facilitates cross-task knowledge transfer.
>
> 2.**Novel Metric (HFC):**
> This paper is the first to propose using the magnitude of gradient correction to measure the degree of hindrance to learning new tasks under orthogonal constraints and to define a concrete numerical metric, HFC (Hindrance Forward Capability). While prior works in gradient-based continual learning, such as [G], have mentioned that strict orthogonality conditions can hinder learning new tasks, no prior work has provided a clear numerical metric to quantify this hindrance. HFC is the first such metric.
>
> 3.**Dynamic Prompt Pool Expansion Using HFC:**
> Furthermore, this paper innovatively proposes using HFC to dynamically determine prompt pool expansion. Compared to methods such as [H] and [I], which manually set thresholds for deciding expansion, the proposed HFC not only eliminates the need for manually setting parameters (which often requires strong prior assumptions) but also provides a solid theoretical foundation to support this decision.
>
> ## Q4.Does the proposed method require computing the representation matrix for every layer after learning each task?
>    We calculate the representation matrix in the same locations and using the same implementation as [I]. Specifically, the calculation of representation matrix depends on the baseline methods being compared. For example, in DualPrompt, the prompt set is only inserted into transformer blocks 0–2, so the representation matrix only needs to be calculated based on the output of transformer blocks 0–2. In S-Prompt++ and HidePrompt, the prompt set is inserted into transformer blocks 0–4, so the representation matrix needs to be calculated based on the output of transformer blocks 0–4.
>
> ## Ref:
> [A] Consistent Prompting for Rehearsal-Free Continual Learning
>
> [B] CODA-Prompt: COntinual Decomposed Attention-based Prompting for Rehearsal-Free Continual Learning
>
> [C] One-stage Prompt-based Continual Learning
>
> [D] Rethinking Gradient Projection Continual Learning: Stability / Plasticity Feature Space Decoupling,CVPR2023
>
> [E] Technical Report for ICCV 2021 Challenge SSLAD-Track3B: Transformers Are Better Continual Learners
>
> [F] Divide and not forget: Ensemble of selectively trained experts in Continual Learning,ICLR2024
>
> [G] Technical Report for ICCV 2021 Challenge SSLAD-Track3B: Transformers Are Better Continual Learners
>
> [H] Divide and not forget: Ensemble of selectively trained experts in Continual Learning,ICLR2024
>
> [I]Prompt Gradient Projection for Continual Learning, ICLR 2024.

---

> ### Comment · Reviewer_rwYk · 2024-11-23
>
> Thanks for the detailed response, which addressed most of my concerns, particularly regarding the performance of the baseline models and the clarification of the proposed contributions. I also appreciate the inclusion of additional comparison results with more baseline models, which further demonstrate the effectiveness of the proposed method. Based on these improvements, I have raised my score accordingly.

---

> ### Author Response · Authors · 2024-11-23
> **Thank you for raising the score!**
>
> ### Thank you for your recognition of our response and additional experiments! We sincerely appreciate the time and effort you've dedicated to our paper.

---

### Official Review · Reviewer_QBBM · 2024-11-02

**Soundness:** 3
**Presentation:** 3
**Contribution:** 2
**Rating:** 5
**Confidence:** 4

**Summary:**

In this paper, the authors propose a method namely Learn Whether to Grow (LW2G), wherein a shared set of prompts is utilized when several tasks share certain commonalities, and a new set is added when there are significant differences between the new task and previous tasks. The proposed  LW2G develops a metric named Hinder Forward Capability (HFC) to measure the hindrance. With HFC, an automated scheme Dynamic Growing Approach adaptively learns whether to grow with a dynamic threshold. Furthermore, a gradient-based constraint is introduced to ensure consistency between the updating prompts and pre-trained knowledge. Extensive experiments are provided.

**Strengths:**

The experiments and analysis are very thorough and well-written, and the proposed method has some novelty

**Weaknesses:**

Weakness：

1: I agree that Prompt Retrieval Accuracy (PRA) has a certain impact on performance and is an urgent problem to be solved before 2023. However, some of the lasted prompt-based methods ([1], [2], etc) and some of the lasted Adapter-Based methods ([3], [4], [5], etc.) do not need to consider PRA issues and surpass the classical methods using the matching mechanism. I have doubts whether the PRA problem should be the main motivation for building continuous learning solutions using pre-trained models.

2: There are also some methods for optimizing the prompt selection (matching) mechanism. For example, [6], [7], [8] all try to improve the matching mechanism. Compared with the above methods, the improvement of the projection method seems to be relatively limited. it may not prove that your method is superior to them.

3: The comparison is not sufficient, lacking performance comparison of some of the latest methods (as above) and the display of plug-in effects in the latest methods, etc.

4: Gradient projection and whether to choose growth are not novel in the field of CL, but they are somewhat novel in the PTM-based CIL task.

[1]: CODA-Prompt: COntinual Decomposed Attention-based Prompting for Rehearsal-Free Continual Learning
[2]: Generating Instance-level Prompts for Rehearsal-free Continual Learning
[3]: InfLoRA: Interference-Free Low-Rank Adaptation for Continual Learning
[4]: Continual Diffusion: Continual Customization of Text-to-Image Diffusion with C-LoRA
[5]: Revisiting class-incremental learning with pre-trained models: Generalizability and adaptivity are all you need
[6]: Consistent Prompting for Rehearsal-Free Continual Learning
[7]: One-stage Prompt-based Continual Learning
[8]: Evolving Parameterized Prompt Memory for Continual Learning

**Questions:**

see the weakness.

I'd like to raise my score if all my concerns are addressed.

---

> ### Author Response · Authors · 2024-11-21
> **Response to Reviewer QBBM_1**
>
> Thank you for the valuable comments. Below, we provide a point-to-point response to these comments and summarize the corresponding revisions. We hope these responses provide sufficient reasons to raise the score. We would be happy to address any further questions.
>
> ## Q1.Concerns about the motivation from PRA.
>
> We appreciate the reviewer’s constructive comments. We regret that some descriptions in our submission were not clear enough, which led to misunderstandings.
>
> 1.**Clarification on Motivation:**
> First, we agree with the reviewer’s opinion that presenting PRA improvement as the motivation for this work is not appropriate. In fact, our true motivation is simple yet intuitive: we raise concerns about the current prompt-based CL methods that "learn a new prompt set for every incremental task" and propose a solution to dynamically decide, based on task differences (via the proposed HFC metric), whether to grow a new prompt set or reuse an existing one. This ultimately forms an effective and efficient prompt set pool, which not only reduces the number of trainable parameters (fewer prompt sets), thereby improving PRA, but also facilitates cross-task knowledge transfer when similar tasks share the same prompt set. Therefore, the improvement in PRA is not the original motivation of this work but rather a by-product of the dynamic growing strategy.
>
> 2.**Comparison with Existing LoRA-Based Methods:**
> Second, we acknowledge that existing LoRA-based methods [A,B] indeed avoid the PRA issue because they aggregate all LoRA branches during inference. However, another LoRA-based method, Hide-Lora [C], does face the PRA issue. Despite this, these methods [A-C] still adhere to the approach of "learning a task-wise LoRA branch for each incremental task."
>
> Regarding the special adapter-based method [D], it avoids the PRA issue by learning a single adapter during the $1$-st incremental task and directly using this adapter for subsequent tasks. However, the information captured in the first task cannot represent all incremental tasks, especially on benchmarks with high task dissimilarity. Consequently, its performance is relatively inferior to other prompt-extend CL methods. This has been analyzed in prior works [E,F], which categorized methods into prompt-extend and prompt-fix approaches.
>
> 3.**Updates to Address Misunderstandings:**
> Finally, we acknowledge that some of our descriptions were indeed unclear and caused misunderstandings. Following the reviewer’s kind suggestions, we have made the following revisions:
>     1. *lines 1566-1647 and Appendix G.*
>     2. *lines 1663-1680 and Appendix I.*
>
> ## Q2 Comparison with DAP.
> Regarding DAP[G], DAP has the problem of batch information leakage which results in comparison unfairness. As stated by [H] in the discussions on comparison fairness, "during inference,... it is equal to directly annotating the task identity. When removing the batch information... a drastic degradation in the performance ...". **The reproduced results reported in [H] are shown in the following Table.**
> | Method  | 20S-CIFAR-100 | 20S-ImageNet-R | 20S-CUB | 10S-ImageNet-A | 20S-ObjectNet | 10S-OmniBenchmark | 5S-VTAB |
> |----------------|---------------|----------------|---------|----------------|---------------|--------------------|---------|
> | DAP  | 90.62  | 74.76   | 94.63   | 46.32  | 59.51| 80.65| 84.64   |
> | DAP w/o BI    | 58.16| 37.99| 52.05   | 21.84| 37.55| 52.53| 79.87|
>
> The accuracy declines by 4.8~42.6% with an average of 27.3% when eliminating the batch information leakage. The official code of DAP, line 210-236 in file vit.py, assumes that test samples are batched and all of them come from the same task, which is an unreasonable assumption.

---

> ### Author Response · Authors · 2024-11-21
> **Response to Reviewer QBBM_2**
>
> ## Q3.Implement comparison with more baselines.
> We thank the reviewer for their suggestion to validate the effectiveness of our method on more baselines.
>
> Since LW2G is proposed as a plug-in module, it is orthogonal to other baseline methods. Following the reviewers' suggestion, we have made extensive efforts over the past week to reproduce 6 other baselines for a more systematic comparison, including CPrompt[I], CODAPrompt[J], OS-Prompt[K], InfLora[A], C-Lora[B], Hide-Lora[C]. *The results are provided in the Table 20 in our revised manuscript.* LW2G demonstrates consistent improvements over each baseline by 0.19% to 2.02%, while significantly reducing the number of parameters (prompt sets or LoRA sets). For instance:
>
> - In Hide-Lora, the number of LoRA sets is reduced from 10 to 2.
> - In CPrompt, the number of prompt sets is reduced from 10 to 5.
>
> **Overall**, the experimental results demonstrate that LW2G, as a plug-in module, exhibits excellent versatility. It can be flexibly integrated with many existing methods while achieving a balance between performance and parameter overhead.
>
> **Updates in the Revised Manuscript:**
> We have added this discussion in our revised manuscript *(refer to lines 1566-1647 and Appendix G).*
>
> | Settings |  | Methods | FAA($\uparrow$) | PRA($\uparrow$) | FFM($\downarrow$) | SSP($\downarrow$) |
> |---|---|---|---|---|---|---|
> | CIFAR_INC10_TASK10 | Lora-based | C-Lora | 82.97 | / | 6.73 | 10 |
> |  |  | C-Lora+LW2G | 84.69 | / | 6.24 | 2 |
> |  |  | InfLorab5 | 86.65 | / | 6.22 | 10 |
> |  |  | InfLorab5+LW2G | 86.81 | / | 6.03 | 2 |
> |  |  | Hide-Lora | 91.21 | 81.60 | 3.36 | 10 |
> |  |  | Hide-Lora+LW2G | 92.89 | 95.30 | 2.97 | 4 |
> |  | Prompt-based | CPrompt | 86.13 | 69.28 | 6.0 | 10 |
> |  |  | CPrompt+LW2G | 86.93 | 80.17 | 4.72 | 5 |
> |  |  | CODAPrompt | 86.72 | / | 4.04 | 10 |
> |  |  | CODAPrompt+LW2G | 87.33 | / | 3.81 | 3 |
> |  |  | OSPrompt | 86.96 | / | 3.90 | 10 |
> |  |  | OSPrompt+LW2G | 87.59 | / | 3.53 | 3 |
> |  IMR_INC20_TASK10 | Lora-based | C-Lora | 71.95 | / | 5.82 | 10 |
> |  |  | C-Lora+LW2G | 72.69 | / | 5.71 | 2 |
> |  |  | InfLorab5 | 73.05 | / | 5.73 | 10 |
> |  |  | InfLorab5+LW2G | 73.32 | / | 4.93 | 3 |
> |  |  | Hide-Lora | 78.86 | 65.13 | 2.07 | 10 |
> |  |  | Hide-Lora+LW2G | 79.65 | 89.64 | 1.85 | 5 |
> |  | Prompt-based | CPrompt | 74.83 | 63.20 | 7.26 | 10 |
> |  |  | CPrompt+LW2G | 76.85 | 81.01 | 6.37 | 2 |
> |  |  | CODAPrompt | 75.73 | / | 5.17 | 10 |
> |  |  | CODAPrompt+LW2G | 76.63 | / | 4.39 | 3 |
> |  |  | OSPrompt | 75.55 | / | 5.36 | 10 |
> |  |  | OSPrompt+LW2G | 76.13 | / | 4.61 | 3 |

---

> ### Author Response · Authors · 2024-11-21
> **Response to Reviewer QBBM_3**
>
> ## Q4.Gradient projection and whether to choose growth are not novel in the field of CL, but they are somewhat novel in the PTM-based CIL task.
>
> We agree with the reviewer’s viewpoint, but we would like to emphasize the contributions of this paper:
>
> 1.**Prompt Pool Design:**
> The LW2G proposed in this paper is the first prompt-based CL method to suggest that task-relatedness should guide whether to expand the prompt pool. This approach results in an efficient and effective prompt pool, which not only reduces the parameter overhead of prompts but also facilitates cross-task knowledge transfer.
>
> 2.**Novel Metric (HFC):**
> This paper is the first to propose using the magnitude of gradient correction to measure the degree of hindrance to learning new tasks under orthogonal constraints and to define a concrete numerical metric, HFC (Hindrance Forward Capability). While prior works in gradient-based continual learning, such as [L], have mentioned that strict orthogonality conditions can hinder learning new tasks, no prior work has provided a clear numerical metric to quantify this hindrance. HFC is the first such metric.
>
> 3.**Dynamic Prompt Pool Expansion Using HFC:**
> Furthermore, this paper innovatively proposes using HFC to dynamically determine prompt pool expansion. Compared to methods such as [O] and [P], which manually set thresholds for deciding expansion, the proposed HFC not only eliminates the need for manually setting parameters (which often requires strong prior assumptions) but also provides a solid theoretical foundation to support this decision.
>
> ## ref:
> [A] InfLoRA: Interference-Free Low-Rank Adaptation for Continual Learning
>
> [B] Continual Diffusion: Continual Customization of Text-to-Image Diffusion with C-LoRA
>
> [C] HiDe-PET: Continual Learning via Hierarchical Decomposition of Parameter-Efficient Tuning
>
> [D] Revisiting class-incremental learning with pre-trained models: Generalizability and adaptivity are all you need
>
> [E] PECTP: Parameter-Efficient Cross-Task Prompts for Incremental Vision Transformer
>
> [F] Expandable subspace ensemble for pre-trained model-based class-incremental learning,CVPR24
>
> [G] Generating Instance-level Prompts for Rehearsal-free Continual Learning
>
> [H] Continual learning with pre-trained models: A survey
>
> [I] Consistent Prompting for Rehearsal-Free Continual Learning
>
> [J] CODA-Prompt: COntinual Decomposed Attention-based Prompting for Rehearsal-Free Continual Learning
>
> [K] One-stage Prompt-based Continual Learning
>
> [L] Rethinking Gradient Projection Continual Learning: Stability / Plasticity Feature Space Decoupling,CVPR2023
>
> [O] Technical Report for ICCV 2021 Challenge SSLAD-Track3B: Transformers Are Better Continual Learners
>
> [P] Divide and not forget: Ensemble of selectively trained experts in Continual Learning,ICLR2024

---

> > ### Comment · Reviewer_QBBM · 2024-11-23
> > **The contributions of this manuscript are not impressive and significant.**
> >
> > Thanks for your comprehensive responses. After carefully reading the response, I have a better view of the method.
> > Overall, though the proposed method improves the prompt-based methods to some extent, the idea and key techniques of this manuscript seem to be incremental, and are not attractive and impressive to me. Thus, I will not change my rating to a positive one.

---

> ### Author Response · Authors · 2024-11-23
> **Continual Response to Reviewer QBBM**
>
> ## We sincerely thank the reviewer for the time and effort spent reviewing our manuscript and communicating with us.
>
> > The contributions of this manuscript are not impressive and significant.
>
> We deeply regret that the reviewer did not find the contributions of this manuscript to be significant. **We attribute this to our failure to clearly articulate certain aspects, which may have led to some misunderstandings.** We would like to take this opportunity to further explain our work to the reviewer.
>
> We acknowledge that both prompt-based CL methods and gradient projection CL methods are either currently being explored or have already been widely studied. However, there are very few works combining these two approaches. For example:
>
>  - PGP and Inflora are two preliminary attempts, but they only perform a **straightforward combination** of the two methods **without addressing the issue of prompt set redundancy in prompt-based methods.**
>
> - Existing gradient projection CL methods are mostly **focused on a fixed model and discuss non-expansion CL methods**, such as GPM and TRGP. These studies completely overlook the potential of using gradient projection as a new perspective to rethink network expansion.
>
> In contrast, this paper delves deeper and is **the first to explore the relationship and combination of gradient projection CL methods and expansion CL methods** in terms of the degree of learning hindrance caused by strict orthogonality conditions (via the proposed HFC metric). **This is our unique and core contribution.** This exploration offers a novel perspective on expansion CL methods.
>
> ### In addition, we have followed the reviewer’s first-round suggestions and added 6 more baselines to verify the broader adaptability and effectiveness of the proposed LW2G. Moreover, we are already working on extending our approach to a wider range of expansion CL methods, which will be addressed in our future work.
>
> Finally, we sincerely ask the reviewer to reconsider their evaluation of our manuscript. We are also more than willing to address any additional concerns the reviewer may have.

---

### Official Review · Reviewer_9d2R · 2024-11-03

**Soundness:** 2
**Presentation:** 3
**Contribution:** 2
**Rating:** 5
**Confidence:** 3

**Summary:**

This paper proposes a plug-in module within existing Prompt-based Continual Learning (PCL),
called Learning Whether To Grow (LW2G). LW2G enables PCL to dynamically learn to whether to add a new set of prompts for each task (to grow) or to utilize an existing set of prompts (not to grow) based on the relationships between tasks. LW2G consists of three components: Dynamic Growing Approach (DGA), Consistency with Pre-trained Knowledge (CPK), and Facilitation for Forward Transfer (FFT). Its superiority across multiple benchmarks and various CL settings is also verified.

**Strengths:**

(1)By learning whether to grow or not to grow set of prompts, this work forms an effective and efficient prompt sets pool where each single set contains knowledge from multiple tasks, thus facilitating cross-task promotion.
(2)LW2G is a plug-in and effective module within existing PCL.

**Weaknesses:**

I have two main concerns:
(1)The proposed LW2G is mainly designed for prompt-based CL methods, and further improves their performances. However, the improvements with respect to the mainstream CL metrics, FAA and FMM, as shown in Table 1, are not significant. In particular, for some stronger baselines, S-prompt++, the performance with LW2G makes no obvious performance.
(2)It seems that the performance reported on the Hide-prompt paper is better than that in this work, in which Hide-prompt even performs better than S-prompt++.
(3) FFA or FAA? It is a little confused.

**Questions:**

(1)Does LW2G still work in the context of various pre-trained models?
(2)Is it still effective in more prompt-based CL methods, such as coda-prompt?
(3)As more recent studies show, lora-based CL methods are more effective than prompt-based in CIL tasks. So, does the proposed strategy have more advantages?
(4)When does the Theorem 1 hold? Any assumptions?

---

> ### Author Response · Authors · 2024-11-21
> **Response to Reviewer 9d2R_1**
>
> Thank you for the valuable comments. Below, we provide a point-to-point response to these comments and summarize the corresponding revisions. We hope these responses provide sufficient reasons to raise the score. We would be happy to address any further questions.
>
> ## Q1.Performance improvement on some datasets is limited.
>
> We appreciate the reviewer for pointing this out. In response, we provide the following clarifications:
>
> 1.**Two core aspects of LW2G to the improvement of FAA:**
> Our method (baseline + LW2G), as a plug-in module, dynamically decides whether to add a new prompt set based on the similarity between tasks. The advantages it brings over the baseline primarily stem from two aspects:
> (1) **Prompt effectiveness**: Sharing a prompt set among similar tasks facilitates cross-task knowledge transfer.
> (2) **Prompt efficiency**: Utilizing fewer prompt sets improves the accuracy of prompt retrieval (PRA). Previous works [A-C] have already mentioned that the accuracy of prompt set prediction significantly impacts FAA and FFM. Therefore, we propose PRA, which represents the average prediction accuracy of the prompt set for all test samples.
>
> 2.**Trade-off between trainable parameters and performance:**
> As shown in Table 1 and Table 2 of the submitted manuscript, SSP (prompt set parameters) demonstrates that **baseline + LW2G only requires 1/5 of the learnable parameters of the baseline**, yet it still consistently outperforms the baseline on FAA and FFM.
>
> 3.**Limited improvement on S-Prompt++:**
> We kindly ask the reviewer to refer to Figure 3 in the submitted manuscript. Specifically, under the CIFAR_IN10_TASK10 setting, the baseline’s PRA is already 99.52%, leaving limited potentials for LW2G to further improve it. However, even in this case, LW2G leverages cross-task knowledge through prompt effectiveness, improving FFM from 4.10% to 3.46%.
>
> ## Q2.Details about baselines HidePrompt. And other prompt-based methods, e.g. CODAPrompt. And other lora-based methods.
>
> We appreciate the reviewer for pointing this out. In response, we provide the following clarifications:
>
> 1.**implementation Issue in HidePrompt:**
> We kindly ask the reviewer to refer to *the caption of Table 1 and Appendix E in the submitted manuscript*, where we mentioned that the **official code of HidePrompt** has an **implementation issue regarding prompt retrieval**. This implementation issue caused a leakage problem, which led to the overestimation of results reported in the original paper. We asked the authors of HidePrompt for the fixed version of the code and reproduced the results.
>
> 2.**Comparsion with Additional Baselines:**
> Following the reviewers' suggestion, we have made extensive efforts over the past week to reproduce 6 other baselines for a more systematic comparison, including CPrompt[D], CODAPrompt[E], OS-Prompt[F], InfLora[G], C-Lora[H], Hide-Lora[I]. *The results are provided in the Table 20 in our revised manuscript.* LW2G demonstrates consistent improvements over each baseline by 0.19% to 2.02%, while significantly reducing the number of parameters (prompt sets or LoRA sets). For instance:
>
> - In Hide-Lora, the number of LoRA sets is reduced from 10 to 2.
> - In CPrompt, the number of prompt sets is reduced from 10 to 5.
>
> **Overall**, the experimental results demonstrate that LW2G, as a plug-in module, exhibits excellent versatility. It can be flexibly integrated with many existing methods while achieving a balance between performance and parameter overhead.
>
> 3.**Updates in the Revised Manuscript:**
> We have added this discussion in our revised manuscript *(refer to lines 1566-1647 and Appendix G).*
>
>
> | Settings |  | Methods | FAA($\uparrow$) | PRA($\uparrow$) | FFM($\downarrow$) | SSP($\downarrow$) |
> |---|---|---|---|---|---|---|
> | CIFAR_INC10_TASK10 | Lora-based | C-Lora | 82.97 | / | 6.73 | 10 |
> |  |  | C-Lora+LW2G | 84.69 | / | 6.24 | 2 |
> |  |  | InfLorab5 | 86.65 | / | 6.22 | 10 |
> |  |  | InfLorab5+LW2G | 86.81 | / | 6.03 | 2 |
> |  |  | Hide-Lora | 91.21 | 81.60 | 3.36 | 10 |
> |  |  | Hide-Lora+LW2G | 92.89 | 95.30 | 2.97 | 4 |
> |  | Prompt-based | CPrompt | 86.13 | 69.28 | 6.0 | 10 |
> |  |  | CPrompt+LW2G | 86.93 | 80.17 | 4.72 | 5 |
> |  |  | CODAPrompt | 86.72 | / | 4.04 | 10 |
> |  |  | CODAPrompt+LW2G | 87.33 | / | 3.81 | 3 |
> |  |  | OSPrompt | 86.96 | / | 3.90 | 10 |
> |  |  | OSPrompt+LW2G | 87.59 | / | 3.53 | 3 |
> |  IMR_INC20_TASK10 | Lora-based | C-Lora | 71.95 | / | 5.82 | 10 |
> |  |  | C-Lora+LW2G | 72.69 | / | 5.71 | 2 |
> |  |  | InfLorab5 | 73.05 | / | 5.73 | 10 |
> |  |  | InfLorab5+LW2G | 73.32 | / | 4.93 | 3 |
> |  |  | Hide-Lora | 78.86 | 65.13 | 2.07 | 10 |
> |  |  | Hide-Lora+LW2G | 79.65 | 89.64 | 1.85 | 5 |
> |  | Prompt-based | CPrompt | 74.83 | 63.20 | 7.26 | 10 |
> |  |  | CPrompt+LW2G | 76.85 | 81.01 | 6.37 | 2 |
> |  |  | CODAPrompt | 75.73 | / | 5.17 | 10 |
> |  |  | CODAPrompt+LW2G | 76.63 | / | 4.39 | 3 |
> |  |  | OSPrompt | 75.55 | / | 5.36 | 10 |
> |  |  | OSPrompt+LW2G | 76.13 | / | 4.61 | 3 |

---

> ### Author Response · Authors · 2024-11-21
> **Response to Reviewer 9d2R_2**
>
> ## Q3. Typos.
> Thanks, we will fix all typos and re-check the representation in the final version.
>
> ## Q4. Evaluation under other pre-trained models.
> We kindly ask the reviewer to *refer to Appendix F.6 in the submitted manuscript*. *In Appendix F.6*, we validate the effectiveness of LW2G on three additional pre-trained models: IBOT21k, IBOT1k, and DINO.
>
> The consistent improvement under different Pre-trained models illustrated that, LW2G exhibits excellent versatility.
>
> ## Q5.When does the Theorem 1 hold? Any assumptions?
>
> In Theorem 1, we prove that, for any given vector $\alpha$ in a Euclidean space and two subspaces $S_{1}$ and ​$S_{2}$ in the same Euclidean space, where the bases of $S_{1}$ are entirely contained within the bases of ​$S_{2}$, the angle between $\alpha$ and its projection onto $S_{1}$ is greater than the angle between $\alpha$ α and its projection onto $S_{2}$.
>
> In continual learning, when using gradient projection methods, the projection subspace gradually becomes larger (as more bases are added to the subspace). Consequently, the angle between real gradient and modified gradient  becomes smaller, meaning that the difference between the real gradient and the corrected gradient grows larger, which ultimately leads to increased hindrance to learning new tasks.
>
> Therefore, HFC effectively measures the degree of hindrance to new task learning under strict orthogonality conditions.
>
> As a result, Theorem 1 only requires the assumption that all vectors $b_{i}$ are in the Euclidean space.
>
> ## ref:
> [A]Hierarchical decomposition of prompt-based continual learning: Rethinking obscured sub-optimality,NIPS23
>
> [B]OVOR: OnePrompt with Virtual Outlier Regularization for Rehearsal-Free Class-Incremental Learning,ICLR24
>
> [C]KOPPA: Improving Prompt-based Continual Learning with Key-Query Orthogonal Projection and Prototype-based One-Versus-All
>
> [D] Consistent Prompting for Rehearsal-Free Continual Learning
>
> [E] CODA-Prompt: COntinual Decomposed Attention-based Prompting for Rehearsal-Free Continual Learning
>
> [F] One-stage Prompt-based Continual Learning
>
> [G] InfLoRA: Interference-Free Low-Rank Adaptation for Continual Learning
>
> [H] Continual Diffusion: Continual Customization of Text-to-Image Diffusion with C-LoRA
>
> [I] HiDe-PET: Continual Learning via Hierarchical Decomposition of Parameter-Efficient Tuning

---

> > ### Comment · Reviewer_9d2R · 2024-11-25
> > **Thank you for your detailed response**
> >
> > Thank you for your detailed response, which has addressed some of my concerns. However, it would be preferable if the proposed strategy could achieve a better balance between performance and parameter overhead. As this balance has not yet been fully achieved, I will maintain my current rating.

---

> > > ### Author Response · Authors · 2024-11-25
> > > **Thank you for recognizing our response!**
> > >
> > > We sincerely appreciate the time and effort you've dedicated to our paper, which real helped in our revised manuscript.

---

> ### Author Response · Authors · 2024-12-02
> **Summary of all the concerns and replies.**
>
> We sincerely appreciate the reviewers for dedicating their valuable time and continuous effort to reviewing and engaging in the discussions of this paper.
>
> In the discussion, reviewer 9d2R raised two main concerns:
>
> (1) The improvement from LW2G is relatively moderate, but not huge.
>
> (2) Whether LW2G is effective with other pre-trained models or baselines.
>
> ### Regarding Concern 1:
>
> We agree with the reviewer's point that a good strategy should achieve a balance between performance and parameter overhead. In fact, current prompt-based CL methods learn a task-specific prompt/adapter/lora module for each incremental task (e.g., the `e-prompt` in `DualPrompt`), which completely **ignores the parameter overhead problem** (the additional learnable modules grow linearly with the number of tasks). **In contrast, LW2G is the first approach in prompt/lora/adapter-based CL methods to consider controlling the growth of learnable modules.** This is driven not only by (1) concerns about parameter overhead, but also (2) the fact that learning separately for each task hinders cross-task knowledge facilitation and leads to low task-specific module selection accuracy (PRA problem).
>
> Therefore, the improvement brought by LW2G may not be very significant in some settings, but regardless of the size of the improvement, it comes with a dramatic reduction in the number of parameters. For example, `DualPrompt+LW2G` reduces the prompt pool from the original 10 prompt sets to 2 prompt sets, and `Hide-Lora+LW2G` reduces the lora pool from the original 10 lora sets to 5 lora sets. Moreover, we also evaluated the method on long-sequence tasks; please refer to Table 2 in the submitted manuscript. In the OMNIBENCHMARK setting, `DualPrompt (baseline)` maintains 30 prompt sets, while `DualPrompt+LW2G` only uses 9 prompt sets, with a performance improvement (FAA) of 1.76.
>
> **In summary, we believe that LW2G, as the first approach in prompt/lora/adapter-based CL methods to control the growth of learnable modules, is able to achieve consistently improved performance while drastically reducing the number of parameters.**
>
> ### Regarding Concern 2:
>
> In our submitted manuscript, we provided validation of the effectiveness of LW2G with other pre-trained models in Appendix F.6.
>
> Additionally, following the reviewer's suggestion, we conducted extensive experiments. Specifically, we evaluated **the effectiveness of LW2G on 9 baselines: DualPrompt, S-Prompt++, HidePrompt, CODAPrompt, OSPrompt, CPrompt, C-Lora, InfLora, and Hide-Lora.** These methods represent **nearly three years of state-of-the-art prompt/lora-based CL methods.** The table below shows that **LW2G consistently improves performance across all 9 baselines**, with average improvements of 0.95 and 1.07 on CIFAR and IMR, respectively. At the same time, LW2G also **significantly reduces the parameter overhead of the baselines**, with reductions of 74.17% and 66.39% on CIFAR and IMR, respectively.
>
>
> |  |  |  | CIFAR_INC10_TASK10 |  | IMR_INC20_TASK10 |  |
> |---|---|---|:---:|:---:|:---:|:---:|
> | Settings | Methods | Publication | **FAA (Performance) \ (%)** | **SSP (Overhead) \ (Num of Set)** | **FAA (Performance) \ (%)** | **SSP (Overhead) \ (Num of Set)** |
> | **Prompt-based** | DualPrompt | ECCV2022 | 85.94 | 10 | 67.30 | 10 |
> |  | DualPrompt+LW2G |  | 86.86 | 2 | 68.68 | 2 |
> |  | S-Prompt++ | Neurips2022 | 89.25 | 10 | 66.03 | 10 |
> |  | S-Prompt+++LW2G |  | 89.32 | 7 | 68.18 | 5 |
> |  | HidePrompt | Neurips2023 | 85.77 | 10 | 65.07 | 10 |
> |  | HidePrompt+LW2G |  | 87.60 | 2 | 65.84 | 6 |
> |  | CPrompt | CVPR2024 | 86.13 | 10 | 74.83 | 10 |
> |  | CPrompt+LW2G |  | 86.93 | 5 | 76.85 | 2 |
> |  | CODAPrompt | CVPR2023 | 86.72 | 10 | 75.73 | 10 |
> |  | CODAPrompt+LW2G |  | 87.33 | 3 | 76.63 | 3 |
> |  | OSPrompt | ECCV2024 | 86.96 | 10 | 75.55 | 10 |
> |  | OSPrompt+LW2G |  | 87.59 | 3 | 76.13 | 3 |
> | **Lora-based** | C-Lora | TMLR2024 | 82.97 | 10 | 71.95 | 10 |
> |  | C-Lora+LW2G |  | 84.69 | 2 | 72.69 | 2 |
> |  | InfLorab5 | CVPR2024 | 86.65 | 10 | 73.05 | 10 |
> |  | InfLorab5+LW2G |  | 86.81 | 2 | 73.32 | 3 |
> |  | Hide-Lora | arXiv2024 | 91.21 | 10 | 78.86 | 10 |
> |  | Hide-Lora+LW2G |  | 92.89 | 4 | 79.65 | 5 |
> | **Average Improvement** |  |  | **+0.95** | **-74.17%** | **+1.07** | **-66.39%** |

---

### Official Review · Reviewer_EZx5 · 2024-11-03

**Soundness:** 4
**Presentation:** 3
**Contribution:** 3
**Rating:** 6
**Confidence:** 4

**Summary:**

The paper proposes a new method called LW2G (Learning Whether to Grow) for prompt-based continual learning.
LW2G decides whether to grow the prompt pool or reuse existing prompts when learning a new task, based on measuring the hindrance of learning the new task on old prompts compared to an ideal hindrance-free scenario. I think the idea of Hinder Forward Capability (HFC) is inspiring, that measures the difference between new and old tasks as well as the hindrance imposed on learning new tasks under a strict orthogonality constraint to old task feature spaces.
Extensive experiments demonstrating the effectiveness of adding LW2G to existing prompt-based continual learning methods across multiple datasets and settings.

**Strengths:**

- LW2G is an approach that actively decides whether to grow new prompts or reuse existing ones for each new task. This aspect distinguishes LW2G from previous methods.

- Introduces a new Hinder Forward Capability (HFC) metric to quantify the hindrance of learning a new task on old prompts under orthogonality constraints.

- The three main components of LW2G - DGA, CPK, and FFT - provide a complete implementation for reusing existing prompts in continual learning tasks. However, to some extent, the approach is quite redundant, as it requires repeatedly training all the prompts to determine the optimal reuse strategy.

- The authors have shared their code and implementation, which, although I haven't personally tested it, lends some credibility to the soundness of their method. At least, from the README.md file, it appears they have provided a full implementation of the Hide-Prompt version.

- Overall, the proposed method is sound as there are so comprehensive experiments.

**Weaknesses:**

- One important aspect missing from the experiments in this paper is the average results from multiple runs. This is particularly evident in the paper, as the experimental results in Tables 1, 2, and 3 show that the proposed method has only a slight difference compared to the baselines. It's quite possible that running the baselines multiple times could yield a better result. I'm not claiming that this paper has done so. But providing the mean and standard deviation would better demonstrate statistical robustness.

- LW2G  needs for significant manual intervention and careful hyperparameter selection to ensure the prompt resule/increase strategy works appropriately. For example, HFC thresholds, CPK's soft constraint coefficient and FFT's top-N selection.

**Questions:**

see above

---

> ### Author Response · Authors · 2024-11-21
> **Response to Reviewer EZx5**
>
> Thank you for the valuable comments. Below, we provide a point-to-point response to these comments and summarize the corresponding revisions. We hope these responses provide sufficient reasons to raise the score. We would be happy to address any further questions.
>
>
> ## Q1.Statistical results with mean and std.
>
> We thank the reviewer for their kind advice and would like to provide the following clarifications:
>
> 1.In fact, we follow [B] to run our benchmarks for several different shuffles of the task class order and report the mean of these runs. We do this with a consistent seed (different for each trial) so that the results can be directly compared. *We have added the standard deviation in our revised manuscript (see lines 328-343 in Table 1).*
>
> 2.Additionally, we have made extensive efforts over the past week to reproduce 6 other baselines for a more systematic comparison, including CPrompt[A], CODAPrompt[B], OS-Prompt[C], InfLora[D], C-Lora[E], Hide-Lora[F]. *The results are provided in the Table 20 in our revised manuscript.* LW2G demonstrates consistent improvements over each baseline by 0.19% to 2.02%, while significantly reducing the number of parameters (prompt sets or LoRA sets). For instance:
>
> - In Hide-Lora, the number of LoRA sets is reduced from 10 to 2.
> - In CPrompt, the number of prompt sets is reduced from 10 to 5.
>
> **Overall**, the experimental results demonstrate that LW2G, as a plug-in module, exhibits excellent versatility. It can be flexibly integrated with many existing methods while achieving a balance between performance and parameter overhead.
>
> 3.**Updates in the Revised Manuscript:**
> We have added this discussion in our revised manuscript *(refer to lines 1566-1647 and Appendix G).*
>
>
> | Settings |  | Methods | FAA($\uparrow$) | PRA($\uparrow$) | FFM($\downarrow$) | SSP($\downarrow$) |
> |---|---|---|---|---|---|---|
> | CIFAR_INC10_TASK10 | Lora-based | C-Lora | 82.97 | / | 6.73 | 10 |
> |  |  | C-Lora+LW2G | 84.69 | / | 6.24 | 2 |
> |  |  | InfLorab5 | 86.65 | / | 6.22 | 10 |
> |  |  | InfLorab5+LW2G | 86.81 | / | 6.03 | 2 |
> |  |  | Hide-Lora | 91.21 | 81.60 | 3.36 | 10 |
> |  |  | Hide-Lora+LW2G | 92.89 | 95.30 | 2.97 | 4 |
> |  | Prompt-based | CPrompt | 86.13 | 69.28 | 6.0 | 10 |
> |  |  | CPrompt+LW2G | 86.93 | 80.17 | 4.72 | 5 |
> |  |  | CODAPrompt | 86.72 | / | 4.04 | 10 |
> |  |  | CODAPrompt+LW2G | 87.33 | / | 3.81 | 3 |
> |  |  | OSPrompt | 86.96 | / | 3.90 | 10 |
> |  |  | OSPrompt+LW2G | 87.59 | / | 3.53 | 3 |
> |  IMR_INC20_TASK10 | Lora-based | C-Lora | 71.95 | / | 5.82 | 10 |
> |  |  | C-Lora+LW2G | 72.69 | / | 5.71 | 2 |
> |  |  | InfLorab5 | 73.05 | / | 5.73 | 10 |
> |  |  | InfLorab5+LW2G | 73.32 | / | 4.93 | 3 |
> |  |  | Hide-Lora | 78.86 | 65.13 | 2.07 | 10 |
> |  |  | Hide-Lora+LW2G | 79.65 | 89.64 | 1.85 | 5 |
> |  | Prompt-based | CPrompt | 74.83 | 63.20 | 7.26 | 10 |
> |  |  | CPrompt+LW2G | 76.85 | 81.01 | 6.37 | 2 |
> |  |  | CODAPrompt | 75.73 | / | 5.17 | 10 |
> |  |  | CODAPrompt+LW2G | 76.63 | / | 4.39 | 3 |
> |  |  | OSPrompt | 75.55 | / | 5.36 | 10 |
> |  |  | OSPrompt+LW2G | 76.13 | / | 4.61 | 3 |
>
> ## Q2.Ablation Study on Hyper-parameters.
>
> We kindly ask the reviewer to refer to Appendix F.1 in the submitted manuscript, where we have conducted extensive ablation experiments on the four hyperparameters used in LW2G: $\epsilon_{\text{task}}$, $\epsilon_{\text{pre}}$, $\phi$, and $N$. As the results in Appendix F.1 demonstrate, LW2G+baseline is not dependent on carefully chosen hyperparameters to outperform the baseline. On the contrary, LW2G+baseline consistently outperforms the baseline across arbitrary choices of $\epsilon$ and $\phi$. This highlights the robustness of LW2G with respect to hyperparameters.
>
> ## ref:
> [A] Consistent Prompting for Rehearsal-Free Continual Learning
>
> [B] CODA-Prompt: COntinual Decomposed Attention-based Prompting for Rehearsal-Free Continual Learning
>
> [C] One-stage Prompt-based Continual Learning
>
> [D] InfLoRA: Interference-Free Low-Rank Adaptation for Continual Learning
>
> [E] Continual Diffusion: Continual Customization of Text-to-Image Diffusion with C-LoRA
>
> [F] HiDe-PET: Continual Learning via Hierarchical Decomposition of Parameter-Efficient Tuning

---

> > ### Comment · Reviewer_EZx5 · 2024-11-21
> >
> > I agree that many previous works lack such multiple-run experiments, and I have read the supplementary material. However, I just want to claim that the hyperparameters are slightly more complex.

---

> > > ### Author Response · Authors · 2024-11-22
> > > **Continual Response to Reviewer EZx5**
> > >
> > > ### We sincerely appreciate your great efforts in reviewing and joining the discussions of this paper.
> > >
> > >
> > > We would like to further clarify the concerns raised by the reviewers regarding the complexity of hyperparameter settings in LW2G:
> > >
> > > For different benchmarks/baselines, **there are only two hyperparameters that require adjustment, $\epsilon$ and $\phi$.** For simplicity, we set  $\epsilon_{task}$=$\epsilon_{pre}$ and $\phi$ is used to balance the pre-trained knowledge and novel knowledge from the current task. Regarding $N$, in all the experiments presented in this paper, we set **$N=1$ as the default and it does not require further tuning**.
> > >
> > >
> > >
> > >
> > > Compared to other gradient projection continual learning (CL) methods [A-C], such as TRGP[A], which adjusts layer-wise thresholds for feature outputs at different layer positions, TRGP requires tuning and setting a number of hyperparameters ($\epsilon$) equal to the number of layers when performing Singular Value Decomposition (SVD) on feature matrices. In contrast, the threshold $\epsilon$ in LW2G is shared across all prompts inserted after the model's output layers. As a result, LW2G further reduces the number of hyperparameters requiring adjustment compared to previous gradient projection CL methods. And having fewer hyperparameters that require adjustment guarantees that LW2G can be effectively adapted to other baselines.
> > >
> > > We sincerely hope you will take our response into consideration during your assessment. Should there be any remaining concerns or unclear explanations, we are more than willing to address them further.
> > >
> > > [A]TRGP: Trust Region Gradient Projection for Continual Learning, ICLR 2022.
> > >
> > > [B]Rethinking Gradient Projection Continual Learning: Stability / Plasticity Feature Space Decoupling,CVPR2023
> > >
> > > [C]InfLoRA: Interference-Free Low-Rank Adaptation for Continual Learning

---

### Official Review · Reviewer_eZ2K · 2024-11-04

**Soundness:** 2
**Presentation:** 2
**Contribution:** 2
**Rating:** 3
**Confidence:** 5

**Summary:**

This work proposes a strategy for determining the dynamic expansion of the prompt pool over time for prompt-based Continual Learning methods, based on techniques inspired by Gradient Projection Memory for Continual Learning.

**Strengths:**

The motivation of the proposed method is clear. There are theoretical supports. The experimental superiority over baselines can be seen.

**Weaknesses:**

1. It seems to me that this paper is the application of the technique in [1] to the existing methods with the criteria to expand the prompt pool. However, this technique is quite similar to those in [2], [3]. The authors should highlight the novelty and innovative contribution of this work.

2. It is not clear how the performance on CUB of HiDE is has minimal change, while those on CIFAR100 and ImageNet-R is decrease significantly. If the codebase of HiDE has some problem, why didn't the authors consider other codebases of other baselines?

3. The authors claim their method can promote positive knowledge transfer, however relevant experimental results seem to be lacking.

4. The method saves the number of parameters to learn, but storing the basis vectors of the subspaces is also expensive.

5. This work is close to [1], thus, it is essential to have the comparisons between them.

[1] Prompt Gradient Projection for Continual Learning, ICLR 2024.

[2] Beyond Not-Forgetting: Continual Learning with Backward Knowledge Transfer, NeurIPS 2022.

[3] TRGP: Trust Region Gradient Projection for Continual Learning, ICLR 2022.

**Questions:**

Please refer to the weaknesses.

---

> ### Author Response · Authors · 2024-11-21
> **Response to Reviewer eZ2K_1**
>
> Thank you for the valuable comments. Below, we provide a point-to-point response to these comments and summarize the corresponding revisions. We hope these responses provide sufficient reasons to raise the score. We would be happy to address any further questions.
>
> ## Q1.Comparison with relevant Work.
>
> We would like to clarify the innovative contributions of this paper relative to [A-C] as follows:
>
> 1.The first key difference is that [B, C] are non-expansion CL methods, whereas LW2G is a plug-in method for expansion prompt-based CL methods. Expansion CL methods do not change the structure or capacity of the neural network; therefore, the total number of network parameters remains unchanged [C]. Prompt-based CL methods, which add a new set of PEFTs for each task, clearly fall under expansion CL methods. The LW2G proposed in this paper is a plug-in method specifically designed for prompt-based CL.
>
> 2.Although both LW2G and [A] introduce gradient projection into prompt-based CL, their objectives are entirely different.
>
> - The goal of [A] is to combine gradient projection with pre-trained model-based CL methods.
> - The goal of LW2G is to dynamically decide whether to learn a new prompt set for each new task based on the relationships between tasks, thus maintaining **an efficient and effective prompt pool**.
>
> In current prompt-based CL methods, the size of the prompt pool is linearly related to the number of tasks, in that a new prompt set is learned for each incremental task. Some studies have already shown that learning a separate set of parameters for each incremental task hinders the potential for cross-task knowledge sharing. For example, in [G], the author claimed that "the use of independent adapters neglects the potential for inter-task knowledge sharing and cooperation, resulting in a limited representation capability and efficacy." Similarly, in [F], it was found that "the ensemble of multi-experts outperforms the best individual expert." **Therefore, exploring an efficient and effective prompt pool can not only achieve a sub-linear increase in learnable parameters with respect to the number of tasks, but also effectively leverage cross-task knowledge.**
>
> 3.Even though LW2G is clearly different from [A], we have also compared LW2G with [A], which was included in the original manuscript. We kindly ask the reviewer to refer to Appendix F.4.

---

> ### Author Response · Authors · 2024-11-21
> **Response to Reviewer eZ2K_2**
>
> ## Q2.Details about baselines HidePrompt.
>
> We appreciate the reviewer for pointing this out. In response, we provide the following clarifications:
>
> 1.**implementation Issue in HidePrompt:**
> We kindly ask the reviewer to refer to *the caption of Table 1 and Appendix E in the submitted manuscript*, where we mentioned that the **official code of HidePrompt** has an **implementation issue regarding prompt retrieval**. This implementation issue caused a leakage problem, which led to the overestimation of results reported in the original paper. We asked the authors of HidePrompt for the fixed version of the code and reproduced the results.
>
> 2.**Comparsion with Additional Baselines:**
> Following the reviewers' suggestion, we have made extensive efforts over the past week to reproduce 6 other baselines for a more systematic comparison, including CPrompt[H], CODAPrompt[I], OS-Prompt[J], InfLora[K], C-Lora[L], Hide-Lora[M]. *The results are provided in the Table 20 in our revised manuscript.* LW2G demonstrates consistent improvements over each baseline by 0.19% to 2.02%, while significantly reducing the number of parameters (prompt sets or LoRA sets). For instance:
>
> - In Hide-Lora, the number of LoRA sets is reduced from 10 to 2.
> - In CPrompt, the number of prompt sets is reduced from 10 to 5.
>
> **Overall**, the experimental results demonstrate that LW2G, as a plug-in module, exhibits excellent versatility. It can be flexibly integrated with many existing methods while achieving a balance between performance and parameter overhead.
>
> 3.**Updates in the Revised Manuscript:**
> We have added this discussion in our revised manuscript *(refer to lines 1566-1647 and Appendix G).*
>
>
> | Settings |  | Methods | FAA($\uparrow$) | PRA($\uparrow$) | FFM($\downarrow$) | SSP($\downarrow$) |
> |---|---|---|---|---|---|---|
> | CIFAR_INC10_TASK10 | Lora-based | C-Lora | 82.97 | / | 6.73 | 10 |
> |  |  | C-Lora+LW2G | 84.69 | / | 6.24 | 2 |
> |  |  | InfLorab5 | 86.65 | / | 6.22 | 10 |
> |  |  | InfLorab5+LW2G | 86.81 | / | 6.03 | 2 |
> |  |  | Hide-Lora | 91.21 | 81.60 | 3.36 | 10 |
> |  |  | Hide-Lora+LW2G | 92.89 | 95.30 | 2.97 | 4 |
> |  | Prompt-based | CPrompt | 86.13 | 69.28 | 6.0 | 10 |
> |  |  | CPrompt+LW2G | 86.93 | 80.17 | 4.72 | 5 |
> |  |  | CODAPrompt | 86.72 | / | 4.04 | 10 |
> |  |  | CODAPrompt+LW2G | 87.33 | / | 3.81 | 3 |
> |  |  | OSPrompt | 86.96 | / | 3.90 | 10 |
> |  |  | OSPrompt+LW2G | 87.59 | / | 3.53 | 3 |
> |  IMR_INC20_TASK10 | Lora-based | C-Lora | 71.95 | / | 5.82 | 10 |
> |  |  | C-Lora+LW2G | 72.69 | / | 5.71 | 2 |
> |  |  | InfLorab5 | 73.05 | / | 5.73 | 10 |
> |  |  | InfLorab5+LW2G | 73.32 | / | 4.93 | 3 |
> |  |  | Hide-Lora | 78.86 | 65.13 | 2.07 | 10 |
> |  |  | Hide-Lora+LW2G | 79.65 | 89.64 | 1.85 | 5 |
> |  | Prompt-based | CPrompt | 74.83 | 63.20 | 7.26 | 10 |
> |  |  | CPrompt+LW2G | 76.85 | 81.01 | 6.37 | 2 |
> |  |  | CODAPrompt | 75.73 | / | 5.17 | 10 |
> |  |  | CODAPrompt+LW2G | 76.63 | / | 4.39 | 3 |
> |  |  | OSPrompt | 75.55 | / | 5.36 | 10 |
> |  |  | OSPrompt+LW2G | 76.13 | / | 4.61 | 3 |
>
> ## Q3.Experiments of positive knowledge transfer.
>
> Following [N], we provide a comparison of the BWT (Backward Transfer) results between LW2G+CPrompt and CPrompt. BWT measures the influence of learning task $j$ on previously learned tasks $i = 1,2,...,j-1$. A larger BWT indicates that learning new tasks has a stronger **positive** impact on previously learned tasks. The results are presented in the following Table I, which shows that LW2G can effectively facilitate positive knowledge transfer.
>
> Table I: Results of BWT.
> ||CIFAR($\uparrow$)|IMR($\uparrow$)|
> |---------|-------|--------|
> ||BWT|BWT|
> |CPrompt|-7.25|-6.0|
> |CPrompt+LW2G|-6.36|-4.75|

---

> ### Author Response · Authors · 2024-11-21
> **Response to Reviewer eZ2K_3**
>
> ## Q4.storage overhead in basis vectors.
>
> We kindly ask the reviewer to refer to Appendix F.3 in the submitted manuscript, where we have already discussed the overhead introduced by storing the bases required in LW2G. Specifically, the extra overhead is divided into two parts: (1) a set of bases for the pre-trained knowledge space and (2) a set of bases for the old task feature space. The size of these two sets depends on the choice of $\epsilon$ during the SVD. In the following Table II, we analyze the memory introduced by Gradient Projection as $\epsilon$ varies. The "Bases" indicates the total number of bases for the two sets; "Extra Memory" represents the additional memory required. Specifically, we calculate the memory by considering each base as a tensor of length 768, stored as float32.
>
> The results from the following Table II demonstrate that the overhead to store bases in LW2G is moderate comparing to the improvement on FAA.
>
>
> Table II: Discussion of the effects of memory on IMR_INC20_TASK10.
>
> | |$\epsilon$ | FFA   | Bases | Extra Memory |
> |-|--------------|-------|-----------|--------------|
> | HidePrompt   | /    | 85.77     | 0            | 0          |
> |   HidePrompt [+ LW2G]           | 0.90  | 86.57     | 429          | ≤5 MB      |
> |              | 0.95  | 86.93     | 509          | ≤5 MB      |
> |              | 0.99  | 87.60     | 640          | ≤5 MB      |
>
>
> ## ref:
> [A]Prompt Gradient Projection for Continual Learning, ICLR 2024.
>
> [B]Beyond Not-Forgetting: Continual Learning with Backward Knowledge Transfer, NeurIPS 2022.
>
> [C]TRGP: Trust Region Gradient Projection for Continual Learning, ICLR 2022.
>
> [D]Rethinking Gradient Projection Continual Learning: Stability / Plasticity Feature Space Decoupling,CVPR2023
>
> [E]Technical Report for ICCV 2021 Challenge SSLAD-Track3B: Transformers Are Better Continual Learners
>
> [F]Divide and not forget: Ensemble of selectively trained experts in Continual Learning,ICLR2024
>
> [G]Boosting Continual Learning of Vision-Language Models via Mixture-of-Experts Adapters. CVPR,2024
>
> [H] Consistent Prompting for Rehearsal-Free Continual Learning
>
> [I] CODA-Prompt: COntinual Decomposed Attention-based Prompting for Rehearsal-Free Continual Learning
>
> [J] One-stage Prompt-based Continual Learning
>
> [K] InfLoRA: Interference-Free Low-Rank Adaptation for Continual Learning
>
> [L] Continual Diffusion: Continual Customization of Text-to-Image Diffusion with C-LoRA
>
> [M] HiDe-PET: Continual Learning via Hierarchical Decomposition of Parameter-Efficient Tuning
>
> [N]A Comprehensive Survey of Continual Learning: Theory, Method and Application, TPAMI,2024

---

> ### Comment · Reviewer_eZ2K · 2024-11-21
>
> 1. Indeed, TRGP [C] allows models learning a set of new parameters (in the form of scaling matrix Q) in each new task, which help the models flexibly adapt to new knowledge. Therefore, literally, **TRGP [C] and LW2G are both expansion CL methods**. Besides, **the proposed HFC in this work is similar to the criteria of Trust Region in TRGP [C].**
>
> 2.  I agree that **the good point of this work is introducing a mechanism to control the growth of the prompt pool over time**. **However**, I have a few related concerns that the authors need to address:
> ***
> -  The authors constantly mention **"effectively leverage cross-task knowledge"**. However, this seems to be a vague expectation and lacks solid reasoning.
>    - **Theorem 1** implies "large HFC and more severe hindrance on learning new tasks". However, this **does not guarantee that if HFC is controlled, the knowledge of the tasks will support each other**. The reason is that even when HFC is small, there is still a high probability that the gradient corresponding to old tasks creates an obtuse angle with the gradient of the current task. That is, updating a new task on the available prompt will likely cause the old task to be forgotten. This raises **the concern about the experimental efficiency of the method** - **why is it that occasionally sharing prompts for many tasks (in LW2G) always yields better results than using separate prompts for each task (baselines like HiDEPrompt, Sprompt, DualPrompt, etc., )?**
>
>     - **The fact that LW2G's BWTs are higher than those of baselines** (as reported by the authors) **cannot be evidence** that LW2G facilitate **positive knowledge transfer** between tasks. The simple reason is that the reported **BWTs are negative**.
>
>    - In addition, in the manuscript, **the authors claim about forward transfer (FWT), but the rebuttal response only provides results related to backward transfer (BWT), without FWT?** It seems to be unnecessary, because the numbers of BWTs are quite similar to FFM?
>
> - The storage cost for LW2G is obviously larger than the related work PGP?

---

> > ### Author Response · Authors · 2024-11-22
> > **Continual Response to Reviewer eZ2K_1**
> >
> > ## We sincerely appreciate the reviewers for dedicating their valuable time and continuous effort to reviewing and engaging in the discussions of this paper. Even if there are still some aspects that might not be entirely clear to the reviewers, we would be more than happy to further clarify and discuss any questions.
> >
> > ## Q5: Differences between LW2G and TRGP.
> >
> > 1.**Expansion or Non-Expansion.**
> >
> > We disagree that TRGP is an expansion-based CL method. The evidence supporting this claim is as follows, all quoted from the TRGP paper:
> >
> > - On page 1:
> > "In order to understand the fundamental limit of a fixed capacity neural network, we focus on non-expansion methods in this work."
> >
> > - On page 1:
> > "most existing non-expansion methods (particularly the orthogonal-projection based methods)"
> >
> > - On page 2:
> > Expansion-based methods are discussed in a separate section, whereas orthogonal-projection-based methods are categorized under another section called Memory-based methods.
> >
> > - On page 8:
> > "In the end, as shown in Table 2, we further compare with the expansion-based methods by using CIFAR-100 Sup setting. It can be seen that TRGP outperforms all other CL methods, with a fixed capacity network."
> >
> > Additionally, corresponding evidence from the official TRGP code is provided as follows:
> > https://github.com/TrellixVulnTeam/TRGP_NI5U/blob/e1a9dd935e6850b17cb3ffadf6b565e0bce9e029/main_cifar100.py#L126-L139
> >
> > ```
> > class Linear(nn.Linear):
> >
> >     def __init__(self, in_features, out_features, bias=True):
> >         super(Linear, self).__init__(in_features, out_features, bias=bias)
> >
> >         # define the scale v
> >         scale = self.weight.data.new(self.weight.size(1), self.weight.size(1))
> >         scale.fill_(0.)
> >         # initialize the diagonal as 1
> >         scale.fill_diagonal_(1.)
> >         # self.scale1 = scale.cuda()
> >         self.scale1 = nn.Parameter(scale, requires_grad=True)
> >         self.scale2 = nn.Parameter(scale, requires_grad=True)
> >
> > ```
> > Here, `self.scale1` and `self.scale2` correspond to the Q matrix mentioned in the TRGP paper. It is evident that the size of the Q matrix does not increase with the number of incremental tasks, meaning that the number of learnable parameters does not grow.
> >
> > Thus, based on the following statement:
> >
> > *"Non-Expansion CL methods do not change the structure or capacity of the neural network; therefore, the total number of network parameters remains unchanged."*
> >
> > **TRGP is a non-expansion method.**
> >
> > 2.**HFC shares similarity with TGPR.**
> >
> > - The HFC metric proposed in this paper exhibits certain formal similarities to TRGP. However, as discussed earlier, TRGP is restricted to the context of fixed model capacity. In contrast, this paper goes deeper and is the first to explore the **relationship and combination of gradient projection CL methods and expansion CL methods** with respect to the degree of learning hindrance for new tasks caused by strict orthogonality conditions, which is our unique contribution. This exploration provides a novel perspective on expansion CL methods. Although our current attempt is focus on a small but actively discussed domain of expansion CL methods, i.e., prompt-based CL methods, we are already working on extending our approach to a broader range of expansion CL methods.

---

> > ### Author Response · Authors · 2024-11-22
> > **Continual Response to Reviewer eZ2K_2**
> >
> > > **I agree that the good point of this work is introducing a mechanism to control the growth of the prompt pool over time. However, I have a few related concerns that the authors need to address:**
> >
> > > **We are deeply encouraged by your recognition of our work and are more than willing to address any potential concerns.**
> >
> >
> >
> > ## Q6 Theorem 1 implies ...... the knowledge of the tasks will support each other.
> >
> > **First and foremost, we must clarify that our submitted manuscript does not include the logic that:
> > "Controlling the size of HFC can promote leveraging cross-task knowledge."**
> > Instead, our logical chain is as follows:
> >
> > - Step 1:
> > "Learning with task-separated prompt sets hinders cross-task knowledge facilitation (as observed in current prompt/LoRA-based CL methods) [A,B]."
> >
> > - Step 2:
> > "Intuitively, similar tasks should share a set of similar parameters, which might help promote cross-task knowledge [A,B]."
> >
> > - Step 3:
> > "For rehearsal-free CL, we chose not to use replay-based methods. However, learning new tasks using the same set of parameters **without any constrain** can degrade the performance of previously learned tasks."
> >
> > - Step 4:
> > "To address this, we adopt gradient projection CL methods, which have proven that **modifying the gradient of a new task to be orthogonal to the feature space of old tasks ensures that learning the new task does not negatively impact the performance of previously learned tasks**[C,D,E]. Thus, when learning new tasks on the same set of parameters, we utilize gradient projection to ensure zero interference with old tasks."
> >
> > - Step 5:
> > "However, when task discrepancies are too large, learning on the same set of parameters becomes intuitively unreasonable and theoretically problematic. This aligns with findings that strict orthogonality conditions in gradient projection CL methods hinder the ability to learn new tasks [D,E]."
> >
> > - Step 6:
> > "To address this issue, we propose the HFC metric, which quantifies the discrepancy between the corrected gradient and the original gradient in terms of angle. If the discrepancy is too large, a new set of parameters (i.e., a new prompt set) is grown. Otherwise, the new task is learned using the existing prompt set (reusing the old prompt set)."
> >
> > ## Q7. That is, updating a new task on the available prompt will likely cause the old task to be forgotten.
> >
> > Based on the provided logical chain, in Step 4, adopting a gradient projection learning approach ensures zero forgetting of previously learned tasks.
> >
> > This is achieved by correcting the gradient of the new task to be orthogonal to the feature space of old tasks, which theoretically eliminates interference with the performance of previously learned tasks while updating the shared parameters.
> >
> > ## Q8. The fact that LW2G's BWTs ..... The simple reason is that the reported BWTs are negative.
> >
> >
> > As outlined in the complete logical chain provided above, our primary goal is to promote mutual assistance between tasks, particularly for similar tasks that share a prompt set. We acknowledge that the use of the term "forward knowledge transfer" in the submitted manuscript was imprecise, and we will revise these statements in the updated version to ensure greater clarity and accuracy.
> >
> > Additionally, BWT can indeed be negative, as reported in [F].
> >
> >
> >
> > ## Q9. The storage cost for LW2G is obviously larger than the related work PGP?
> >
> > Yes, the storage cost of LW2G is indeed higher than that of PGP.
> >
> > Specifically:
> >
> > - In PGP, a single memory is maintained, which incrementally incorporates task knowledge by storing bases that can span the feature space.
> >
> > - In LW2G, multiple sub-memories are maintained. For example:
> >
> >     If tasks 1, 2, 3, and 4 share a prompt set $p_1$, then a $\text{submemory}_1$ is maintained for $p_1$, which incrementally adds bases from task 1, 2, 3, and 4. Similarly, if tasks 5, 6, 7, and 8 share another prompt set $p_2$, a $\text{submemory}_2$ is maintained for $p_2$, which also incrementally adds bases from task 5, 6, 7, and 8.
> >
> > Thus, the total $\sum_{i}{\text{submemory}_i}$ in LW2G is larger than the single memory in PGP, as the storage requirement in LW2G depends on the number of prompt sets dynamically expanded during the process. Each new prompt set comes with an associated submemory, leading to a higher cumulative storage cost.
> >
> > [A]Divide and not forget: Ensemble of selectively trained experts in Continual Learning,ICLR2024
> >
> > [B]Boosting Continual Learning of Vision-Language Models via Mixture-of-Experts Adapters. CVPR,2024
> >
> > [C]GRADIENT PROJECTION MEMORY FOR CONTINUAL LEARNING,ICLR21
> >
> > [D]TRGP: Trust Region Gradient Projection for Continual Learning, ICLR 2022.
> >
> > [E]Rethinking Gradient Projection Continual Learning: Stability / Plasticity Feature Space Decoupling,CVPR2023
> >
> > [F]Beyond Not-Forgetting: Continual Learning with Backward Knowledge Transfer, NeurIPS 2022.

---

> ### Comment · Reviewer_eZ2K · 2024-11-23
>
> - Regarding **whether TRGP is an expansion/non-expansion CL method**:
>   - The code you showed is not comprehensive. Through the link you provided, we can see that during training, for each task, they initialize a **"memory"** **(Lines 843-851)**, which will store the trained/computed value of scaling matrix Q, which is specific for each task **(Lines 956-962)**. When in inference, they retrieve the corresponding value of Q, for each task **(Lines 987, 995)**, to predict the final result.
>
>   - Therefore, it is evident that TRGP is an expansion CL method, which is similar to prompt-based CL method (LW2G), which do not directly change the original architecture, but inject into model the specific components which aid in improving model performance. And thus, the contributions/novelties of HFC and also this work seems to be not significant.
>
> -  Regarding **Theorem 1 and "knowledge transfer" effect**, I still concern that the paper does not ensure the positive knowledge transfer between task.
>
>     -  Normally, when using gradient projection, the performance of the following task will be hinder to some extent, compare to training independently (i.e, using distinct prompt set for tasks).
>
>     - Therefore, the superior performance over the baselines as observe can result from having fewer prompts means the probability of selecting the correct prompt when testing is higher, leading to higher performance. Not because of knowledge transfer.
>
>    - In addition, negative BWT means negative backward transfer. Achieving higher BWT, but negative ---> Only means that you prevent forgetting better, cannot be sure about "promote mutual assistance between tasks".
>
> ---------------------------------
> Overall, the reviewer has made a good effort to address the concerns raised. However, the responses provided do not fully satisfy the concerns, especially regarding the novelty of the method as well as the author’s inaccurate claims regarding the method’s effectiveness. Finally, I would like not to change my rating to a positive one.

---

> > ### Author Response · Authors · 2024-11-24
> > **A plea for reassessment**
> >
> > ## We appreciate the reviewer's continued feedback. We remain committed to addressing any possible misunderstandings the reviewer may have about our work.
> >
> > We would like to first emphasize the core contributions of this paper, followed by a point-by-point response to the reviewer's further concerns.
> >
> > ### 1. Core Contribution
> > We have consistently emphasized that **the core contribution of this paper lies in proposing a dynamic strategy to determine whether to expand the prompt set** for current prompt-based CL methods. Since 2021, prompt-based CL have been actively explored, **but no prior work has proposed controlling the size of the trainable parameter set**, which is otherwise naively proportional to the number of tasks. By controlling the trainable parameter set, LW2G enables the prompt pool in prompt-based CL methods to **achieve two major advantages:**
> >
> > - Efficient:
> >
> >     We reduce the size of the prompt set, **leading to an improvement in prompt retrieval accuracy (PRA).** The PRA issue has been widely recognized in many prompt-based CL as affecting the final prediction.
> >
> > - Effective:
> >
> >     By consolidating knowledge from multiple tasks into a single prompt set, LW2G promotes cross-task knowledge transfer.
> >
> > These two advantages have played a decisive role in driving improvement.
> >
> > These contributions have already been acknowledged in the reviewer's previous responses:
> >
> > > "The good point of this work is introducing a mechanism to control the growth of the prompt pool over time."
> >
> > Therefore, we kindly ask the reviewer to re-examine the contributions of this paper rather than disproportionately overlooking certain aspects.
> >
> > ### 2. Gradient Projection Hinders Learning of New Tasks
> > We agree with the reviewer's perspective, as this is precisely what Theorem 1 demonstrates. Current gradient projection CL methods **only consider fixed-capacity models**, which increasingly hinder the learning of new tasks under strict orthogonality conditions.
> >
> > ### 3. Improvement Comes Solely from PRA, Not Knowledge Transfer.
> > LW2G's advantages stem from two factors: 1. Knowledge transfer and 2. PRA improvements brought about by reducing the size of the prompt set. We request the reviewer not to overlook LW2G's contributions to PRA improvement, which is a core aspect of this work.
> >
> > Furthermore, we understand the reviewer's concerns about knowledge transfer. **[A] has define knowledge transfer, i.e., Forward Knowledge Transfer, as follows: after learning task $i$, the knowledge gained from task $i$ can facilitate the learning of task $i+1$.**
> >
> > To further clarify, we have conducted additional experiments. In these experiments, **each row represents task $i$'s performance after training solely on that task, with the prediction results for task $i$ alone provided.** The values in parentheses indicate the PRA.
> >
> > The table includes the following methods:
> >
> >  - `DualPrompt`:
> >
> >     The baseline method.
> >
> >  - `DualPrompt+Given Task ID`:
> >
> >     Baseline with task ID explicitly provided during prompt retrieval. This guarantees the testing sample selects the correct task-specific prompt set, ensuring a PRA of 100%.
> >
> >  - `DualPrompt+LW2G`:
> >
> >     Baseline augmented with the proposed LW2G method. By reducing the size of the prompt set, PRA is noticeably improved.
> >
> > ## Conclusions:
> >
> > **`DualPrompt` learns only on the specific task $i$.** Due to its inability to select the correct prompt set during testing, its performance is hindered. Even when task IDs are explicitly provided to guarantee accurate prompt set selection during testing(`DualPrompt+Given Task ID`), performance improves but **still lags behind** `DualPrompt+LW2G`. This demonstrates that:
> >
> > #### **gradually consolidating knowledge from tasks $(i)$ and $(i+1)$ into a prompt set**
> > results in a more effective prompt set than
> > #### **one learned solely on task $i+1$**.
> >
> >
> > Therefore, we believe this provides strong evidence supporting the knowledge transfer of LW2G.
> >
> >
> > Finally, we understand the reviewer's concerns. However, the combination of experimental results, theoretical analyses, and discussions with the reviewer demonstrates that **LW2G achieves a well-balanced trade-off between parameter efficiency and model performance.** Furthermore, we have validated LW2G on an extensive set of baselines **(up to 9 baselines)**. All of this evidence suggests that LW2G performs commendably.
> >
> > ## We respectfully request that the reviewer reconsider the contributions of this work and take this into account in their rating.

---

> > > ### Author Response · Authors · 2024-11-24
> > > **Continual response to Reviewer eZ2K**
> > >
> > > | CIFAR \| Task ID | 1 | 2 | 3 | 4 | 5 | 6 | 7 | 8 | 9 | 10 |
> > > |---|---|---|---|---|---|---|---|---|---|---|
> > > | DualPrompt | 99.30(100.00) | 96.10(84.40) | 93.90(85.00) | 91.00(69.90) | 92.50(64.40) | 82.70(68.90) | 89.00(60.50) | 90.70(60.10) | 92.60(74.20) | 88.06(42.70) |
> > > | DualPrompt+Given Task ID | 99.30(100.00) | 96.32(100.00) | 94.39(100.00) | 92.17(100.00) | 93.51(100.00) | 83.07(100.00) | 89.41(100.00) | 91.00(100.00) | 92.76(100.00) | 89.91(100.00) |
> > > | DualPrompt+LW2G | 99.40(100.00) | 96.40(100.00) | 95.30(99.90) | 92.50(100.00) | 93.70(100.00) | 83.10(68.10) | 89.30(67.60) | 90.80(73.60) | 92.80(78.90) | 90.00(75.70) |
> > >
> > > | IMR \| Task ID | 1 | 2 | 3 | 4 | 5 | 6 | 7 | 8 | 9 | 10 |
> > > |---|---|---|---|---|---|---|---|---|---|---|
> > > | DualPrompt | 70.88(100.00) | 74.64(77.21) | 68.93(71.35) | 67.67(56.20) | 67.12(56.44) | 68.11(42.20) | 67.68(55.56) | 59.47(27.27) | 75.98(42.11) | 69.28(61.20) |
> > > | DualPrompt+Given Task ID | 70.88(100.00) | 76.30(100.00) | 73.59(100.00) | 71.33(100.00) | 70.90(100.00) | 71.36(100.00) | 68.43(100.00) | 63.21(100.00) | 76.88(100.00) | 71.32(100.00) |
> > > | DualPrompt+LW2G | 71.16(100.00) | 76.64(100.00) | 76.70(100.00) | 73.12(99.70) | 72.10(100.00) | 73.10(100.00) | 70.71(100.00) | 64.40(100.00) | 76.97(52.63) | 71.43(67.86) |
> > >
> > > [A] A Comprehensive Survey of Continual Learning:
> > >  Theory, Method and Application, TPAMI,2024

---

> > > > ### Author Response · Authors · 2024-11-24
> > > > **Continual response to Reviewer eZ2K_1**
> > > >
> > > > ### 4. Regarding expansion or not.
> > > >
> > > > whether TRGP is an expansion or non-expansion method
> > > > We prefer not to continue debating this point with the reviewer, as we believe this discussion neither should nor needs to influence the evaluation of this paper. As we mentioned in our previous response, **the authors of TRGP explicitly described it multiple times in the original paper as a non-expansion method.**
> > > >
> > > > Additionally, as noted earlier, **TRGP was designed for task-incremental learning (TIL)**, where the task ID is provided during testing. Therefore, the lines referenced by the reviewer (Lines 956–962 and Lines 987, 995) involve retrieval with a given task ID. **When TRGP is applied to class-incremental learning (CIL)**, it inevitably faces the query retrieval problem, and retrieval accuracy would impact its overall performance.
> > > >
> > > > From this perspective, the LW2G proposed in this paper has potential as a plug-in for the retrieval stage in TRGP, further enhancing its applicability.

---

> ### Comment · Reviewer_eZ2K · 2024-11-24
>
> 1. The reason I discussed the essence of TRGP (not just the claims in their paper) is that I find there are quite many similarities between your method and TRGP, which makes your method seem incremental.
>
> 2. Regarding your extended experiment you have provided, please let me know how you apply LW2G into DualP. Because this method has 2 kinds of prompts, including G-prompts (shared), and E-prompts (task-specific). This is important to evaluate the role of knowledge transfer in model performance.

---

> > ### Author Response · Authors · 2024-11-24
> > **Thanks for continual feedback**
> >
> > ## We greatly appreciate the reviewer’s prompt response and constructive feedback. Below, we address the concerns raised:
> >
> >  - **Scope of TRGP and Core Contribution**
> > We have emphasized that TRGP specifically addresses **task-incremental learning while focusing on fixed-size models**, which significantly limits its applicability. In contrast, our core contribution lies in tackling the expansion problem, where we introduce HFC, a method leveraging gradient projection. Furthermore, we have transparently acknowledged in the main text that our metric was inspired by gradient projection-based continual learning methods and have appropriately cited them in the main paper.
> >
> >  - **Clarifications on DualPrompt and Additional Results**
> > Regarding DualPrompt, we do not modify the `g-prompt`; the LW2G approach only determines whether to add or reuse the `e-prompt`, which aligns with the code implementation in [A]. Additionally, to alleviate the reviewer’s concerns, we further provide the following results, where we adopt S-Prompt++ as the baseline. Since S-Prompt++ exclusively uses the e-prompt, we believe this comparison better highlights our method's advantages.
> >
> > Finally, we thank the reviewer for the continued discussions and insightful comments. We remain happy to address further questions or provide additional clarifications.
> >
> > | IMR \| Task ID | 1 | 2 | 3 | 4 | 5 | 6 | 7 | 8 | 9 | 10 |
> > |---|---|---|---|---|---|---|---|---|---|---|
> > | S-Prompt++ | 74.39(100.00) | 74.36(68.09) | 69.90(81.07) | 67.07(50.15) | 70.14(61.10) | 65.12(57.14) | 63.30(68.35) | 60.61(54.92) | 75.99(52.96) | 67.14(60.23) |
> > | S-Prompt++/+Given Task ID | 74.39(100.00) | 76.33(100.00) | 72.29(100.00) | 74.16(100.00) | 76.50(100.00) | 70.19(100.00) | 69.31(100.00) | 65.34(100.00) | 77.01(100.00) | 70.20(100.00) |
> > | S-Prompt++/+LW2G | 74.67(100.00) | 76.35(100.00) | 74.76(100.00) | 75.22(100.00) | 76.44(100.00) | 73.75(100.00) | 70.03(100.00) | 65.53(100.00) | 77.30(52.96) | 70.47(66.905) |
> >
> >
> > [A]Prompt Gradient Projection for Continual Learning, ICLR 2024.

---

> ### Comment · Reviewer_eZ2K · 2024-11-24
> **The experimental results are unreliable**
>
> Thank you for your response.
>
> I am afraid that the experimental results you provided are not convincing for me. I will point out the reasons below - regarding the knowledge transferability between tasks of this method:
>
> - Regarding Backward transfer (BWT): It is very hard to transfer knowledge from new task to previous ones. The reason is that when you apply gradient projection, the behavior of model w.r.t old tasks will be almost unchanged.
>
> - Regarding Forward transfer (FWT):
>   - In the post (https://openreview.net/forum?id=QZuZmfLLRG&noteId=OjthBJVEZs), you agreed with me that "Gradient Projection Hinders Learning of New Tasks".
>   - Besides, in another post (https://openreview.net/forum?id=QZuZmfLLRG&noteId=aeCyGhrh3z), you admitted that "the use of the term "forward knowledge transfer" in the submitted manuscript was imprecise".
>
>   ----> These arguments mean there is no positive FWT
>
> Based on these, the experimental results you provided, and also your state about "cross task knowledge transfer" are unbelievable. It seems to me that the authors are trying to polish their paper, but it is not reasonable.

---

> ### Author Response · Authors · 2024-11-29
> **Summary of all the concerns and replies. (1/2)**
>
> We sincerely appreciate the reviewers for dedicating their valuable time and continuous effort to reviewing and engaging in the discussions of this paper.
>
> Below, we summarize Reviewer eZ2K’s main concerns during the rebuttal process and provide a point-by-point response.
>
> > ### Main Concern1: Is LW2G very similar to TRGP or other gradient projection CL methods, thus making the contribution of LW2G incremental?
>
> > ### Reply: **No.**
>
> LW2G is fundamentally different from previous gradient projection CL methods. The evidence is as follows:
>
> 1.Different Types of Continual Learning Methods.
>
> Existing gradient projection CL methods focus on **fixed models and non-expansion CL methods**, such as GPM and TRGP. In contrast, LW2G focus on **expansion CL methods** (prompt-based CL methods). Specifically, LW2G is **the first to explore** the relationship and combination of **gradient projection CL methods with *expansion CL methods***. It examines the degree of learning hindrance caused by strict orthogonality conditions (via the proposed HFC metric). Current prompt-based CL methods learn a task-specific prompt/adapter/lora module for each incremental task (e.g., e-prompt in DualPrompt), which hinders cross-task knowledge facilitation and leads to a low task-specific module selection accuracy (PRA problem). **The core contribution of LW2G** is the design of **a dynamic mechanism to control the growth of the prompt set**, which brings two key advantages:
>
>   - **Prompt effectiveness**: "Gradually consolidating knowledge from multiple tasks into one prompt set" is more effective than using "task-specific prompt sets," thus **promoting cross-task knowledge transfer**.
>   - **Prompt efficiency**: "Fewer prompt sets" **improve PRA and reduce parameter overhead**.
>
> Therefore, LW2G enables the formation of an efficient and effective prompt pool for expansion CL methods(prompt/lora/adapter-based CL methods)
>
> 2.Different Continual Learning Scenarios.
>
> Existing gradient projection CL methods mainly focus on **task-incremental learning** (with task IDs provided during testing). However, current prompt-based CL methods consider the more challenging class-incremental learning scenario (without task IDs during testing). **LW2G is designed as a plug-in module for prompt-based CL methods, addressing this more difficult scenario.**
>
> 3.Some details.
>
> Regarding the reviewer's mention of the TRPG official code ((Lines 956-962)), we conducted a line-by-line inspection, and the fact is that **the scaling matrix Q does not expand with the task**, but rather, it is a **fixed-size matrix Q** that learns from scratch for each task. This differs from the definition of expansion CL methods and therefore **belongs to non-expansion CL methods.**
>
> Regarding the reviewer's mention of the TRPG official code ((Lines 987, 995)), we also performed a line-by-line check. The fact is that since **TRGP is designed for task-incremental learning**, during inference, the scaling matrix Q is retrieved with the task-id. **Therefore, if TRGP is used to address class-incremental learning, it will also face the issue of scaling matrix Q retrieval accuracy (PRA problem).** In this case, the scaling matrix Q is similar to the prompt set in prompt-based CL methods, which learns specifically for each task. Since the accuracy of selecting the scaling matrix Q also affects the performance of TRGP, **LW2G+TRGP can also be used to control the growth of the scaling matrix Q in TRGP.**
>
> In summary, LW2G has a wider range of applications than TRGP and can also serve as an optional enhancement choice when TRGP is used in class-incremental learning scenarios.

---

> > ### Author Response · Authors · 2024-11-29
> > **Summary of all the concerns and replies. (2/2)**
> >
> > > ### Main Concern2:  Can Baseline+LW2G Facilitate Cross-Task Knowledge Promotion?
> >
> > > ### Reply: **Yes.**
> >
> > `Baseline+LW2G` is able to effectively utilize cross-task knowledge compared to the `baseline`. The evidence is as follows:
> >
> > According to the definition in [1], cross-task knowledge transfer specifically refers to Forward Knowledge Transfer (FWT) and Backward Knowledge Transfer (BWT). FWT represents the influence that learning a task $t$ has on the performance of a previous task $k<t$, while BWT represents the influence that learning a task $t$ has on the performance of a future task $k>t$.
> >
> > From Table 1, the BWT of `CPrompt (baseline)` is -7.25. After combining with `LW2G (baseline+LW2G)`, the BWT improves by 0.89 and 1.25 on CIFAR and IMR, respectively. This indicates that LW2G enables a positive forward influence on previous tasks $k<t$ after learning task $t$.
> >
> > From Tables 2, 3, and 4, `Baseline+LW2G` consistently outperforms `Baseline` in prediction accuracy for every incremental task. This shows that LW2G enables a positive influence on future tasks $k>t$ after learning task $t$.
> >
> > Furthermore, we found that the performance of `Baseline+LW2G` even exceeds that of `Baseline+Given Task ID` (where the correct task-specific prompt set is used for each testing sample). This suggests that through LW2G, **gradually consolidating knowledge from multiple tasks, e.g., task $(i)$ and task $(i+1)$, into a single prompt set**, is much more effective than using **a prompt set that is learned solely on task $(i+1)$**.
> >
> > Table1
> > ||CIFAR($\uparrow$)|IMR($\uparrow$)|
> > |---------|-------|--------|
> > ||BWT|BWT|
> > |CPrompt|-7.25|-6.0|
> > |CPrompt+LW2G|-6.36|-4.75|
> >
> > Table2
> > | CIFAR \| Task ID | 1 | 2 | 3 | 4 | 5 | 6 | 7 | 8 | 9 | 10 |
> > |---|---|---|---|---|---|---|---|---|---|---|
> > | DualPrompt | 99.30(100.00) | 96.10(84.40) | 93.90(85.00) | 91.00(69.90) | 92.50(64.40) | 82.70(68.90) | 89.00(60.50) | 90.70(60.10) | 92.60(74.20) | 88.06(42.70) |
> > | DualPrompt+Given Task ID | 99.30(100.00) | 96.32(100.00) | 94.39(100.00) | 92.17(100.00) | 93.51(100.00) | 83.07(100.00) | 89.41(100.00) | 91.00(100.00) | 92.76(100.00) | 89.91(100.00) |
> > | DualPrompt+LW2G | 99.40(100.00) | 96.40(100.00) | 95.30(99.90) | 92.50(100.00) | 93.70(100.00) | 83.10(68.10) | 89.30(67.60) | 90.80(73.60) | 92.80(78.90) | 90.00(75.70) |
> >
> > Table3
> > | IMR \| Task ID | 1 | 2 | 3 | 4 | 5 | 6 | 7 | 8 | 9 | 10 |
> > |---|---|---|---|---|---|---|---|---|---|---|
> > | DualPrompt | 70.88(100.00) | 74.64(77.21) | 68.93(71.35) | 67.67(56.20) | 67.12(56.44) | 68.11(42.20) | 67.68(55.56) | 59.47(27.27) | 75.98(42.11) | 69.28(61.20) |
> > | DualPrompt+Given Task ID | 70.88(100.00) | 76.30(100.00) | 73.59(100.00) | 71.33(100.00) | 70.90(100.00) | 71.36(100.00) | 68.43(100.00) | 63.21(100.00) | 76.88(100.00) | 71.32(100.00) |
> > | DualPrompt+LW2G | 71.16(100.00) | 76.64(100.00) | 76.70(100.00) | 73.12(99.70) | 72.10(100.00) | 73.10(100.00) | 70.71(100.00) | 64.40(100.00) | 76.97(52.63) | 71.43(67.86) |
> >
> > Table4
> > | IMR \| Task ID | 1 | 2 | 3 | 4 | 5 | 6 | 7 | 8 | 9 | 10 |
> > |---|---|---|---|---|---|---|---|---|---|---|
> > | S-Prompt++ | 74.39(100.00) | 74.36(68.09) | 69.90(81.07) | 67.07(50.15) | 70.14(61.10) | 65.12(57.14) | 63.30(68.35) | 60.61(54.92) | 75.99(52.96) | 67.14(60.23) |
> > | S-Prompt++/+Given Task ID | 74.39(100.00) | 76.33(100.00) | 72.29(100.00) | 74.16(100.00) | 76.50(100.00) | 70.19(100.00) | 69.31(100.00) | 65.34(100.00) | 77.01(100.00) | 70.20(100.00) |
> > | S-Prompt++/+LW2G | 74.67(100.00) | 76.35(100.00) | 74.76(100.00) | 75.22(100.00) | 76.44(100.00) | 73.75(100.00) | 70.03(100.00) | 65.53(100.00) | 77.30(52.96) | 70.47(66.905) |
> >
> > [1]Gradient Episodic Memory for Continual Learning,nips,2017

---

### Author Response · Authors · 2024-11-23
**To all Reviewers**

We sincerely appreciate the time and effort you have dedicated to reviewing our paper. We have carefully considered your valuable feedback and have made comprehensive revisions throughout the manuscript. Detailed responses to your comments will be provided in the respective replies. We remain open to any additional suggestions you may have. If our revisions address your concerns satisfactorily, we would be truly grateful if you could consider raising our score.

---

### Meta-Review · Area_Chair_34si · 2024-12-17

**Metareview:**

This paper aims to improve the efficiency and knowledge transfer for prompt-based continual learning. The authors propose a method called Learning Whether to Grow (LW2G) to decide whether to grow the prompt pool or reuse existing prompts when learning a new task. Extensive experiments are conducted to evaluate the performance of the proposed method. Several reviewers raise their concerns about the novelty, rationality, definition of concepts, experimental support, comparison results and manual intervention, some key concerns remain after the rebuttal. Based on the above considerations, I think the current manuscript does not match the ICLR’s requirement and I do not recommend to accept this manuscript.

**Additional Comments On Reviewer Discussion:**

The authors provided rebuttals for each reviewer and all reviewers provided responses. Reviewer rwYk increased the rating score during the rebuttal period, but some concerns raised by Reviewer QBBM, 9d2R and eZ2K have not been well addressed. Several reviewers pointed out the obvious limitations of this work in terms of novelty and experimental results. The contributions of this manuscript are not impressive and significant, and the balance between performance and parameter overhead has not yet been fully achieved.

---

### Decision · Program_Chairs · 2025-01-22

Reject